# Learning with Selectively Labeled Data from Multiple Decision-makers

Jian Chen [* 1]   Zhehao Li [* 1]   Xiaojie Mao [* 1]

## Abstract

We study the problem of classification with selectively labeled data, whose distribution may differ from the full population due to historical decision-making. We exploit the fact that in many applications historical decisions were made by multiple decision-makers, each with different decision rules. We analyze this setup under a principled instrumental variable (IV) framework and rigorously study the identification of classification risk. We establish conditions for the exact identification of classification risk and derive tight partial identification bounds when exact identification fails. We further propose a unified cost-sensitive learning (UCL) approach to learn classifiers robust to selection bias in both identification settings. Finally, we theoretically and numerically validate the efficacy of our proposed method.

## 1. Introduction

The problem of *selective labels* is common in many decision-making applications involving human subjects. In these applications, each individual receives a certain decision, which in turn determines whether the individual's outcome label is observed. For example, in judicial bail, the label of interest is whether a defendant returns to court without committing another crime if released. However, this label cannot be observed if bail is denied. Similarly, in lending, the default status of a loan applicant cannot be observed if the loan is not approved. In hiring, a candidate's job performance cannot be observed if the candidate is not hired.

The selective label problem presents significant challenges for developing effective machine learning (ML) algorithms to support decision-making (Lakkaraju et al., 2017; Kleinberg et al., 2018; De-Arteaga et al., 2021). Specifically, labeled data may not be representative of the broader popu-

lation to which decisions will be applied, creating a classic *selection bias* issue. As a result, models trained on selectively labeled data may perform well on the observed subjects but fail to generalize to the full population when deployed in real-world settings. This challenge becomes even more critical when historical decisions relied on *unobservable* factors not captured in the available data, as is common when human decision-makers are involved. In such cases, the labeled data can differ substantially from the broader population due to unknown factors, further complicating the development of reliable ML models.

Our paper addresses the selective label problem by leveraging the heterogeneity of decision-making processes in many real-world applications. Specifically, we consider settings where decisions are made by multiple decision-makers with distinct decision rules, potentially leading to different outcomes for the same subject. Furthermore, since decision-makers often evaluate similar pools of subjects, each subject can be viewed as being quasi-randomly assigned to a specific decision-maker. This structure offers a way to mitigate the selective label problem: **a subject who remains unlabeled under one decision-maker might have been labeled if assigned to another**. For instance, judicial decisions are typically made by different judges who vary in their leniency when deciding whether to release the same defendant. The heterogeneous decision-maker framework has also been explored in prior research to address selection bias in model evaluation (e.g., Lakkaraju et al., 2017; Kleinberg et al., 2018; Rambachan et al., 2021; Arnold et al., 2022).

**Main Contributions** In this paper, we leverage the *instrumental variable* framework in causal inference to formalize the selective label problem with multiple heterogeneous decision-makers. This enables principled identification analyses to characterize the identifiability of classification risks from the observed data, revealing fundamental power and limits of the multiple decision-maker structure in tackling the selective label problem. This provides valuable insights beyond the heuristic approaches in the prior literature. Moreover, unlike the existing selective label literature that primarily focus on *evaluating* fixed models, our work focuses on *learning* a robust classification model from the selective label data.

Specifically, we describe the problem setup and the IV

---

*Alphabetical Order. [1]School of Economics and Management, Tsinghua University, Beijing, China. Correspondence to: Xiaojie Mao <maoxj@sem.tsinghua.edu.cn>.

*Proceedings of the 42nd International Conference on Machine Learning*, Vancouver, Canada. PMLR 267, 2025. Copyright 2025 by the author(s).

framework in Section 3. Then in Section 4 we establish identification conditions for the classification risk from the observed data with multiple decision-makers and highlight the identification assumptions are overly strong. In Section 5, we further derive tight bounds on the plausible value of classification risk when the exact identification fails. In Section 6, we propose a Unified Cost-sensitive Learning (UCL) algorithm to learn classifiers robust to selection bias in both identification settings. We briefly demonstrate the superior performance of our proposed method through numerical experiments in Section 7, with the comprehensive experiments in Appendix E. We defer theoretical guarantees for our method to Appendix D.

**Notations**  Define $x^+ = \max\{x, 0\}$, $x \wedge y = \min\{x, y\}$ and $x \vee y = \max\{x, y\}$ for $x, y \in \mathbb{R}$. We use the symbol $[N]$ as a shorthand for the set $\{1, 2, \cdots, N\}$ and $|\cdot|$ to denote the absolute value of number or cardinality of set.

## 2. Related Work

**Selective Label Problem**  The selective label problem arises when labels are observed only for individuals receiving certain decisions. Kleinberg et al. (2018) and Lakkaraju et al. (2017) study this in recidivism prediction using jail bail data, leveraging quasi-random judge assignment and variation in leniency to compare algorithmic recommendations with human decisions. By using a heuristic "contraction" method, they show that algorithms can outperform some judges in reducing recidivism. Bertsimas & Fazel-Zarandi (2022) extend this approach to immigration enforcement, using machine learning to assist detention decisions. They exploit quasi-random prison assignments and variation in release rates to enable credible counterfactual comparisons between strict and lenient detention practices. De-Arteaga et al. (2018; 2021) propose to leverage decision-maker consistency to impute the missing labels. They impute missing labels as negative when multiple decision-makers tend to consistently reject a case, assuming that consistent rejection implies a negative outcome. Compared to these prior works, we use an instrumental variable framework to analyze the selective label problem from a formal identification perspective. Moreover, these prior works all focus on model evaluation, while our work focuses on learning a robust classification model from selectively labeled data.

Our work also connects to the literature on counterfactual evaluation under unmeasured confounding. Rambachan et al. (2022) derive IV-based partial identification bounds for evaluating the predictive performance of given risk scores and cite selective labels as an example. In contrast, we focus on learning robust classification rules under both point and partial identification (see Theorems 4.4 and 5.3).

Another line of work addresses selective labels via online learning (Kilbertus et al., 2020; Wei, 2021; Yang et al., 2022), where the decision-maker can control whether to collect the true label of each unit by making appropriate decisions. The goal is to effectively balance the exploration and exploitation. In contrast, our work considers an offline setting where the dataset is already given and the decision-maker can no longer collect labels through exploration.

**Identification Analysis with IV**  This work builds on the instrumental variables (IV) literature in statistics and econometrics. IV methods are widely used to address unmeasured confounding in causal inference (Angrist & Pischke, 2009; Angrist & Imbens, 1995; Angrist et al., 1996). Several empirical studies, such as Kling (2006) and Mueller-Smith (2015), have already leveraged heterogeneity among decision-makers—e.g., judge assignment—as an IV. However, these analyses typically rely on *parametric linear* models, whereas our identification strategy is fully *nonparametric* and does not impose parametric assumptions.

Our *exact identification* analysis builds on Wang & Tchetgen (2018) and Cui & Tchetgen Tchetgen (2021), who study the identification of average treatment effects using binary-valued instruments. We extend their results to multi-valued IVs. Our *partial identification* results extend the classic bounds introduced by Manski (Manski, 1990; Manski & Pepper, 1998). We show that our bounds also reduce to those of Balke & Pearl (1994) in the case of a binary instrument and binary outcome. Furthermore, we establish the sharpness of our IV partial bounds using techniques from Kédagni & Mourifie (2017). For a broader overview of IV partial identification, see also Swanson et al. (2018).

**Individual Treatment Rule Learning with IV**  There is a growing body of work on individualized treatment rule (ITR) learning, which aims to determine the optimal treatment for each individual based on observed features. While early literature focuses on the unconfoundedness setting (e.g., Qian & Murphy, 2011; Zhao et al., 2012; Athey & Wager, 2021), recent literature also considers robust ITR learning under unmeasured confounding. In particular, Cui & Tchetgen Tchetgen (2021) and Pu & Zhang (2021) study ITR learning with instrumental variables (IV) under point and partial identification, respectively. Both are restricted to a binary IV and a binary treatment, and Pu & Zhang (2021) additionally focus on a binary outcome. Our work differs from them in that we consider a multi-valued IV and a multi-class outcome, aiming to learn a classifier that maps features to one label class rather than one treatment arm. Our work advances both works in several ways. Specifically, our point identification extends the identification in Cui & Tchetgen Tchetgen (2021) to a multi-valued IV. Our partial identification bounds extend the bounds in Pu & Zhang (2021) to a multi-valued IV and a multi-class outcome. Our resulting minimax learning formulation for a multi-class

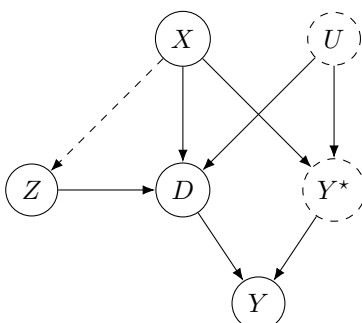

*Figure 1.* Causal graph of selective label problem. The dashed nodes $U$ and $Y^\star$ are unobserved. The dashed line from $X$ to $Z$ means that $Z$ is allowed but not required to be affected by $X$.

outcome is much more complicated than that in Pu & Zhang (2021), but we give a closed-form formulation for the inner maximization and provide a cost-sensitive learning reformulation. Importantly, our work also provides a unified cost-sensitive learning framework to tackle point and partial identification simultaneously rather than separately.

## 3. Problem Formulation

We address the selective label problem within a multiclass classification setting. Consider a dataset comprising $N$ units, indexed by $i = 1, \ldots, N$. For each unit $i$, let $Y_i^\star \in [K]$ denote the true label of interest for $K \geq 2$ classes. Additionally, let $X_i \in \mathcal{X}$ and $U_i \in \mathcal{U}$ represent observable and unobservable features, respectively, both of which may be correlated with $Y_i^\star$. Here, "observable" features are those accessible to the ML practitioner, while "unobservable" features are unavailable in the dataset.

In the selective label problem, the true label $Y_i^\star$ is not always observed; its observability depends on a binary decision $D_i \in \{0, 1\}$ made by a decision-maker based on the features $(X_i, U_i)$. Specifically, we consider $J \geq 2$ different decision-makers and let $Z_i \in [J]$ denote the specific decision-maker assigned to unit $i$. The observed label $Y_i$ is defined as:

$$Y_i = \begin{cases} Y_i^\star & \text{if } D_i = 1, \\ \text{NaN} & \text{if } D_i = 0. \end{cases} \quad (1)$$

When $D_i = 1$ (e.g., bail, loan approval, hiring), the observed label $Y_i$ is exactly the true label $Y_i^\star$, and when $D_i = 0$, the observed label is recorded as $Y_i = \text{NaN}$ (Not a Number), indicating that $Y_i^\star$ remains unobserved in this case. Thus, each unit's label is selectively observed according to the decision $D_i$ made by decision-maker $Z_i$.

We assume that $\{(Y_i, Y_i^\star, D_i, Z_i, X_i, U_i)\}_{i=1}^N$ represents independent and identically distributed samples from a population denoted by $(Y, Y^\star, D, Z, X, U)$. The causal structure of these variables is plotted in Figure 1. However, we can-

not observe all variables but instead only observe the data $S_N = \{(Y_i, D_i, X_i, Z_i)\}_{i=1}^N$.

### 3.1. Learning under Selective Labels

Our goal is to learn a classifier $t : \mathcal{X} \mapsto [K]$ that can accurately predict the true label $Y^\star$ based on observed features $X$. This classifier will aid in future decision-making processes through accurately predicting $Y^\star$. The effectiveness of the classifier upon deployment across the full population is evaluated using the expected risk with the zero-one loss:

$$\min_{t:\mathcal{X}\to[K]} \quad \mathcal{R}(t) := \mathbb{E}\left[\mathbb{I}(Y^\star \neq t(X))\right]. \quad (2)$$

Here $\mathbb{I}(\cdot)$ is an indicator function, which equals 1 if the event inside occurs and 0 otherwise. The expectation is taken over the unknown joint distribution of $Y^\star$ and $X$. For a given classifier $t$, the risk $\mathcal{R}(t)$ quantifies its misclassification error rate $\mathbb{P}(Y^\star \neq t(X))$.

Conventionally, one can train a classifier through empirical risk minimization (ERM), i.e., minimizing an empirical approximation for the classification risk $\mathcal{R}(t)$ based on the sample data. However, this is challenging under selective labels since the true label $Y^\star$ is not fully observed. One natural alternative is to perform ERM on the labeled subsample (subsample size denoted by $N_1$):

$$\min_{t:\mathcal{X}\mapsto[K]} \widehat{\mathcal{R}}_{\text{label}}(t) := \frac{1}{N_1} \sum_{i:D_i=1} \mathbb{I}\{Y_i \neq t(X_i)\}. \quad (3)$$

However, the empirical risk $\widehat{\mathcal{R}}_{\text{label}}(t)$ may not capture the true risk even when total sample size $N \to \infty$:

$$\widehat{\mathcal{R}}_{\text{select}}(t) \to \mathbb{P}(Y^\star \neq t(X) \mid D = 1) \neq \mathcal{R}(t).$$

This is known as a *selection bias* problem, where the distribution of $(Y^\star, X) \mid D = 1$ in the labeled subpopulation and that of $(Y^\star, X)$ in the full population are different. The selection bias arises because the historical decision $D$ can depend on the features $X$ and $U$ that are also correlated with the true outcome $Y^\star$. Hence, one has to adjust for both $X$ and $U$ to eliminate the selection bias, as formalized below.

**Assumption 3.1.** Suppose (1) **Selection-on-observables-and-unobservables**: $D \perp Y^\star \mid X, U$; (2) **Overlap**: $\mathbb{P}(D = 1 \mid X, U) > 0$ almost surely.

The condition $D \perp Y^\star \mid X, U$ in Assumption 3.1 means that the labeled subpopulation and the full population would have the same $Y^\star$ distribution if properly adjusting for $(X, U)$. The *overlap condition* means that the subpopulation with almost any $X, U$ value has a positive chance to be labeled. Both conditions are standard in the missing data literature (Little & Rubin, 2019). If features $X, U$ were both observed, then one could correct for the selection bias by adjusting for $X, U$ via many methods in the literature

(e.g., Rosenbaum & Rubin, 1983; Rubin, 2005; Imbens & Rubin, 2015). However, adjusting for the unobservable $U$ is infeasible, so selection bias remains a significant problem in our setting. This setting is referred to the missing-not-at-random (MNAR) problem in the missing data literature (Little & Rubin, 2019). Addressing MNAR is notoriously difficult and typically requires additional information.

### 3.2. Multiple Decision-makers and IV

To address the selective label problem, we draw on recent literature that leverage the fact that in many applications the selective labels involve multiple decision-makers with heterogeneous decision-making preference (Lakkaraju et al., 2017; Kleinberg et al., 2018; De-Arteaga et al., 2018). **The rationale behind this idea is that, the unobserved true label $Y^\star$ for a unit in the missing group ($D = 0$) could have been observed, if the unit had been assigned to a different decision-maker**. This is particularly useful when the assignment of decision-makers $Z$ is random, independent of the unobserved featured $U$. For instance, Lakkaraju et al. (2017); Kleinberg et al. (2018) consider that in the judicial bail, a defendant is randomly assigned to a judge, so a unit who is not granted bail (and thus unlabeled) could have been assigned to a more lenient judge, received bail, and consequently been labeled.

Although the previous works have already considered the structure of multiple decision-makers, they largely rely on some heuristic approaches. In fact, the assignment of decision-makers can be formalized as an instrumental variable (IV). As we will show shortly, this IV formalization allows us to understand the role of multiple decision-makers in a principled way. IV is a well-known concept in causal inference. Following standard literature (Angrist et al., 1996; Cui & Tchetgen Tchetgen, 2021; Wang & Tchetgen, 2018, e.g.,), a valid IV has to satisfy the following conditions.

**Assumption 3.2** (IV conditions). The assignment $Z$ satisfies the following three conditions: 1. **IV relevance**: $Z \not\perp D \mid X$; 2. **IV independence**: $Z \perp (U, Y^\star) \mid X$; 3. **Exclusion restriction**: $Z \perp Y \mid D, Y^\star$.

The IV conditions above are particularly suitable for the random assignment of decision-makers. The *IV relevance* means that even after controlling for the observed features $X$, different decision-makers could have systematically different decision rules so that the assignment $Z$ is dependent with the decision $D$. The *IV independence* trivially holds when the decision-maker assignment is random. The *exclusion restriction* automatically holds because the observed label $Y$ is completely determined by the decision $D$ and true label $Y^\star$ so it is independent with any other variable given $D, Y^\star$. We note that these conditions are also explicitly or implicitly assumed in the prior literature. Assumption 3.2 connects these conditions to the IV framework.

Based on the IV framework, we will next understand the power and limits of the multiple decision-maker structure in addressing the selective label problem. In particular, we will analyze the *identification* of the classification risk $\mathcal{R}(t)$ in Equation (2), studying under what conditions the risk can be determined from the distribution of observed data with multiple decision-makers.

## 4. Exact Identification of Classification Risk

In this section, we establish conditions under which the classification risk $\mathcal{R}(t)$ can be exactly identified from the observed data. This represents settings where consistently evaluating the classification risk is possible, if one properly leverage the multiple decision-maker structure.

Before delving into our detailed analysis, we define $\eta_k^\star(X) := \mathbb{P}(Y^\star = k \mid X)$ as the conditional probability that $Y^\star$ equals $k$, given the observable features $X$. The Bayes optimal classifier, which minimizes the classification risk $\mathcal{R}(t)$, is determined by $s^\star(X) := \arg\max_{k \in [K]} \eta_k^\star(X)$. In fact, $s^\star$ also minimizes the *excess risk*, defined as:

$$\mathcal{R}(t, s^\star) = \mathbb{P}(Y^\star \neq t(X)) - \mathbb{P}(Y^\star \neq s^\star(X)). \quad (4)$$

Minimizing $\mathcal{R}(t)$ over classifier $t$ is equivalent to minimizing $\mathcal{R}(t, s^\star)$ over $t$ as the Bayes optimal risk $\mathcal{R}(s^\star)$ is a constant. Therefore, we can equivalently study the identification of excess risk $\mathcal{R}(t, s^\star)$. The following lemma gives a weighted formulation of excess risk in terms of the conditional probabilities.

**Lemma 4.1.** *The excess risk $\mathcal{R}(t, s^\star)$ satisfies*

$$\mathcal{R}(t, s^\star) = \mathbb{E}\Big[ \sum_{k=1}^K \eta_k^\star(X) \cdot \big( \mathbb{I}_{s^\star(X)=k} - \mathbb{I}_{t(X)=k} \big) \Big].$$

Lemma 4.1 indicates to identify the excess risk we need first identify the conditional probabilities $\eta_k^\star \in [0, 1]$ of the latent $Y^\star$ from the observed data. We therefore focus on the identification of $\boldsymbol{\eta}^\star(x) = (\eta_1^\star(x), \ldots, \eta_K^\star(x))$ in the sequel.

**Identification of Conditional Probabilities** In Section 3.2, we show that the assignment of decision-makers $Z$ can be viewed as an IV. By adapting the IV identification theory in the causal inference literature (Wang & Tchetgen, 2018; Cui & Tchetgen Tchetgen, 2021), we can identify the classification risk under the following No Unmeasured Common Effect Modifiers (NUCEM) assumption.

**Assumption 4.2** (NUCEM). We have, for each $x \in \mathcal{X}$, $\mathrm{Cov}\left(\mathrm{Cov}(D, Z \mid X, U), \mathbb{P}(Y^\star = k \mid X, U) \mid x\right) = 0$.

**Theorem 4.3** (Point Identification). *Define $Y_k := \mathbb{I}\{Y = k\}$. Under Assumptions 3.1, 3.2 and 4.2, $\eta_k^\star(X)$ is identified,*

$$\eta_k^\star(X) = \mathrm{Cov}(DY_k, Z \mid X) / \mathrm{Cov}(D, Z \mid X). \quad (5)$$

*Conversely, if (5) holds, Assumption 4.2 is also satisfied.*

Theorem 4.3 shows that Assumption 4.2 is sufficient and necessary for identifying the conditional probability function $\eta_k^\star$'s through Equation (5). This identification equation connects the conditional probability of the latent true label $Y^\star$ with fully observed variables $D, Y, Z, X$. This lays the foundation for learning $\eta_k^\star$'s (and classification risk) from the observed data using the decision-maker assignment IV.

**Identification of Excess Classification Risk**  We can now identify the excess classification risk from observed data.

**Theorem 4.4.** *Based on Theorem 4.3, the excess classification risk $\mathcal{R}(t, s^\star)$ in Equation (4) is also identified:*

$$\mathcal{R}(t, s^\star) = \mathbb{E}\Big[\sum_{k=1}^{K} w_k(X)\mathbb{I}\{t(X) = k\}\Big], \qquad (6)$$

*where $w_k(x) := \max_{p\in[K]}(\eta_p^\star(x) - \eta_k^\star(x))$ denotes the weight function and $\eta_k^\star(x)$ is identified in Equation (5). This can be simplified for binary classification with $K = 2$:*

$$\mathcal{R}(t, s^\star) = \mathbb{E}\big[|w(X)|\mathbb{I}\{t(X) \neq \mathrm{sgn}[w(X)]\}\big], \quad (7)$$

*where $w(x) := \mathrm{Cov}(DY, Z \mid x)/\mathrm{Cov}(D, Z \mid x) - 1/2$ denotes the weight and $\mathrm{sgn} : \mathcal{X} \to \{\pm 1\}$ is a sign function.*

Theorem 4.4 does not only establish the identification of the excess classification risk under Assumption 4.2, but also formulates the identification into a weighted classification form. Specifically, minimizing the right hand side of Equation (6) corresponds to searching a cost-sensitive classifier, while minimizing the right hand side of Equation (7) corresponds to a weighted classification problem with the sign of $w(X)$ as the label and $|w(X)|$ as the misclassification cost. In Section 6, we will show that these formulations are useful in the design of effective learning algorithms.

**The NUCEM Identification Assumption**  The identification results in Theorems 4.3 and 4.4 crucially rely on Assumption 4.2, which generalizes the NUCEM assumption in Cui & Tchetgen Tchetgen (2021). Notably, their NUCEM assumption involves the difference of certain conditional expectations given the two different IV values, which heavily relies on the binary nature of IV. We consider a more general assumption that can tackle multi-valued and continuous IV, which strictly generalizes their framework. These identification analyses reveal that additional assumptions are needed to fully identify the classification risk, even when the decision-maker structure already satisfies the IV conditions in Assumption 3.2. This fact is not shown in the prior selective label literature without the formal IV framework.

To understand the NUCEM assumption, it is instructive to consider examples where this assumption holds.

*Example* 1 (Full Information). If all features needed to predict the true label $Y^\star$ are recorded, then $\mathbb{E}[Y^\star \mid X, U] = \mathbb{E}[Y^\star \mid X]$ almost surely and Assumption 4.2 holds.

*Example* 2 (Separable and Independent Unobservables). Suppose unobservables $U$ can be divided into two categories: $U_1$ influences the decision $D$, and $U_2$ affects the outcome $Y^\star$. Hence, we have $\mathrm{Cov}(D, Z \mid X, U) = \mathrm{Cov}(D, Z \mid X, U_1)$ and $\mathbb{E}[Y^\star \mid X, U] = \mathbb{E}[Y^\star \mid X, U_2]$ almost surely. If $U_1$ and $U_2$ are (conditionally) independent, then Assumption 4.2 holds.

*Example* 3 (Additive Decision Probability). Suppose that for every decision-maker $j \in [J]$, the conditional decision probability satisfies $\mathbb{P}(D = 1 \mid U, X, Z = j) = g_j(X) + q(U)$ for a common function $q$ and potentially different functions $g_j$. That is, all decision-makers use the unobservables $U$ in a common and additive way. Under this condition, we can prove that Assumption 4.2 also holds (see Appendix A.2).

The examples above show that the NUCEM assumption needs to impose restrictions on the impact of unobserved features $U$ on either the decision $D$ or the true label $Y^\star$ or both. However, these restrictions may be too strong in practice so the NUCEM assumption may fail. As a result, the identification formula in Theorems 4.3 and 4.4 are no longer valid. In the next section, we will drop the NUCEM assumption and only impose some mild conditions, leading to weaker identification results accordingly.

# 5. Partial Identification of Classification Risk

The last section shows that the exact identification of classification risk often requires assumptions that are untenable in practice (see Assumption 4.2). Hence it is desirable to avoid such stringent assumptions and instead consider weaker and more reasonable assumptions. Nevertheless, without the stringent assumptions, it is generally impossible to exactly identify the classification risk from the observed data. Instead, there may exist a range of plausible values of the classification risk, all compatible with the observed data. This plausible range is characterized by the so-called *partial identification bounds*.

Below we impose a mild bound condition on the conditional probability of the true label for the partial bound analysis.

**Assumption 5.1.** For each fixed $x$ and $k$, there exist two known functions $a_k(x) \in [0, 1]$ and $b_k(x) \in [0, 1]$ such that $a_k(x) \leq \mathbb{P}(Y^\star = k \mid U, X = x) \leq b_k(x)$ almost surely.

Assumption 5.1 mandates known lower and partial bounds on the conditional probability $\mathbb{P}(Y^\star = k \mid U, X = x)$, while permitting arbitrary dependence on unobserved features $U$. This assumption is mild—it is trivially satisfied by setting $a_k(x) = 0$ and $b_k(x) = 1$. In all our experiments, we adopt this default choice for all $k \in [K]$. With these parameters, we derive explicit partial bound for $\eta_k^\star(X)$.

**Partial Identification of Conditional Probabilities**  Based on this condition, we can derive *tight* IV partial identi-

fication bounds on the conditional probability $\eta_k^\star(X) = \mathbb{P}(Y^\star = k \mid X)$ for each $k \in [K]$.

**Theorem 5.2.** *Define $Y_k := \mathbb{I}\{Y = k\}$ and assume Assumptions 3.1 and 3.2 and Assumption 5.1 hold. Then for any $x \in \mathcal{X}$ and $k \in [K]$, the value of $\eta_k^\star(x)$ must fall within the interval $[l_k(x), u_k(x)]$, where*

$$l_k(x) := \max_{z \in [J]} \mathbb{E}[DY_k + a_k(x)(1 - D) \mid X = x, Z = z],$$
$$u_k(x) := \min_{z \in [J]} \mathbb{E}[DY_k + b_k(x)(1 - D) \mid X = x, Z = z].$$

*Conversely, any value within the interval $[l_k(x), u_k(x)]$ can be a plausible value of $\eta_k^\star(x)$. Moreover, the interval endpoints satisfy that $a_k(x) \leq l_k(x) \leq u_k(x) \leq b_k(x)$.*

Theorem 5.2 generalizes the well-known Balke and Pearl's bound and Manski's bound in Balke & Pearl (1994); Manski & Pepper (1998). Specifically, our bounds coincide with Balke and Pearl' bounds when applied to a binary IV (see Appendix B.2). Furthermore, if $a(X), b(X)$ are two constants independent of $X$ (such as 0, 1), our bounds recover Manski's bounds. Theorem 5.2 also demonstrates the tightness of our IV partial bounds, in the sense that our bounds tightly give the range of all plausible values of the conditional probability $\eta_k^\star$'s (see Appendix B.1).

---

**Algorithm 1** Unified Cost-sensitive Learning

**Require:** function class $\mathcal{H}$, mode $\in \{\text{point}, \text{partial}\}$, data $S_N = \{(Y_i, X_i, D_i, Z_i)\}_{i=1}^N$, cross-fitting folds $L$.

1: Randomly split the data into $L$ (even) batches with indices denoted by $S_N^1, \ldots, S_N^L$ respectively.
2: **for** each batch $l \in [L]$ **do**
3:    **if** mode = point **then**
4:       For $k \in [K]$, use dataset $S_N \setminus S_N^l$ to estimate the weight $\widehat{w}_k^{[l]}(x) = \max_{p \in [K]} \left( \widehat{\eta}_p(x) - \widehat{\eta}_k(x) \right)$.
5:    **else if** mode = partial **then**
6:       For $k \in [K]$, use dataset $S_N \setminus S_N^l$ to estimate the weight $\widehat{w}_k^{[l]}(x) = \max_{p \neq k} \left( \widehat{u}_p(x) - \widehat{l}_k(x) \right)^+$.
7:    **end if**
8: **end for**
9: Construct the empirical risk for each batch $l \in [L]$:

$$\widehat{\mathcal{R}}_{\exp}^{[l]}(\boldsymbol{h}) = \frac{1}{|I_l|} \sum_{i \in I_l} \sum_{k=1}^K \widehat{w}_k^{[l]}(X_i) \frac{\exp(h_k(X_i))}{\sum_{p=1}^K \exp(h_p(X_i))}.$$

10: Return the score function $\widehat{\boldsymbol{h}}$ by minimizing:

$$\widehat{\boldsymbol{h}} \in \underset{h \in \mathcal{H}}{\arg\min} \frac{1}{L} \sum_{l=1}^L \widehat{\mathcal{R}}_{\exp}^{[l]}(\boldsymbol{h}). \qquad (8)$$

---

**Partial Identification of Excess Classification Risk** When $\{\eta_k^\star\}_{k \in [K]}$ are only partially identified within $[l_k, u_k]$ for

each $k \in [K]$, the excess classification risk $\mathcal{R}(t, s^\star)$ is also partially identified. Define the feasible region for the conditional probability vector $\boldsymbol{\eta}^\star = (\eta_1, \ldots, \eta_K)$ as

$$S := \left\{ \boldsymbol{\eta} : \|\boldsymbol{\eta}\|_1 = 1, \eta_k \in [l_k, u_k], k \in [K] \right\}, \quad (9)$$

Then the range of plausible value of the excess risk $\mathcal{R}(t, s^\star)$ is given by the interval $[\underline{\mathcal{R}}(t), \overline{\mathcal{R}}(t)]$, where

$$\overline{\mathcal{R}}(t) := \mathbb{E}\left[ \max_{\boldsymbol{\eta} \in S} \sum_{k=1}^K \eta_k(X) \left( \mathbb{I}_{s^\star(X)=k} - \mathbb{I}_{t(X)=k} \right) \right] \quad (10)$$

and $\underline{\mathcal{R}}(t)$ is symmetrically defined by changing the maximization over $\eta \in S$ into minimization over the same set. We are particularly interested in the upper bound $\overline{\mathcal{R}}(t)$ as it gives the worst-case risk of a classifier $t$. Minimizing this upper bound, i.e., $\min_{t: \mathcal{X} \to [K]} \overline{\mathcal{R}}(t)$, hence leads to a robust classifier safeguarding the perfrmance in the worst case. Below we further provide a reformulation of the worst-case risk under a mild regularity condition.

**Theorem 5.3.** *Suppose conditions in Theorem 5.2 hold and the set $S$ defined in eq. (9) is non-empty. Given the bounds $[l_k, u_k]$ in Theorem 5.2, we define the "realizable" partial bounds for $\eta_k^\star$ as*

$$\tilde{l}_k(x) := [1 - \sum_{p \neq k} u_p(x)] \vee l_k(x),$$
$$\tilde{u}_k(x) := [1 - \sum_{p \neq k} l_p(x)] \wedge u_k(x).$$

*We have*

$$\overline{\mathcal{R}}(t) = \mathbb{E}\left[ \sum_{k=1}^K w_k(X) \mathbb{I}\{t(X) = k\} \right],$$

*where $w_k(x) := \max_{p \neq k, p \in [K]} \left( \tilde{u}_p(x) - \tilde{l}_k(x) \right)^+$ denotes the weight function. This can be simplified for binary classification problems:*

$$\overline{\mathcal{R}}(t) = \mathbb{E}\left[ |w(X)| \mathbb{I}\{t(X) \neq \text{sgn}[w(X)]\} \right] + C,$$

*where $C$ is a constant not relying on the classifier $t$ and $w(x) := \max\{2u_1(x) - 1, 0\} + \min\{2l_1(x) - 1, 0\}$.*

The non-emptiness of set $S$ defined in Equation (9) is a mild regularity condition. It is easy to verify from Theorem 5.2 that this condition is automatically satisfied for $a_k(X) = 0$ and $b_k(X) = 1$ for all $k \in [K]$. Like Theorem 4.4, here Theorem 5.3 formulates the worst-case risk into a cost-sensitive form for general multi-class classification problems and a weighted classification form for binary classification problems. This reformulation will be handy for the learning algorithm design in the next section.

**Discussions** Notably, the partial identification results in this section are quite different than the exact identification results in Section 4. Section 4 relies on a strong NUCEM

identification assumption (Assumption 4.2). This strong assumption has a strong implication: we can pinpoint the exact value of classification risk from the observed data with the decision-maker assignment as an IV. However, the key NUCEM identification assumption is very strong and may be often implausible in practice. In this section, we instead consider a much weaker bound condition in Assumption 5.1. This condition is arguably more plausible yet meanwhile it also has weaker identification implication. Under this condition, even with the decision-maker assignment as a valid IV, the classification risk is still intrinsically ambiguous, so we can at most get a range of its values from the observed data. In the next subsection, we will design algorithms to learn robust classifiers by approximately minimizing the worst-case classification risk. These analyses reveal the power and limits of the multiple decision-maker data structure in tackling the selective label problem, which highlights the value of the principled IV framework adopted in our paper.

# 6. Unified Cost-sensitive Learning

In this section, we propose a unified cost-sensitive learning (UCL) algorithm tailored to learn a classifier from selectively labeled data. This algorithm accommodates both the exact identification setting in Section 4 and the partial identification setting in Section 5 in a unified manner.

Typically, a classifier $t(x)$ is represented by some score function $\boldsymbol{h} : \mathcal{X} \to \mathbb{R}^K$ through $t(x) = \arg\max_{k \in [K]} h_k(x)$. Here for any given $x$, $h_k(x)$ quantifies the score of class $k$.

According to Theorems 4.4 and 5.3, the excess classification risk of a classifier $t$ with score function $\boldsymbol{h}$ in the two identification settings can be written into the common weighted form as follows:

$$\mathcal{R}(\boldsymbol{h}, \boldsymbol{w}) = \mathbb{E}\Big[ \sum_{k=1}^{K} w_k(X) \mathbb{I}\{\arg\max_{p \in [K]} h_p(x) = k\} \Big]. \quad (11)$$

Specifically, for each $k$, we set $w_k(x) = \max_{p \in [K]} (\eta_p^\star(x) - \eta_k^\star(x))$ to recover the exact identification in Theorem 4.4 and set $w_k(x) = \max_{p \neq k, p \in [K]} (\tilde{u}_p(x) - \tilde{l}_k(x))^+$ to recover the worst-case excess risk $\overline{\mathcal{R}}(t)$ in Theorem 5.3. In this section, we aim to minimize these risk objectives to learn effective classifiers. This corresponds to a cost-sensitive multi-classification problem (Pires et al., 2013).

Theorems 4.4 and 5.3 also show that for binary classification problems with $K = 2$, the risk function simplifies to:

$$\mathcal{R}(h, w) = \mathbb{E}\big[ \big| w(X) \big| \mathbb{I}\{ \operatorname{sgn}[h(x)] \neq \operatorname{sgn}[w(X)] \} \big].$$

By specifying $w(x) = \frac{\operatorname{Cov}(DY, Z|x)}{\operatorname{Cov}(D, Z|x)} - 1/2$, we recover the classification risk in the exact identification setting. By specifying $w(x) = \max\{2u_1(x) - 1, 0\} + \min\{2l_1(x) - 1, 0\}$, we recover the worst-case risk (up to some constants) under partial identification. Minimizing these risks correspond to

solving weighted binary classification with a pseudo label $\operatorname{sgn}[w(X)]$ and misclassification cost $|w(X)|$.

## 6.1. Calibrated Surrogate Risk

The cost-sensitive classification risk $\mathcal{R}(\boldsymbol{h}, \boldsymbol{w})$ in Equation (11) involves non-convex and non-smooth indicator functions $\mathbb{I}\{\cdot\}$, making the optimization challenging. To address this issue, we introduce a surrogate risk $\mathcal{R}_{\exp}(\boldsymbol{h}, w)$ that replaces the indicator function with the softmax function. This substitution allows us to define the surrogate for cost-sensitive classification risk:

$$\mathcal{R}_{\exp}(\boldsymbol{h}, \boldsymbol{w}) = \mathbb{E}\Big[ \sum_{k=1}^{K} w_k(X) \frac{\exp(h_k(X))}{\sum_{p=1}^{K} \exp(h_p(X))} \Big]. \quad (12)$$

For the binary case, we introduce the weighted $\phi$-risk as

$$\mathcal{R}_\phi(h, w) = \mathbb{E}\Big[ \big| w(X) \big| \cdot \phi\big( h(X) \cdot \operatorname{sgn}[w(X)] \big) \Big]. \quad (13)$$

There are many possible choices for the convex surrogate loss $\phi$, including the Hinge loss $\phi(\alpha) = \max\{1 - \alpha, 0\}$, the logistic loss $\phi(\alpha) = \log(1 + e^{-\alpha})$, and the exponential loss $\phi(\alpha) = e^{-\alpha}$, and so on (Bartlett et al., 2006).

Theorem 6.1 establishes a calibration bound linking the cost-sensitive risk $\mathcal{R}(\boldsymbol{h}, \boldsymbol{w})$ to its surrogate $\mathcal{R}_{\exp}(\boldsymbol{h}, \boldsymbol{w})$.

**Theorem 6.1** (Calibration Bound for Cost-sensitive Risk). *For any given weight function $\boldsymbol{w} : \mathcal{X} \to \mathbb{R}^K$, we have*

$$\mathcal{R}(\boldsymbol{h}, \boldsymbol{w}) - \inf_{\boldsymbol{h}} \mathcal{R}(\boldsymbol{h}, \boldsymbol{w}) \leq K\big[ \mathcal{R}_{\exp}(\boldsymbol{h}, \boldsymbol{w}) - \inf_{\boldsymbol{h}} \mathcal{R}_{\exp}(\boldsymbol{h}, \boldsymbol{w}) \big].$$

For binary case, calibration bound can also be established for $\mathcal{R}_\phi(h, w)$ (Bartlett et al., 2006; Tewari & Bartlett, 2007).

**Proposition 6.2** (Calibration Bound for Weighted Risk). *For any fixed weight function $w : \mathcal{X} \to \mathbb{R}$, we have*

$$\mathcal{R}(h, w) - \inf_{h} \mathcal{R}(h, w) \leq \mathcal{R}_\phi(h; w) - \inf_{h} \mathcal{R}_\phi(h, w).$$

Theorem 6.1 and Proposition 6.2 show that the target risks can be well-bounded by the excess surrogate risks. Hence we can use sample data to approximately minimize the surrogate risks, which can be efficiently solved.

## 6.2. Empirical Risk Minimization

We now propose Algorithm 1 to minimize the empirical surrogate risk $\mathcal{R}_{\exp}(\boldsymbol{h}, \boldsymbol{w})$ in Equation (12) using observed data $S_N$. We also establish the finite-sample error bound for the resulting score function $\boldsymbol{h}$ (see Appendix D).

**Weights Estimation** The weight functions $\{w_k\}_{k=1}^{K}$ differ between point-identified and partial-identified settings, requiring separate estimation. For $\operatorname{mode} = \operatorname{point}$, we first estimate $\operatorname{Cov}(D, Z \mid X)$ and $\operatorname{Cov}(DY_k, Z \mid X)$ in eq. (5),

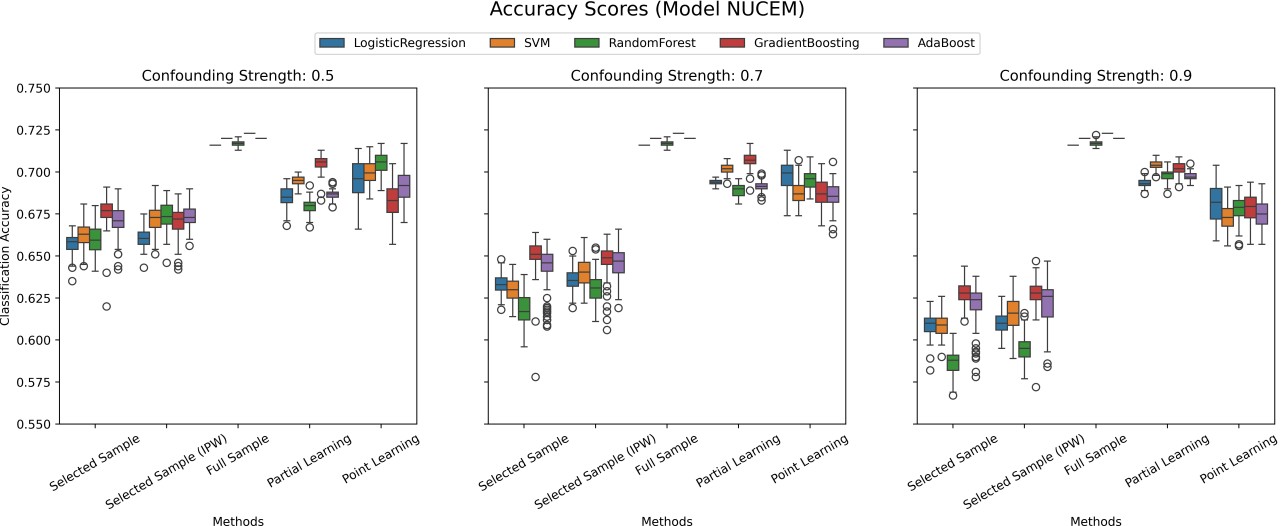

*Figure 2.* The testing accuracy of methods with $\alpha \in \{0.5, 0.7, 0.9\}$ for model NUCEM in FICO dataset.

then compute their ratio. For example, $\text{Cov}(DY_k, Z \mid x)$ can be expressed as $\mathbb{E}[DZY_k \mid x] - \mathbb{E}[DY_k \mid x] \cdot \mathbb{E}[Z \mid x]$, with conditional expectations estimated via regression on $X = x$. Alternatively, forest methods, such as Generalized Random Forests, can directly estimate the conditional covariance ratio in one step (Athey et al., 2019).

For $\text{mode} = \text{partial}$, we estimate the bounds $l_k$ and $u_k$ for each $k \in [K]$ as defined in Theorem 5.2, involving conditional expectations $\mathbb{E}[DY_k \mid x, z]$ and $\mathbb{E}[D \mid x, z]$ via regression. These bounds are then used to construct $w_k(x)$. The precision of weights estimation do affect the accuracy of downstream classification task. To illustrate this, we implement the ablation study by injecting the noise to the weights estimation. See Appendix E.2 for the discussion.

**Cross-fitting** In Algorithm 1, we divide the dataset into $L$ folds and apply the *cross-fitting* to estimate these weight functions. This approach has been widely used in statistical inference and learning with ML nuisance function estimators (e.g., Chernozhukov et al., 2018; Athey & Wager, 2021), which effectively alleviates the over-fitting bias by avoiding using the same data to both train weight function estimators and evaluate the weight function values.

The cross-fitting estimation entails some additional computational costs, but this typically only involves a series of standard regression/classification fitting, which is usually manageable and can be accelerated by parallel computation.

**Classifier Optimization** The empirical surrogate risk in Equation (8) defines a cost-sensitive classification problem. Once the weight function $\{w_k\}_{k=1}^K$ are estimated, we can then solve for the score function $\boldsymbol{h}$, which can be implemented using $\text{PyTorch}$ by defining a cost-sensitive loss function with $\text{softmax}$ being the output layer. We can also incorporate regularization in Equation (8) to reduce overfitting. In the binary setting, we can simply solve a weighted classification problem with $\text{sgn}[\hat{w}(X)]$'s as the labels and $|\hat{w}(X)|$ as weights, and this can be easily implemented by off-the-shelf ML packages that take additional weight inputs. For example, considering the surrogate risk in Equation (13) with $\phi$ as the logistic loss, we can simply run a logistic regression with the aforementioned labels and weights.

# 7. Numeric Experiments

We conduct experiments on a synthetic dataset with a multiclass label ($K = 3$) and a semi-synthetic dataset with a binary label ($K = 2$). Below, we briefly summarize our experiment on the semi-synthetic data, with comprehensive results provided in Appendix E. Refers the code and data to https://github.com/Zhehao97/Learning-Selective-Labels.git.

**Semi-synthetic Dataset** This dataset consists of 10459 observations of approved home loan applications. The dataset records whether the applicant repays the loan within 90 days overdue, which we view as the true label $Y^*$, and various transaction information of the bank account. The dataset also includes a variable called $\text{ExternalRisk}$, which is a risk score assigned to each application by a proprietary algorithm. We use $\text{ExternalRisk}$ and all transaction features as the observed features $X$. In this dataset, the label of interest is fully observed, we synthetically create selective labels on top of the dataset. Specifically, we simulate $J = 10$ decision-makers (e.g., bank officers who handle the loan applications) and randomly assign one to each case. We generate the decision $D$ from a Bernoulli distribution with a success rate $p_D := \mathbb{P}(D = 1 \mid X, U)$ that depends on an

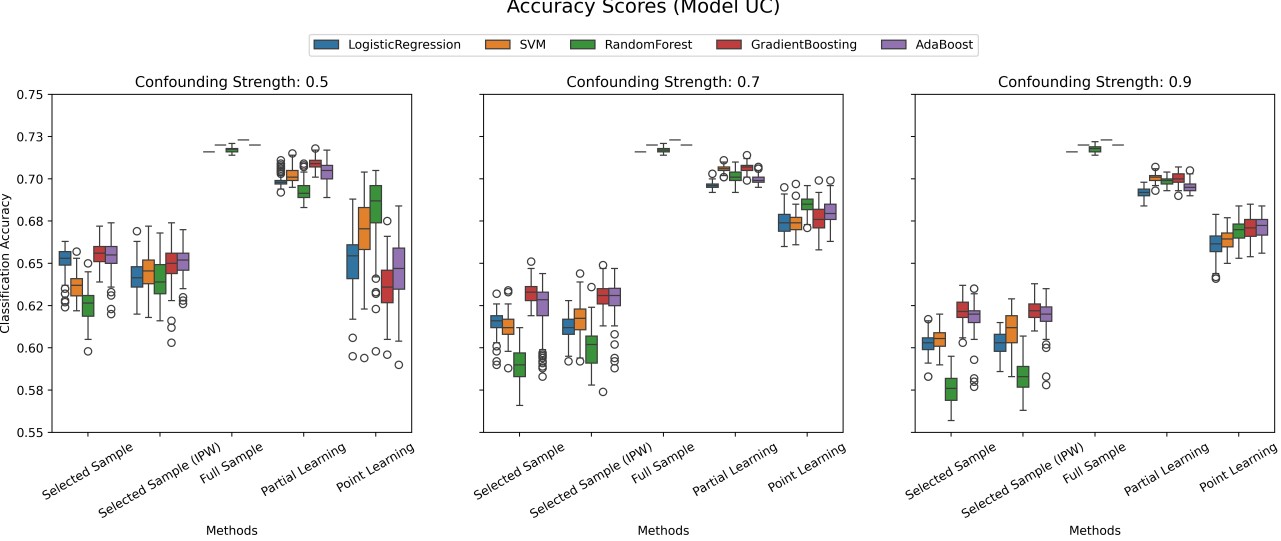

*Figure 3.* The testing accuracy of methods with $\alpha \in \{0.5, 0.7, 0.9\}$ for model UC in FICO dataset.

"unobservable" variable $U$, the decision-maker identity $Z$, and the ExternalRisk variable:

$$p_D^{\text{NUCEM}} = \alpha\sigma(U) + (1 - \alpha)\sigma((1 + Z) \cdot \text{ExternalRisk}),$$
$$p_D^{\text{UC}} = \sigma(\alpha U + (1 - \alpha)(1 + Z) \cdot \text{ExternalRisk}).$$

Here we define $\sigma(t) = \frac{1}{1+\exp(-t)}$. The parameter $\alpha_D \in (0, 1)$ controls the impact of $U$ on the labeling process and thus the degree of selection bias. The unobservables $U$ is constructed as the residual of a HistGradientBoosting regression of $Y^\star$ with respect to $X$, which is naturally dependent with $Y^\star$. Finally, we blind the true label $Y^\star$ for observations with $D = 0$.

**Methods and Evaluation** We randomly split our data into training and testing sets at $7 : 3$ ratio. We compare five methods: SelectedSample (training classifiers only on the labeled sample), SelectedSample-IPW (using labeled sample only but with additional inverse propensity weighting to correct selection bias due to observed features $X$), FullSample (ideal benchmark that observes all labels on the full sample), PointLearning and PartialLearning (our proposed methods based on the exact/point and partially identified risks using decision-maker assignment as IV). The first three methods serve as the benchmark of our methods. For each method, we try multiple algorithms including AdaBoost, Gradient Boosting, Logistic Regression, Random Forest, and SVM. All hyperparameters are chosen via 5-fold cross-validation, and we evaluate the classification accuracy of classifiers on the full-labeled testing data.

**Results and Discussions** Figures 2 and 3 reports the testing accuracy of each method in 50 replications of the experiment when $\alpha \in \{0.5, 0.7, 0.9\}$ under both NUCEM and UC models. We find the performance of SelectedSample(IPW)

is comparable with the baseline method which run classification algorithms directly on the selective labeled data. Moreover, we observe that as the degree of selection bias $\alpha$ grows, the gains from using our proposed method relative to the SelectedSample baseline also grows, especially for the PartialLearning method. Interestingly, the PartialLearning method has better performance even under NUCEM model when the point identification condition holds. This is perhaps because the PointLearning method requires estimating a conditional variance ratio, which is often difficult in practice. In contrast, the PartialLearning method only requires estimating conditional expectations and tends to be more stable.

In addition, we conduct experiments on a synthetic dataset with a multiclass label in Appendix E. The results demonstrate that our methods, particularly PartialLearning, achieve higher classification accuracy than benchmark methods under varying degrees of selection bias.

## 8. Conclusion

We address multiclass classification with selectively labeled data, where the label distribution deviates from the full population due to historical decision-making. Leveraging variations in decision rules across multiple decision-makers, we analyze this problem using an instrumental variable (IV) framework, providing conditions for exact classification risk identification. When exact identification is unattainable, we derive sharp partial risk bounds. To address label selection bias, we propose a unified cost-sensitive learning (UCL) approach, supported by theoretical guarantees, and demonstrate its effectiveness through comprehensive numerical experiments.

## Acknowledgements

We thank the anonymous reviewers and meta-reviewers of ICML 2025 for their valuable feedback, which led to a greatly improved manuscript. Xiaojie Mao acknowledges the support from National Natural Science Foundation of China (grant numbers 72322001, 72201150, and 72293561) and National Key R&D Program of China (grant number 2022ZD0116700).

## Impact Statement

This paper presents work whose goal is to advance the field of Machine Learning. There are many potential societal consequences of our work, none which we feel must be specifically highlighted here.

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

# A. Supplements for Point Identification

## A.1. Proofs in Section 4

*Proof of Lemma 4.1.* By the definition of excess risk in Equation (4), we have

$$
\begin{aligned}
\mathcal{R}(t, s^\star) &= \mathbb{P}(Y_i^\star \neq t(X_i)) - \mathbb{P}(Y_i^\star \neq s^\star(X_i)) \\
&= \mathbb{E}\Big[\mathbb{P}(Y_i^\star \neq t(X_i) \mid X_i) - \mathbb{P}(Y_i^\star \neq s^\star(X_i) \mid X_i)\Big] \\
&= \mathbb{E}\left[\sum_{k=1}^{K} \eta_k^\star(X_i) \cdot \Big(\mathbb{P}(k \neq t(X_i) \mid X_i) - \mathbb{P}(k \neq s^\star(X_i) \mid X_i)\Big)\right] \\
&= \mathbb{E}\left[\sum_{k=1}^{K} \eta_k^\star(X_i) \cdot \Big(\mathbb{P}(s^\star(X_i) = k \mid X_i) - \mathbb{P}(t(X_i) = k \mid X_i)\Big)\right] \\
&= \mathbb{E}\left[\sum_{k=1}^{K} \eta_k^\star(X_i) \cdot \mathbb{E}\Big[\mathbb{I}\{s^\star(X_i) = k\} - \mathbb{I}\{t(X_i) = k\} \mid X_i\Big]\right] \\
&= \mathbb{E}\left[\sum_{k=1}^{K} \eta_k^\star(X_i) \cdot \Big(\mathbb{I}\{s^\star(X_i) = k\} - \mathbb{I}\{t(X_i) = k\}\Big)\right] \\
&= \mathbb{E}\left[\sum_{k=1}^{K} \eta_k^\star(X_i) \cdot \Big(\mathbb{I}\{\arg\max_{p \in [K]} \eta_p^\star(X_i) = k\} - \mathbb{I}\{t(X_i) = k\}\Big)\right].
\end{aligned}
$$

We apply the iterated law of expectation in second equality, and the total law of expectation in third equality. The fourth equality comes from the sum-to-one constraint of probability. The fifth equality is established from the definition of indicator function and the sixth equality follows again from the iterated law of expectation. Finally we apply the definition of Bayes optimal $s^\star(X_i) = \arg\max_{k \in [K]} \eta_k^\star(X_i)$.

$\square$

*Proof of Theorem 4.3.* In this part, we prove that, under Assumptions 3.1 and 3.2, Assumption 4.2 is the necessary and sufficient condition of the exact identification of $\mathbb{P}(Y^\star = k \mid X)$. To simplify the notation, we denote two new variables $Y_k^\star := \mathbb{I}\{Y^\star = k\}$ and $Y_k := \mathbb{I}\{Y = k\}$.

- **Sufficiency**: For any $z \in \mathcal{Z}$, the conditional probability $\mathbb{P}(Y^\star = k \mid X, Z = z) = \mathbb{E}[Y_k^\star \mid X, Z = z]$ could be expressed as

$$
\begin{aligned}
\mathbb{E}[Y_k^\star \mid X, Z = z] &= \mathbb{E}\big[\mathbb{E}[Y_k^\star \mid X, U, Z = z] \mid X, Z = z\big] \\
&= \mathbb{E}\big[\mathbb{E}[D Y_k^\star \mid X, U, Z = z] \mid X, Z = z\big] + \mathbb{E}\big[\mathbb{E}[(1-D) Y_k^\star \mid X, U, Z = z] \mid X, Z = z\big] \\
&= \mathbb{E}\big[\mathbb{E}[D Y_k \mid X, U, Z = z] \mid X, Z = z\big] + \mathbb{E}\big[\mathbb{E}[(1-D) Y_k^\star \mid X, U, Z = z] \mid X, Z = z\big] \\
&= \mathbb{E}\big[\mathbb{E}[D Y_k \mid X, U, Z = z] \mid X, Z = z\big] + \mathbb{E}\big[\mathbb{E}[Y_k^\star \mid X, U, Z = z] \cdot \mathbb{E}[(1-D) \mid X, U, Z = z] \mid X, Z = z\big] \\
&= \mathbb{E}\big[\mathbb{E}[D Y_k \mid X, U, Z = z] \mid X, Z = z\big] + \mathbb{E}\big[\mathbb{E}[Y_k^\star \mid X, U] \cdot \mathbb{E}[(1-D) \mid X, U, Z = z] \mid X, Z = z\big] \\
&= \mathbb{E}[D Y_k \mid X, Z = z] + \mathbb{E}[Y_k^\star \mid X, Z = z] - \mathbb{E}\big[\mathbb{E}[Y_k^\star \mid X, U] \cdot \mathbb{E}[D \mid X, U, Z = z]\big].
\end{aligned} \tag{14}
$$

In the first and second equalities, we use the iterated law of expectation and the linearity of expectation. The third equality follows from the consistency between observed label and true label from Equation (1), that is, $Y_k = Y_k^\star$ when $D = 1$. The fourth equality follows from the unconfoundedness of $Y_k^\star$ and $D$ given $X$ and $U$, which is guaranteed by the general unconfoundedness ($Y^\star \perp D \mid X, U$) in Assumption 3.1. We use the IV independence ($Z \perp Y_k^\star \mid X$) from Assumption 3.2 in the fifth equality. The last equality follows again by the iterated law of expectation and linearity of expectation.

By subtracting $\mathbb{E}[Y_k^\star \mid X]$ on both sides of Equation (14), we have for any $z \in \mathcal{Z}$,

$$
\begin{aligned}
0 &= \mathbb{E}\big[D Y_k \mid X, Z = z\big] - \mathbb{E}\big[\mathbb{E}[Y_k^\star \mid X, U] \cdot \mathbb{E}[D \mid X, U, Z = z] \mid X, Z = z\big] \\
&= \mathbb{E}\big[D Y_k \mid X, Z = z\big] - \mathbb{E}\big[\mathbb{E}[Y_k^\star \mid X, U] \cdot \mathbb{E}[D \mid X, U, Z = z] \mid X\big],
\end{aligned}
$$

where the last equality follows from the IV independence $(Z \perp U \mid X)$ from Assumption 3.2. Now, by multiplying the weight $z \cdot \mathbb{P}(Z = z \mid X)$ on both sides of equation above, we have

$$
\begin{aligned}
0 &= \mathbb{E}[DY_k \mid X, Z = z] - \mathbb{E}\big[\mathbb{E}[Y_k^\star \mid X, U] \cdot \mathbb{E}[D \mid X, U, Z = z] \mid X\big] \\
&= \mathbb{E}[zDY_k \mid X, Z = z] \cdot \mathbb{P}(Z = z \mid X) - \mathbb{E}\big[\mathbb{E}[Y_k^\star \mid X, U] \cdot \mathbb{E}[zD \mid X, U, Z = z] \cdot \mathbb{P}(Z = z \mid X) \mid X\big] \\
&= \mathbb{E}[zDY_k \mid X, Z = z] \cdot \mathbb{P}(Z = z \mid X) - \mathbb{E}\big[\mathbb{E}[Y_k^\star \mid X, U] \cdot \mathbb{E}[zD \mid X, U, Z = z] \cdot \mathbb{P}(Z = z \mid X, U) \mid X\big].
\end{aligned}
$$

He we apply the IV independence $(Z \perp U \mid X)$ again in the last equality above. Taking the summation (or integral) for $z$ over $\mathcal{Z}$ yields the following results:

$$
0 = \mathbb{E}[ZDY_k \mid X] - \mathbb{E}\big[\mathbb{E}[Y_k^\star \mid X, U] \cdot \mathbb{E}[ZD \mid X, U] \mid X\big].
$$

Moreover, observe that

$$
\begin{cases}
\mathbb{E}[ZDY_k \mid X] = \mathrm{Cov}(DY_k, Z \mid X) + \mathbb{E}[DY_k \mid X] \cdot \mathbb{E}[Z \mid X] \\
\mathbb{E}[ZD \mid X, U] = \mathrm{Cov}(D, Z \mid X, U) + \mathbb{E}[D \mid X, U] \cdot \mathbb{E}[Z \mid X, U],
\end{cases}
$$

we have

$$
\begin{aligned}
0 = \underbrace{\mathrm{Cov}(DY_k, Z \mid X)}_{\text{I}} + \underbrace{\mathbb{E}[DY_k^\star \mid X] \cdot \mathbb{E}[Z \mid X]}_{\text{II}} - \underbrace{\mathbb{E}\big[\mathbb{E}[Y_k^\star \mid X, U] \cdot \mathrm{Cov}(D, Z \mid X, U) \mid X\big]}_{\text{III}} \\
- \underbrace{\mathbb{E}\big[\mathbb{E}[Y_k^\star \mid X, U] \cdot \mathbb{E}[D \mid X, U] \cdot \mathbb{E}[Z \mid X, U] \mid X\big]}_{\text{IV}},
\end{aligned}
\tag{15}
$$

Here again we use the consistency that $\mathbb{E}[DY_k \mid X] = \mathbb{E}[DY_k^\star \mid X]$. By the unconfoundedness $(Y^\star \perp D \mid X, U)$ from Assumption 3.1 and IV independence $(Z \perp U \mid X)$ from Assumption 3.2, we have $\mathbb{E}[Z \mid X, U] = \mathbb{E}[Z \mid X]$ and

$$
\mathbb{E}[Y_k^\star \mid X, U] \cdot \mathbb{E}[D \mid X, U] \cdot \mathbb{E}[Z \mid X, U] = \mathbb{E}[DY_k^\star \mid X, U] \cdot \mathbb{E}[Z \mid X].
$$

Therefore the II and IV terms in Equation (15) cancel out, which lead us to the result

$$
0 = \mathrm{Cov}(DY_k, Z \mid X) - \mathbb{E}\big[\mathbb{E}[Y_k^\star \mid X, U] \cdot \mathrm{Cov}(D, Z \mid X, U) \mid X\big].
\tag{16}
$$

Finally, by assuming Assumption 4.2 holds, that is,

$$
\mathrm{Cov}\big(\mathrm{Cov}(D, Z \mid X, U),\ \mathbb{E}[Y_k^\star \mid X, U] \mid X\big) = 0,
$$

we have

$$
\begin{aligned}
\mathbb{E}\big[\mathbb{E}[Y_k^\star \mid X, U] \cdot \mathrm{Cov}(D, Z \mid X, U) \mid X\big] &= \mathbb{E}\big[\mathbb{E}[Y_k^\star \mid X, U] \mid X\big] \cdot \mathbb{E}\big[\mathrm{Cov}(D, Z \mid X, U) \mid X\big] \\
&= \mathbb{E}[Y_k^\star \mid X] \cdot \mathbb{E}\big[\mathrm{Cov}(D, Z \mid X, U) \mid X\big].
\end{aligned}
$$

According to conditional covariance identity and IV independence $(Z \perp U \mid X)$, we have

$$
\begin{aligned}
\mathrm{Cov}(D, Z \mid X) &= \mathbb{E}\big[\mathrm{Cov}(D, Z \mid X, U) \mid X\big] + \mathbb{E}\big[\mathrm{Cov}(\mathbb{E}[D \mid X, U], \mathbb{E}[Z \mid X, U]) \mid X\big] \\
&= \mathbb{E}\big[\mathrm{Cov}(D, Z \mid X, U) \mid X\big] + \mathbb{E}\big[\mathrm{Cov}(\mathbb{E}[D \mid X, U], \mathbb{E}[Z \mid X]) \mid X\big] \\
&= \mathbb{E}\big[\mathrm{Cov}(D, Z \mid X, U) \mid X\big].
\end{aligned}
\tag{17}
$$

Combining (17) with (16) leads us to the desired result, that is,

$$
\eta_k^\star(X) = \mathbb{E}[Y_k^\star \mid X] = \frac{\mathrm{Cov}(DY_k, Z \mid X)}{\mathrm{Cov}(D, Z \mid X)}.
$$

- **Necessity**: Given the identification result $\mathbb{E}[Y_k^\star \mid X] = \frac{\mathrm{Cov}(DY_k, Z \mid X)}{\mathrm{Cov}(D, Z \mid X)}$ and the Equation (17), we have

$$
\begin{aligned}
\mathrm{Cov}(DY_k, Z \mid X) &= \mathbb{E}[Y_k^\star \mid X] \cdot \mathrm{Cov}(D, Z \mid X) \\
&= \mathbb{E}\big[\mathbb{E}[Y_k^\star \mid X, U] \mid X\big] \cdot \mathbb{E}\big[\mathrm{Cov}(D, Z \mid X, U) \mid X\big].
\end{aligned}
$$

On the other hand, we have the identity (16) established when every Assumptions 3.1 and 3.2 hold, that is,

$$\mathrm{Cov}(DY_k, Z \mid X) = \mathbb{E}\big[\mathbb{E}[Y_k^\star \mid X, U] \cdot \mathrm{Cov}(D, Z \mid X, U) \mid X\big].$$

Therefore, combining these results together, we have

$$\mathbb{E}\big[\mathbb{E}[Y_k^\star \mid X, U] \cdot \mathrm{Cov}(D, Z \mid X, U) \mid X\big] - \mathbb{E}\big[\mathbb{E}[Y_k^\star \mid X, U] \mid X\big] \cdot \mathbb{E}\big[\mathrm{Cov}(D, Z \mid X, U) \mid X\big] = 0,$$

which then recovers condition (4.2).

$\square$

*Proof of Theorem 4.4.* By the result in Lemma 4.1, we have

$$\begin{aligned}
\mathcal{R}(t, s^\star) &= \mathbb{E}\left[ \sum_{k=1}^{K} \eta_k^\star(X_i) \cdot \left( \mathbb{I}\{\arg\max_{p \in [K]} \eta_p^\star(X_i) = k\} - \mathbb{I}\{t(X_i) = k\} \right) \right] \\
&= \mathbb{E}\left[ \sum_{k=1}^{K} \left[ \max_{p \in [K]} \eta_p^\star(X_i) - \eta_k^\star(X_i) \right] \cdot \mathbb{I}\{t(X_i) = k\} \right] \\
&= \mathbb{E}\left[ \sum_{k=1}^{K} w_k(X_i) \cdot \mathbb{I}\{t(X_i) = k\} \right].
\end{aligned}$$

For each $k \in [K]$, we define $w_k(X) := \max_{p \in [K]}\{\eta_p^\star(X) - \eta_k^\star(X)\}$. According to Theorem 4.3, when NUCEM conditions in Assumption 4.2 are satisfied for each $k \in [K]$, the conditional probability $\eta_k^\star$ can be uniquely identified by Equation (5). Therefore, the weight function can be further identified as

$$w_k := \max_{p \in [K]} \left\{ \frac{\mathrm{Cov}(DY_p, Z \mid X)}{\mathrm{Cov}(D, Z \mid X)} - \frac{\mathrm{Cov}(DY_k, Z \mid X)}{\mathrm{Cov}(D, Z \mid X)} \right\}, \quad \forall\, k \in [K].$$

Now, consider a binary classification with the true label $Y^\star \in \{-1, 1\}$. Let $\eta^\star(x) := \mathbb{P}(Y^\star = 1 \mid X = x)$ denote the conditional probability of $Y^\star = 1$ given $X = x$. The Bayes optimal classifier is $s^\star(x) = \mathrm{sgn}[\eta^\star(x) - 1/2]$. As a consequence, the excess risk can be then written as

$$\begin{aligned}
\mathcal{R}(t, s^\star) &= \mathbb{P}\big(Y^\star \neq t(X)\big) - \mathbb{P}\big(Y^\star \neq s^\star(X)\big) \\
&= \mathbb{E}\Big[ \mathbb{I}\{Y^\star \neq t(X)\} - \mathbb{I}\{Y^\star \neq s^\star(X)\} \Big] \\
&= \mathbb{E}\Big[ \eta^\star(X) \cdot \big( \mathbb{I}\{1 \neq t(X)\} - \mathbb{I}\{1 \neq s^\star(X)\} \big) + (1 - \eta^\star(X)) \cdot \big( \mathbb{I}\{-1 \neq t(X)\} - \mathbb{I}\{-1 \neq s^\star(X)\} \big) \Big] \\
&= \mathbb{E}\Big[ (2\eta^\star(X) - 1) \cdot \Big( \frac{s^\star(X) - t(X)}{2} \Big) \Big] \\
&= \mathbb{E}\Big[ \big| \eta^\star(X) - 1/2 \big| \cdot \big| s^\star(X) - t(X) \big| \Big] \\
&= \mathbb{E}\Big[ \big| \eta^\star(X) - 1/2 \big| \cdot \mathbb{I}\{t(X) \neq \mathrm{sgn}[\eta^\star(X) - 1/2]\} \Big].
\end{aligned}$$

By defining the weight function $w^{\mathrm{exact}}(X) := \eta^\star(X) - 1/2$, we have

$$\mathcal{R}(t, s^\star) = \mathbb{E}\Big[ \big| w^{\mathrm{exact}}(X) \big| \cdot \mathbb{I}\{t(X) \neq \mathrm{sgn}[w^{\mathrm{exact}}(X)]\} \Big] := \mathcal{R}(t, w^{\mathrm{exact}}).$$

Applying the identification result from Theorem 4.3, we obtain

$$w^{\mathrm{exact}}(X) = \frac{\mathrm{Cov}(DY, Z \mid X)}{\mathrm{Cov}(D, Z \mid X)} - 1/2.$$

$\square$

## A.2. Sufficient Condition for Point Identification

**Proposition A.1** (A Sufficient Condition for Point Identification). *Suppose Assumptions 3.1 and 3.2 hold. For any $z_1, z_2 \in [m]$, define $\Delta_{jl}(X,U) := \mathbb{E}[D \mid X, U, Z = j] - \mathbb{E}[D \mid X, U, Z = l]$. Assumption 4.2 holds if, for any $j, l \in [J]$,*

$$\mathrm{Cov}(\Delta_{jl}(X,U), \mathbb{P}(Y^\star = k \mid X, U) \mid X) = 0, \ a.s. \tag{18}$$

Proposition A.1 outlines a sufficient condition for Assumption 4.2. This condition permits unobserved features $U$ to influence both the decision $D$ and the true outcome $Y^\star$, but constrains this influence to a specific form. Specifically, the condition is met when $\mathbb{E}[D \mid U, X, Z = j] = g_j(X) + q(U)$ almost surely, with functions $g_j$ and $q$ reflecting the impact of $U$ is additive and consistent across all decision-makers.

*Proof of Proposition A.1.* Notice that the conditional covariance $\mathrm{Cov}(D, Z \mid X, U)$ could be decomposed as below:

$$\begin{aligned}
&\mathrm{Cov}(D, Z \mid X, U) = \mathbb{E}[DZ \mid X, U] - \mathbb{E}[D \mid X, U] \cdot \mathbb{E}[Z \mid X, U] \\
&= \sum_{j=1}^{m} \mathbb{E}[DZ \mid X, U, Z = j] \cdot \mathbb{P}(Z = j \mid X, U) - \mathbb{E}[D \mid X, U] \cdot \sum_{j=1}^{m} j \cdot \mathbb{P}(Z = j \mid X, U) \\
&= \sum_{j=1}^{m} j \cdot \mathbb{E}[D \mid X, U, Z = j] \cdot \mathbb{P}(Z = j \mid X, U) - \sum_{l=1}^{m} \mathbb{E}[D \mid X, U, Z = l] \cdot \mathbb{P}(Z = l \mid X, U) \cdot \sum_{j=1}^{m} j \cdot \mathbb{P}(Z = j \mid X, U) \\
&= \sum_{j=1}^{m} \sum_{l=1}^{m} j \cdot \mathbb{P}(Z = l \mid X, U) \cdot \mathbb{P}(Z = k \mid X, U) \cdot \left[ \mathbb{E}[D \mid X, U, Z = j] - \mathbb{E}[D \mid X, U, Z = l] \right].
\end{aligned}$$

Define $\Delta_{j,l}(X,U) = \mathbb{E}[D \mid X, U, Z = j] - \mathbb{E}[D \mid X, U, Z = l]$. If for any $j, l \in [m]$, we have

$$\mathrm{Cov}(\Delta_{jl}(X,U), \ \mathbb{P}(Y^\star = k \mid X, U) \mid X) = 0 \ \text{ almost surely,}$$

Assumption 4.2 is then guaranteed.

$\square$

# B. Supplements for Partial Identification

## B.1. Tightness of IV Partial Bounds in Theorem 5.2

**Theorem B.1** (Tightness of IV Partial Bound). *Assume Assumption 3.1 is valid, and the conditional joint distribution of observable variables $(Y_k, D, Z) \mid X = x$ is fixed. Consider a new set of random variables $(\tilde{Y}_k^\star, \tilde{D}, \tilde{Z}, \tilde{X}, \tilde{U})$ where $\tilde{Y}_k^\star$ and $\tilde{D}$ are binary, and $\tilde{Z} = Z$. Define the observed label $\tilde{Y}_k$ as $\tilde{D} \cdot \tilde{Y}_k^\star + (1 - \tilde{D}) \cdot NaN$. If the expectation $\mathbb{E}[\tilde{Y}_k^\star \mid \tilde{X} = x]$ falls within the interval $[l_k(x), u_k(x)]$ for all $x \in \mathcal{X}$, then it is possible to establish a conditional joint distribution for $(\tilde{Y}_k^\star, \tilde{D}, \tilde{Z}, \tilde{U})$ given $\tilde{X} = x$ that satisfies:*

1. *The* IV independence *in Assumption 3.2 is met.*

2. *The conditional joint distribution of $(\tilde{Y}_k, \tilde{D}, \tilde{Z}) \mid \tilde{X} = x$ matches the initial distribution $(Y_k, D, Z) \mid X = x$.*

3. *The natural bounds in Assumption 5.1 are maintained.*

Theorem B.1 along with Theorem 5.2 establish a two-way correspondence between the conditional mean outcome $\mathbb{E}[\tilde{Y}_k^\star \mid X = x]$ and the IV partial bound $[l_k(x), u_k(x)]$. On one hand, any value of $\mathbb{E}[\tilde{Y}_k^\star \mid X = x]$ within this interval corresponds to a feasible conditional joint distribution of $(\tilde{Y}_k^\star, \tilde{D}, \tilde{Z}) \mid \tilde{X} = x$ that is consistent with the observed data $(Y_k, D, Z) \mid X = x$. On the other hand, any conditional joint distribution of $(\tilde{Y}_k^\star, \tilde{D}, \tilde{Z}) \mid \tilde{X} = x$ consistent with the observed data must yield a conditional mean outcome $\mathbb{E}[\tilde{Y}_k^\star \mid \tilde{X} = x]$ within the interval $[l_k(x), u_k(x)]$, as established in Theorem 5.2.

These results underscore the tightness of the IV partial bound in Theorem 5.2 and the necessity of the bounded label assumption in Assumption 5.1 for the partial identification of the conditional probability $\eta_k^\star$.

*Proof of Theorem B.1.* The proof is adapted from (Kédagni & Mourifie, 2017). Let $(\tilde{Y}_k^\star, \tilde{D}, \tilde{Z}, \tilde{X}, \tilde{U})$ be a sequence of random variables such that $\tilde{Z} = Z$ almost surely and $l_k(x) \leq \mathbb{E}[\tilde{Y}_k^\star \mid \tilde{X} = x] \leq u_k(x)$ for every $x \in \mathcal{X}$. Given that Assumption 3.1 held and fix $\tilde{X} = x$ for some $x \in \mathcal{X}$, we aim to prove that, for every possible value of $\mathbb{E}[\tilde{Y}_k^\star \mid \tilde{X} = x] \in [l_k(x), u_k(x)]$, there exists a conditional joint distribution of $(\tilde{Y}_k^\star, \tilde{D}, \tilde{Z}, \tilde{U})$ given $\tilde{X} = x$ such that: (1) IV independence in Assumption 3.2 holds, that is, $\tilde{Z} \perp \tilde{Y}_k^\star \mid \tilde{X} = x$ for every $x \in \mathcal{X}$; (2) the new observable variables $(\tilde{Y}_k, \tilde{D}, \tilde{Z}) \mid \tilde{X} = x$ has the same conditional joint distribution as the fixed observable variables $(Y_k, D, Z) \mid X = x$; (3) Assumption 5.1 holds.

Without loss of generality, we first consider the case when $\mathbb{E}[\tilde{Y}_k^\star \mid \tilde{X} = x] = l_k(x)$. As a consequent, we let

$$
\begin{aligned}
\mathbb{P}(\tilde{Y}_k^\star = 1 \mid \tilde{X} = x) &:= l_k(x) = \max_{z \in [m]} \left\{ \mathbb{E}[DY_k \mid X = x, Z = z] + a(x) \cdot \left(1 - \mathbb{E}[D \mid X = x, Z = z]\right) \right\} \\
\mathbb{P}(\tilde{Y}_k^\star = 0 \mid X = x) &:= 1 - l_k(x),
\end{aligned}
\tag{19}
$$

where the last expression follows by sum-to-one constraint of conditional probability. Note that we does not impose any observable restrictions on the conditional joint distribution of $(\tilde{Y}_k, \tilde{D})$ given $\tilde{X} = x$ and $\tilde{Z} = z$, we define the following: for all $z \in \mathcal{Z}$,

$$
\begin{aligned}
\mathbb{P}(\tilde{Y}_k^\star = 1, \tilde{D} = 1 \mid \tilde{X} = x, \tilde{Z} = z) &:= \mathbb{E}[DY_k \mid X = x, Z = z] \\
\mathbb{P}(\tilde{Y}_k^\star = 1, \tilde{D} = 0 \mid \tilde{X} = x, \tilde{Z} = z) &:= l_k(x) - \mathbb{E}[DY_k \mid X = x, Z = z] \\
\mathbb{P}(\tilde{Y}_k^\star = 0, \tilde{D} = 1 \mid \tilde{X} = x, \tilde{Z} = z) &:= \mathbb{E}[D \mid X = x, Z = z] - \mathbb{E}[DY_k \mid X = x, Z = z] \\
\mathbb{P}(\tilde{Y}_k^\star = 0, \tilde{D} = 0 \mid \tilde{X} = x, \tilde{Z} = z) &:= \left(1 - l_k(x)\right) - \left(\mathbb{E}[D \mid X = x, Z = z] - \mathbb{E}[DY_k \mid X = x, Z = z]\right).
\end{aligned}
\tag{20}
$$

Notice that all of the conditional probabilities are well-established as they are all between $0$ and $1$, and the sum-to-one constraint is established as follow:

$$
\sum_{i \in \{0,1\}} \sum_{j \in \{0,1\}} \mathbb{P}(\tilde{Y}_k^\star = i, \tilde{D} = j \mid X, Z = z) = 1, \ \forall z \in [m].
$$

We now prove the three statements mentioned above.

1. IV Independence: To check whether we have $\tilde{Z} \perp \tilde{Y}_k^\star \mid \tilde{X} = x$, notice that for any $z \in \mathcal{Z}$,

$$\mathbb{P}(\tilde{Y}_k^\star = 1 \mid \tilde{X} = x, \tilde{Z} = z) = \mathbb{P}(\tilde{Y}_k^\star = 1, \tilde{D} = 1 \mid \tilde{X} =, \tilde{Z} = z) + \mathbb{P}(\tilde{Y}_k^\star = 1, \tilde{D} = 0 \mid \tilde{X} = x, \tilde{Z} = z) = l_k(x)$$
$$\mathbb{P}(\tilde{Y}_k^\star = 0 \mid \tilde{X} = x, \tilde{Z} = z) = \mathbb{P}(\tilde{Y}_k^\star = 0, \tilde{D} = 1 \mid \tilde{X} = x, \tilde{Z} = z) + \mathbb{P}(\tilde{Y}_k^\star = 0, \tilde{D} = 0 \mid \tilde{X} = x, \tilde{Z} = z) = 1 - l_k(x).$$

We can see that the conditional distribution of $\tilde{Y}^\star$ given $\tilde{X} = x$ and $\tilde{Z} = z$ does not depends on $z$ for any $z \in [m]$, and therefore we are able to claim that $\tilde{Z}$ is conditional independent with $\tilde{Y}_k^\star$ given $\tilde{X} = x$.

2. $(\tilde{Y}_k, \tilde{D}, \tilde{Z}) \mid \tilde{X} = x$ has same conditional joint distribution as $(Y_k, D, Z) \mid X = x$: Recall that the observed label is defined as $\tilde{Y}_k = \tilde{D}\tilde{Y}_k^\star + (1 - \tilde{D}) \cdot \text{NaN}$. By Equation (20), we have for any $z \in \mathcal{Z}$,

$$
\begin{aligned}
\mathbb{P}(\tilde{Y}_k = 1, \tilde{D} = 1 \mid \tilde{X} = x, \tilde{Z} = z) &= \mathbb{P}(\tilde{Y}_k^\star = 1, \tilde{D} = 1 \mid \tilde{X} = x, \tilde{Z} = z) \\
&= \mathbb{E}[DY_k \mid X = x, Z = z] \\
&= \mathbb{P}(Y_k = 1, D = 1 \mid X = x, Z = z) \\
\mathbb{P}(\tilde{Y}_k = 0, \tilde{D} = 1 \mid \tilde{X} = x, \tilde{Z} = z) &= \mathbb{P}(\tilde{Y}_k^\star = 0, \tilde{D} = 1 \mid \tilde{X} = x, \tilde{Z} = z) \\
&= \mathbb{E}[D \mid X = x, Z = z] - \mathbb{E}[DY_k \mid X = x, Z = z] \\
&= \mathbb{P}(Y_k = 0, D = 1 \mid X = x, Z = z) \\
\mathbb{P}(\tilde{Y}_k = \text{NaN}, \tilde{D} = 0 \mid \tilde{X} = x, \tilde{Z} = z) &= \mathbb{P}(\tilde{D} = 0 \mid \tilde{X} = x, \tilde{Z} = z) \\
&= \sum_{k \in \{0,1\}} \mathbb{P}(\tilde{Y}_k^\star = k, \tilde{D} = 0 \mid \tilde{X} = x, \tilde{Z} = z) \\
&= 1 - \mathbb{E}[D \mid X = x, Z = z] = \mathbb{P}(D = 0 \mid X = x, Z = z) \\
&= \mathbb{P}(Y_k = \text{NA}, D = 0 \mid X = x, Z = z).
\end{aligned}
\tag{21}
$$

As a result, we have shown that the observed variables $(\tilde{Y}_k, \tilde{D}, \tilde{Z}) \mid \tilde{X} = x$ has exactly the same conditional joint distribution with $(Y_k, D, Z) \mid X = x$.

3. Finally, we show that when $\mathbb{E}[\tilde{Y}_k^\star \mid \tilde{X} = x] = l_k(x)$, there exists a joint distribution on $(\tilde{Y}_k^\star, \tilde{X}, \tilde{U})$ such that $\mathbb{E}[\tilde{Y}_k^\star \mid \tilde{U}, \tilde{X} = x] = a(x)$. Without the loss of generality, we assume

$$z_0 := \arg\max_{z \in [m]} \left\{ \mathbb{E}[DY_k \mid X = x, Z = z] + a_k(x) \cdot \mathbb{E}[(1 - D) \mid X = x, Z = z] \right\}$$

and then we have

$$\mathbb{E}[\tilde{Y}_k^\star \mid \tilde{X} = x] = l_k(x) = \mathbb{E}[DY_k \mid X = x, Z = z_0] + a_k(x) \cdot \mathbb{E}[(1 - D) \mid X = x, Z = z_0]. \tag{22}$$

Meanwhile, by the statement of IV independence that $\tilde{Z} \perp \tilde{Y}_k^\star \mid \tilde{X} = x$, we have for $Z = z_0$,

$$
\begin{aligned}
\mathbb{E}[\tilde{Y}_k^\star \mid \tilde{X} = x] &= \mathbb{E}[\tilde{Y}_k^\star \mid \tilde{X} = x, \tilde{Z} = z_0] \\
&= \mathbb{E}[\tilde{D}\tilde{Y}_k^\star \mid \tilde{X} = x, \tilde{Z} = z_0] + \mathbb{E}[(1 - \tilde{D})\tilde{Y}_k^\star \mid \tilde{X} = x, \tilde{Z} = z_0] \\
&= \mathbb{E}[\tilde{D}\tilde{Y}_k \mid \tilde{X} = x, \tilde{Z} = z_0] + \mathbb{E}[\mathbb{E}[(1 - \tilde{D})\tilde{Y}_k^\star \mid \tilde{U}, \tilde{X} = x, \tilde{Z} = z_0] \mid \tilde{X} = x, \tilde{Z} = z_0] \\
&= \mathbb{E}[\tilde{D}\tilde{Y}_k \mid \tilde{X} = x, \tilde{Z} = z_0] + \mathbb{E}[\mathbb{E}[\tilde{Y}_k^\star \mid \tilde{U}, \tilde{X} = x] \cdot \mathbb{E}[(1 - \tilde{D}) \mid \tilde{U}, \tilde{X} = x, \tilde{Z} = z_0] \mid \tilde{X} = x, \tilde{Z} = z_0].
\end{aligned}
$$

The first line and second line come from the IV independence and linearity of conditional expectation. The third line follows by the consistency of observed outcome as well as the iterated law of expectation. The last line follows by Assumption 3.1 ($\tilde{D} \perp \tilde{Y}_k^\star \mid \tilde{X}, \tilde{U}$) and the IV independence.

From Equation (21), we have $\mathbb{E}[\tilde{D}\tilde{Y}_k \mid \tilde{X} = x, \tilde{Z} = z_0] = \mathbb{E}[DY_k \mid X = x, Z = z_0)$, and therefore

$$
\begin{aligned}
\mathbb{E}[\tilde{Y}_k^\star \mid \tilde{X} = x] = {}& \mathbb{E}[DY_k \mid X = x, Z = z_0] \\
&+ \mathbb{E}[\mathbb{E}[\tilde{Y}_k^\star \mid \tilde{U}, \tilde{X} = x] \cdot \mathbb{E}[(1 - \tilde{D}) \mid \tilde{U}, \tilde{X} = x, \tilde{Z} = z_0] \mid \tilde{X} = x, \tilde{Z} = z_0].
\end{aligned}
\tag{23}
$$

Now, suppose that there is not any conditional joint distribution of $(\tilde{Y}_k^\star, \tilde{D}, \tilde{U})$ given $\tilde{X} = x$ such that $\mathbb{E}[\tilde{Y}_k^\star \mid \tilde{U}, \tilde{X} = x] = a_k(x)$ almost surely. Alternatively, we assume there is a conditional joint distribution for $(\tilde{Y}_k^\star, \tilde{D}, \tilde{U}) \mid \tilde{X} = x$ such that $\mathbb{E}[\tilde{Y}_k^\star \mid \tilde{U}, \tilde{X} = x] = a_k(x) + \varepsilon(x, \tilde{U})$ almost surely. Substitute this condition into Equation (23), we have

$$
\begin{aligned}
\mathbb{E}[\tilde{Y}_k^\star \mid \tilde{X} = x] &= \mathbb{E}[DY_k \mid X = x, Z = z_0] + a_k(x) \cdot \mathbb{E}\big[(1 - \tilde{D}) \mid \tilde{X} = x, \tilde{Z} = z_0\big] \\
&\quad + \mathbb{E}\Big[\varepsilon(\tilde{X}, \tilde{U}) \cdot \mathbb{E}[(1 - \tilde{D}) \mid \tilde{U}, \tilde{X} = x, \tilde{Z} = z_0] \mid \tilde{X} = x, \tilde{Z} = z_0\Big] \\
&= l_k(x) + \mathbb{E}\Big[\varepsilon(\tilde{X}, \tilde{U}) \cdot \mathbb{E}[(1 - \tilde{D}) \mid \tilde{U}, \tilde{X} = x, \tilde{Z} = z_0] \mid \tilde{X} = x, \tilde{Z} = z_0\Big].
\end{aligned}
\tag{24}
$$

In the first equality, we use the iterated law of expectation and fact that $a_k(x)$ is independent with unobserved variable $\tilde{U}$. The second equality follows from Equation (21) that $\mathbb{P}(\tilde{D} = 0 \mid \tilde{X} = x, \tilde{Z} = z_0) = \mathbb{P}(D = 0 \mid X = x, Z = z)$ for any fixed $x \in \mathcal{X}$ and $z \in [m]$ along with the definition of $l_k(x)$.

As a consequence, if

$$
\mathbb{E}\big[\varepsilon(\tilde{X}, \tilde{U}) \cdot \mathbb{E}[(1 - \tilde{D}) \mid \tilde{U}, \tilde{X} = x, \tilde{Z} = z_0] \mid \tilde{X} = x, \tilde{Z} = z_0\big] \neq 0,
$$

then Equation (24) contradicts with Equation (22), implying that there is indeed a conditional joint distribution for $(\tilde{Y}^\star, \tilde{D}, \tilde{U}) \mid \tilde{X} = x$ such that $\mathbb{E}[\tilde{Y}^\star \mid \tilde{U}, \tilde{X} = x] = a_k(x)$. On the other hand, if

$$
\mathbb{E}\big[\varepsilon(\tilde{X}, \tilde{U}) \cdot \mathbb{E}[(1 - \tilde{D}) \mid \tilde{U}, \tilde{X} = x, \tilde{Z} = z_0] \mid \tilde{X} = x, \tilde{Z} = z_0\big] = 0,
\tag{25}
$$

then the conditional joint distribution for $(\tilde{Y}_k, \tilde{D}, \tilde{U}) \mid \tilde{X} = x$ induced by the Equation (25) is exactly the "right" distribution for the establishment of $\mathbb{E}[\tilde{Y}_k^\star \mid \tilde{U}, \tilde{X} = x] = a_k(x)$.

Therefore, we have shown that when $\mathbb{E}[\tilde{Y}_k^\star \mid \tilde{X} = x] = l_k(x)$, there is always a conditional joint distribution for $(\tilde{Y}_k^\star, \tilde{D}, \tilde{U}) \mid \tilde{X} = x$ such that $\mathbb{E}[\tilde{Y}_k^\star \mid \tilde{U}, \tilde{X} = x] = a_k(x)$ almost surely.

To wrap up, we can see that the establishment of statements 1 and 2 do not rely on the condition $\mathbb{P}(\tilde{Y}^\star = 1 \mid \tilde{X} = x) = l_k(x)$ in Equation (19). In fact, whenever we choose $g_k(x) \in [l_k(x), u_k(x)]$ for a fixed $x \in \mathcal{X}$, we can simply choose $\mathbb{P}(\tilde{Y}_k^\star = 1 \mid \tilde{X} = x) = g_k(x)$. As a result, the IV independence $\tilde{Z} \perp \tilde{Y}_k^\star \mid \tilde{X} = x$ always holds, and the new observable variables $(\tilde{Y}_k, \tilde{D}, \tilde{Z}) \mid \tilde{X} = x$ still has the same conditional distribution as $(Y_k, D, Z) \mid X = x$, which is fixed as a prior.

However, the value of $\mathbb{P}(\tilde{Y}_k^\star = 1 \mid \tilde{X} = x)$ condition does affect the value of $\mathbb{E}[\tilde{Y}_k^\star \mid \tilde{U}, \tilde{X} = x]$. If we let $\mathbb{E}[\tilde{Y}_k^\star \mid \tilde{X} = x] = u_k(x)$, we can show in a similar way that $\mathbb{E}[\tilde{Y}_k^\star \mid \tilde{U}, \tilde{X} = x] = b_k(x)$ almost surely. In fact, following the proof of statement 3, one can prove that, for any $g_k(x) \in [l_k(x), u_k(x)]$ such that $\mathbb{E}[\tilde{Y}_k^\star \mid \tilde{X} = x] = g_k(x)$, there is always a conditional joint distribution for $(\tilde{Y}_k, \tilde{D}, \tilde{U}) \mid \tilde{X} = x$ such that $\mathbb{E}[\tilde{Y}_k^\star \mid \tilde{U}, \tilde{X} = x] = c_k(x)$ almost surely and $c_k(x) \in [a_k(x), b_k(x)]$.

Therefore, we have shown that under Assumption 3.1, whenever we have $\mathbb{E}[\tilde{Y}_k^\star \mid \tilde{X} = x] \in [l_k(x), u_k(x)]$, there is always a conditional joint distribution for $(\tilde{Y}_k^\star, \tilde{D}, \tilde{Z}, \tilde{U}) \mid \tilde{X} = x$ such that the three statements in Theorem B.1 hold. To conclude, we claim that the IV partial bound $[l_k(x), u_k(x)]$ introduced in Theorem 5.2 is **sharp** for $\mathbb{E}[Y_k^\star \mid X = x]$ given the establishment of Assumption 3.1, the IV independence in Assumption 3.2 and Assumption 5.1.

$\square$

## B.2. Balke and Pearl's Bound

(Balke & Pearl, 1994) provides partial identification bounds for the average treatment effect of a binary treatment with a binary instrumental variable. In this section, we adapt their bound to our setting with partially observed labels and a binary IV (i.e., the assignment to one of two decision-makers). Under Assumption 3.2, we have following decomposition of joint probability distribution of $(Y, D, Z, U)$

$$
\mathbb{P}(Y, D, Z, U) = \mathbb{P}(Y \mid D, U) \cdot \mathbb{P}(D \mid Z, U) \cdot \mathbb{P}(Z) \cdot \mathbb{P}(U).
\tag{26}
$$

Here we omit the observed covariates $X$ for simplicity, or alternatively, all distributions can be considered as implicitly

conditioning on $X$. Now we define three response functions which characterize the values of $Z$, $D(0)$, $D(1)$, and $Y^\star$:

$$r_Z = \begin{cases} 0 & \text{if } Z = 0 \\ 1 & \text{if } Z = 1 \end{cases}, \quad r_D = \begin{cases} 0 & \text{if } D(0) = 0 \text{ and } D(1) = 0 \\ 1 & \text{if } D(0) = 0 \text{ and } D(1) = 1 \\ 2 & \text{if } D(0) = 1 \text{ and } D(1) = 0 \\ 3 & \text{if } D(0) = 1 \text{ and } D(1) = 1 \end{cases}, \quad r_Y = \begin{cases} 0 & \text{if } Y^\star = 0 \\ 1 & \text{if } Y^\star = 1 \end{cases}.$$

Next, we specify the joint distribution of unobservable variables $r_D$ and $r_Y$ as follows:

$$q_{kj} = \mathbb{P}(r_D = k, r_Y = j) \quad \forall\, k \in \{0, 1, 2, 3\},\ j \in \{0, 1\},$$

which satisfies the constraint $\sum_{k=0}^{3}(q_{k0} + q_{k1}) = 1$. Then the target mean parameter of the true outcome can be written as a linear combinations of the $q$'s. Moreover, we note that the observable distribution $\mathbb{P}(Y, D \mid Z)$ is fully specified by the following six variables

$$
\begin{aligned}
p_{na,0} &= \mathbb{P}(D = 0 \mid Z = 0) & p_{na,1} &= \mathbb{P}(D = 0 \mid Z = 1) \\
p_{01,0} &= \mathbb{P}(Y = 0, D = 1 \mid Z = 0) & p_{01,1} &= \mathbb{P}(Y = 0, D = 1 \mid Z = 1) \\
p_{11,0} &= \mathbb{P}(Y = 1, D = 1 \mid Z = 0) & p_{11,1} &= \mathbb{P}(Y = 1, D = 1 \mid Z = 1),
\end{aligned}
$$

with constraints $p_{11,0} + p_{01,0} + p_{na,0} = 1$ and $p_{11,1} + p_{01,1} + p_{na,1} = 1$. We also have the following relation between $p$'s and $q$'s:

$$
\begin{aligned}
p_{na,0} &= q_{00} + q_{01} + q_{10} + q_{11} & p_{na,1} &= q_{00} + q_{01} + q_{20} + q_{21} \\
p_{01,0} &= q_{20} + q_{30} & p_{01,1} &= q_{10} + q_{30} \\
p_{11,0} &= q_{21} + q_{31} & p_{11,1} &= q_{11} + q_{31}.
\end{aligned}
$$

Therefore, we have $p = Pq$ where $p = (p_{na,0}, \ldots, p_{11,1})$, $q = (q_{00}, \ldots, q_{31})$, and

$$P = \begin{bmatrix} 1 & 1 & 0 & 0 & 1 & 1 & 0 & 0 \\ 0 & 0 & 1 & 1 & 0 & 0 & 0 & 0 \\ 0 & 0 & 0 & 0 & 0 & 0 & 1 & 1 \\ 1 & 0 & 1 & 0 & 1 & 0 & 1 & 0 \\ 0 & 1 & 0 & 1 & 0 & 0 & 0 & 0 \\ 0 & 0 & 0 & 0 & 0 & 1 & 0 & 1 \end{bmatrix}.$$

Then the lower bound on $\mathbb{P}(Y^\star = 1)$ can be written as the optimal value of the following linear programming problem

$$
\begin{aligned}
\min \quad & q_{01} + q_{11} + q_{21} + q_{31} \\
\text{subject to} \quad & \sum_{k=0}^{3}\sum_{j=0}^{1} q_{kj} = 1 \\
& Pq = p \\
& q_{kj} \geq 0 \quad k \in \{0, 1, 2, 3\},\ j \in \{0, 1\}.
\end{aligned}
\tag{27}
$$

Similarly, the upper bound on $\mathbb{P}(Y^\star = 1)$ can be written as the optimal value of the following optimization problem:

$$
\begin{aligned}
\max \quad & q_{01} + q_{11} + q_{21} + q_{31} \\
\text{subject to} \quad & \sum_{k=0}^{3}\sum_{j=0}^{1} q_{kj} = 1 \\
& Pq = p \\
& q_{kj} \geq 0 \quad k \in \{0, 1, 2, 3\},\ j \in \{0, 1\}.
\end{aligned}
\tag{28}
$$

In fact, by simply comparing the variables in the objective function $\mathbb{P}(Y^\star = 1) = q_{01} + q_{11} + q_{21} + q_{31}$ and those in constraints, one could find that

$$p_{11,0} = q_{21} + q_{31} \leq \mathbb{P}(Y^\star = 1)$$
$$p_{11,1} = q_{11} + q_{31} \leq \mathbb{P}(Y^\star = 1)$$
$$p_{11,0} + p_{na,0} = q_{00} + q_{10} + q_{01} + q_{11} + q_{21} + q_{31} \geq \mathbb{P}(Y^\star = 1)$$
$$p_{11,1} + p_{na,1} = q_{00} + q_{20} + q_{01} + q_{11} + q_{21} + q_{31} \geq \mathbb{P}(Y^\star = 1).$$

If we let

$$L = \max\{p_{11,0}, p_{11,1}\} = \max_{z \in \{0,1\}} \{\mathbb{P}(Y = 1, D = 1 \mid Z = z)\}$$

$$U = \min\{p_{11,0} + p_{na,0}, p_{11,1} + p_{na,1}\} = \min_{z \in \{0,1\}} \{\mathbb{P}(Y = 1, D = 1 \mid Z = z) + \mathbb{P}(D = 0 \mid Z = z)\}. \tag{29}$$

We then have the following partial bounds of $\mathbb{P}(Y^\star = 1)$.

$$L \ \leq \ \mathbb{P}(Y^\star = 1) \ \leq \ U.$$

According to (Balke & Pearl, 1994), the bounds above are tight for $\mathbb{P}(Y^\star = 1)$. We note that if we condition on $X$ in these bounds, then the corresponding bound on $\mathbb{P}(Y^\star = 1 \mid X)$ coincide with the bounds in Theorem 5.2 specialized to a binary instrument.

## B.3. Proofs in Section 5

*Proof of Theorem 5.2.* We first prove the construction of IV partial bound, and then show that the IV partial bound $[l_k(x), u_k(x)]$ is tighter than the natural bound $[a_k(x), b_k(x)]$ for all $k \in [K]$ when $x$ is fixed.

1. Under Assumptions 3.1 and 3.2, Equation (14) gives that

$$\mathbb{P}(Y^\star = k \mid X = x) = \mathbb{E}[Y_k^\star \mid X = x]$$
$$= \mathbb{E}[DY_k \mid X = x, Z = z] + \mathbb{E}\big[\mathbb{E}[Y_k^\star \mid U, X = x] \cdot \mathbb{E}[(1 - D) \mid U, X = x, Z = z] \mid X = x, Z = z\big].$$

As we have assumed that $\mathbb{E}[Y_k^\star \mid U, X = x] \in [a_k(x), b_k(x)]$ almost surely in Assumption 5.1, we then have

$$\mathbb{E}[Y_k^\star \mid X = x] \geq \mathbb{E}[DY_k \mid X = x, Z = z] + a_k(x) \cdot \mathbb{E}\big[\mathbb{E}[(1 - D) \mid U, X = x, Z = z] \mid X = x, Z = z\big]$$
$$= \mathbb{E}[DY_k \mid X = x, Z = z] + a_k(x) \cdot \mathbb{E}[(1 - D) \mid X = x, Z = z]$$

and

$$\mathbb{E}[Y_k^\star \mid X = x] \leq \mathbb{E}[DY_k \mid X = x, Z = z] + b_k(x) \cdot \mathbb{E}\big[\mathbb{E}[(1 - D) \mid U, X = x, Z = z] \mid X = x, Z = z\big]$$
$$= \mathbb{E}[DY_k \mid X = x, Z = z] + b_k(x) \cdot \mathbb{E}[(1 - D) \mid X = x, Z = z]$$

for every $z \in [m]$. Optimizing the lower and upper bounds of $\mathbb{E}[Y_k^\star \mid X = x]$ over $z \in \mathcal{Z}$ yields the desired results:

$$\max_{z \in \mathcal{Z}} \left\{ \mathbb{E}[DY_k \mid X = x, Z = z] + a_k(x) \cdot \big(1 - \mathbb{E}[D \mid X = x, Z = z]\big) \right\}$$
$$\leq \mathbb{E}[Y_k^\star \mid X = x] \leq$$
$$\min_{z \in \mathcal{Z}} \left\{ \mathbb{E}[DY_k \mid X = x, Z = z] + b_k(x) \cdot \big(1 - \mathbb{E}[D \mid X = x, Z = z]\big) \right\}.$$

2. By taking the conditional expectation on both sides of the inequality, a direct consequence of Assumption 5.1 induces a natural bound for $\mathbb{E}[Y_k^\star \mid X = x]$, that is,

$$a_k(x) \leq \mathbb{E}[Y_k^\star \mid X = x] \leq b_k(x).$$

We now show that the IV partial bound $[l_k(x), u_k(x)]$ is tighter than the natural bound $[a_k(x), b_k(x)]$ for any $x \in \mathcal{X}$. Without loss of generality, let

$$z_0 := \arg\max_{z \in [m]} \left\{ \mathbb{E}[DY_k \mid X = x, Z = z] + a_k(x) \cdot \mathbb{E}[(1 - D) \mid X = x, Z = z] \right\}$$

and then we have

$$
\begin{aligned}
l_k(x) &= \mathbb{E}[DY_k \mid X = x, Z = z_0] + a_k(x) \cdot \mathbb{E}[(1 - D) \mid X = x, Z = z_0] \\
&= \underbrace{a_k(x) - a_k(x) \cdot \mathbb{P}(D = 1 \mid X = x, Z = z_0)}_{(\cdots)} + \mathbb{E}[DY_k^\star \mid X = x, Z = z_0] \\
&= (\cdots) + \mathbb{E}\Big[ \mathbb{E}[DY_k^\star \mid U, X = x, Z = z_0] \mid X = x, Z = z_0 \Big] \\
&= (\cdots) + \mathbb{E}\Big[ \mathbb{E}[D \mid U, X = x, Z = z_0] \cdot \mathbb{E}[Y_k^\star \mid U, X = x, Z = z_0] \mid X = x, Z = z_0 \Big] \\
&= (\cdots) + \mathbb{E}\Big[ \mathbb{P}(D = 1 \mid U, X = x, Z = z_0) \cdot \mathbb{E}[Y_k^\star \mid U, X = x] \mid X = x, Z = z_0 \Big] \\
&= a(x) + \mathbb{E}\Big[ \mathbb{P}(D = 1 \mid U, X = x, Z = z_0) \cdot \big( \mathbb{E}[Y_k^\star \mid U, X = x] - a_k(x) \big) \mid X = x, Z = z_0 \Big].
\end{aligned}
\tag{30}
$$

The first line comes from the definition, while the second line holds by the consistency between the observed label and the true label ($DY_k = DY_k^\star$) and the linearity of conditional expectation. The third line holds by iterated law of expectation, and the fourth line holds by Assumption 3.1 ($D \perp Y_k^\star \mid X, U$). The fifth line follows by Assumption 3.2 ($Z \perp Y_k^\star \mid X$) and the fact that $\mathbb{E}[D \mid U, X = x, Z = z] = \mathbb{P}(D = 1 \mid U, X = x, Z = z)$. The last line again follows by the linearity and the iterated law of expectation.

Recall that Assumption 5.1 states that $E[Y_k^\star \mid U, X = x] \geq a_k(x)$ almost surely, along with Equation (30) and the fact that $\mathbb{P}(D = 1 \mid U, X = x, Z = z_0) \geq 0$, we have

$$l_k(x) \geq a_k(x) \ \forall\, x \in \mathcal{X}.$$

Similarly, we can show that $u_k(x) \leq b_k(x)$ for every fixed $x \in \mathcal{X}$. We therefore complete the proof.

$\square$

*Proof of Theorem 5.3.* To simplify the notation, for any classifier $t : \mathcal{X} \mapsto [K]$ and vector function $\eta : \mathcal{X} \mapsto [0, 1]^K$, define the function

$$A(t, \eta; x) = \sum_{k=1}^{K} \eta_k(x) \cdot \Big( \mathbb{I}\{\arg\max_{k \in [K]} \eta_k(x) = k\} - \mathbb{I}\{t(x) = k\} \Big) \geq 0.$$

The worst-case risk function can then be written as

$$\overline{\mathcal{R}}(t) = \mathbb{E}\Big[ \max_{\eta \in S} A(t, \eta; X) \Big]$$

with $S = \{\eta \in [0, 1]^K : \|\eta\|_1 = 1, \eta_k \in [l_k, u_k], k = 1, \ldots, K\}$. The following proof will rely on computing the maximization of $A(t, \eta; x)$ over all possible values of $t(x) \in [K]$ for any fixed $x \in \mathcal{X}$.

- **Step 1**. We start with the case when $t(x) = k_0$ for some $k_0 \in [K]$, we have

$$
\begin{aligned}
\max_{\eta \in S} \Big\{ A(t, \eta; x) \Big\} \cdot \mathbb{I}\{t(x) = k_0\} &= \max_{\eta \in S} \Big\{ A(t, \eta; x) \cdot \mathbb{I}\{t(x) = k_0\} \Big\} \\
&= \max_{\eta \in S} \Big\{ \Big( \sum_{k \in [K]} \eta_k(x) \cdot \mathbb{I}\{\arg\max_{p \in [K]}\{\eta_p(x)\} = k\} - \sum_{k \in [K]} \eta_k(x) \cdot \mathbb{I}\{t(x) = k\} \Big) \cdot \mathbb{I}\{t(x) = k_0\} \Big\} \\
&= \max_{\eta \in S} \Big\{ \Big( \sum_{k \in [K]} \eta_k(x) \cdot \mathbb{I}\{\arg\max_{p \in [K]}\{\eta_p(x)\} = k\} - \eta_{k_0}(X) \Big) \cdot \mathbb{I}\{t(x) = k_0\} \Big\}
\end{aligned}
$$

Given any vector $\eta(x) \in S$, we denote $p_0 := \arg\max_{p \in [K]} \eta_p(x)$ as the label with maximal conditional probability. We have

$$
\begin{aligned}
\max_{\eta \in S} \left\{ A(t, \eta; x) \right\} \cdot \mathbb{I}\{t(x) = k_0\} &= \max_{\eta \in S} \left\{ \left( \eta_{p_0}(x) - \eta_{k_0}(x) \right) \right\} \cdot \mathbb{I}\{t(x) = k_0\} \\
&= \max_{p_0 \neq k_0, \eta \in S} \left\{ \left( \eta_{p_0}(x) - \eta_{k_0}(x) \right)^+ \right\} \cdot \mathbb{I}\{t(x) = k_0\} \geq 0,
\end{aligned}
\tag{31}
$$

where $(z)^+ := \max(z, 0)$ for any real-valued $z$. In the second equality of Equation (31), we exclude the case when $p_0 = k_0$ from the maximization, as this situation only occurs when the lower bound of $\eta_{k_0}(x)$ exceeds the upper bound of $\eta_p(x)$ for all $p \neq k_0$. Consequently, we have $p_0 = \arg\max_{p \in [K]} \eta_p(x) = k_0$, which implies that

$$
\max_{p_0 = k_0, \eta \in S} \{ A(t, \eta; x) \} \cdot \mathbb{I}\{t(x) = k_0\} = \max_{p_0 = k_0, \eta \in S} \{ (\eta_{k_0}(x) - \eta_{k_0}(x)) \} \cdot \mathbb{I}\{t(x) = k_0\} = 0.
$$

Therefore, we expect the maximization of $A(t, \eta; x) \cdot \mathbb{I}\{t(x) = k_0\}$ over $\eta \in S$ to be non-negative. We exclude the case $p_0 = k_0$ from the maximization to facilitate further analysis.

- **Step 2**. As shown in Equation (31), for any fixed $x \in \mathcal{X}$ and $p_0 \neq k_0$, the term $(\eta_{p_0}(x) - \eta_{k_0}(x))^+$ is a non-decreasing ($\uparrow$) function of $\eta_{p_0}(x)$ and a non-increasing ($\downarrow$) function of $\eta_{k_0}(x)$. Therefore, one may expect that the maximization of $(\eta_{p_0}(x) - \eta_{k_0}(x))^+$ over $\eta(x) \in S(x)$ is realized when $\eta_{p_0}(x)$ and $\eta_{k_0}(x)$ achieving their upper-bound and lower-bound respectively. However, due to the constraint $\|\eta(x)\|_1 = 1$ for any $x \in \mathcal{X}$, the orginal upper-bound $u_{p_0}(x)$ for $\eta_{p_0}(x)$ may not be reached, and similar concern arises when treating $\eta_{k_0}(x)$. To fix these issues, we define a "realizable" upper-bound for $\eta_{p_0}(x)$ as

$$
\tilde{u}_{p_0}(x) := \left[ 1 - \sum_{j \neq p_0} l_j(x) \right] \wedge u_{p_0}(x),
\tag{32}
$$

which is the maximal value that $\eta_{p_0}(x)$ can realize under the constraint $\sum_{k \in [K]} \eta_k(x) = 1$. Meanwhile, we define a "realizable" lower-bound for $\eta_{k_0}(x)$ as

$$
\begin{aligned}
\tilde{l}_{k_0}(x) &:= \left[ 1 - \tilde{u}_{p_0}(x) - \sum_{j \neq p_0, k_0} u_j(x) \right] \vee l_{k_0}(x) \\
&= \left\{ \left[ 1 - \left( 1 - \sum_{j \neq p_0} l_j(x) \right) - \sum_{j \neq p_0, k_0} u_j(x) \right] \vee \left[ 1 - u_{p_0}(x) - \sum_{j \neq p_0, k_0} u_j(x) \right] \right\} \vee l_{k_0}(x) \\
&= \left\{ \left[ l_{k_0}(x) + \sum_{j \neq p_0, k_0} (l_j(x) - u_j(x)) \right] \vee \left[ 1 - \sum_{j \neq k_0} u_j(x) \right] \right\} \vee l_{k_0}(x) \\
&= \left[ 1 - \sum_{j \neq k_0} u_j(x) \right] \vee l_{k_0}(x).
\end{aligned}
\tag{33}
$$

The last equality is established as $l_j(x) - u_j(x) \leq 0$ for any $j \in [K]$, leading to the fact that $l_{k_0}(x) + \sum_{j \neq p_0, k_0} [l_j(x) - u_j(x)] \leq l_{k_0}(x)$. In Equation (33), we replace the upper-bound $u_{p_0}(x)$ with $\tilde{u}_{p_0}(x)$ in the definition of $\tilde{l}_{k_0}(x)$, because in Equation (31), the lowest possible values of $\eta_{k_0}(x)$ depends on the "realizable" maximal value of $\eta_{p_0}(x)$. However, notice that $\tilde{l}_{k_0}(x)$ does not depend on the label $p_0$ involved in the definition, indicating that the "realizable" lower-bound is an intrinsic property for the conditional probability for each $k_0 \in [K]$. Similar statement can be applied for the "realizable" upper-bound $\tilde{u}_{k_0}(x)$ for each $k_0 \in [K]$.

- **Step 3**. Back to the maximization of $A(t, \eta; x) \cdot \mathbb{I}\{t(x) = k_0\}$ in Equation (31), we have

$$
\begin{aligned}
\max_{\eta \in S} \left\{ A(t, \eta; x) \right\} \cdot \mathbb{I}\{t(x) = k_0\} &= \max_{p_0 \neq k_0 \eta \in S} \left( \eta_{p_0}(x) - \eta_{k_0}(x) \right)^+ \cdot \mathbb{I}\{t(x) = k_0\} \\
&= \max_{p_0 \neq k_0, p_0 \in [K]} \left( \tilde{u}_{p_0}(x) - \tilde{l}_{k_0}(x) \right)^+ \cdot \mathbb{I}\{t(x) = k_0\}.
\end{aligned}
\tag{34}
$$

By defining

$$
w_{k_0}(x) = \max_{p_0 \neq k_0, p_0 \in [K]} \left( \tilde{u}_{p_0}(x) - \tilde{l}_{k_0}(x) \right)^+
$$

as the weight function corresponding to the event $\{t(x) = k_0\}$, the Equation (34) can be written as

$$\max_{\eta(x) \in S(x)} \left\{ A(t, \eta; x) \right\} \cdot \mathbb{I}\{t(x) = k_0\} = w_{k_0}(x) \cdot \mathbb{I}\{t(x) = k_0\}.$$

The worst-case excess risk $\overline{\mathcal{R}}(t)$ can then be written as

$$
\begin{aligned}
\overline{\mathcal{R}}(t) &= \mathbb{E}\left[ \max_{\eta \in S} \sum_{k_0 \in [K]} A(t, \eta; X) \cdot \mathbb{I}\{t(X) = k_0\} \right] \\
&= \mathbb{E}\left[ \sum_{k_0 \in [K]} \max_{\eta \in S} A(t, \eta; X) \cdot \mathbb{I}\{t(X) = k_0\} \right] \\
&= \mathbb{E}\left[ \sum_{k_0 \in [K]} w_{k_0}(X) \cdot \mathbb{I}\{t(X) = k_0\} \right].
\end{aligned}
$$

The second inequality holds because $\{t(x) = k_0\}, k_0 \in [K]$ defines a sequence of disjoint events for every fixed $x \in \mathcal{X}$. By replacing the subscript $k_0$ with $k$, we complete the proof.

- **Step 4**. We are now left to check whether there exists a vector $\eta \in S$ such that the second equality in Equation (34) is guaranteed, that is, there is indeed a feasible solution of $\eta(x)$ with $\eta_{p_0}(x) = \tilde{u}_{p_0}(x)$, $\eta_{k_0}(x) = \tilde{l}_{k_0}(x)$ such that $\|\eta(x)\|_1 = 1$ and $\eta_j(x) \in [l_j(x), u_j(x)]$ for every $j \in [K]$. Before we go deep into the details, the non-emptyness of set $S$ reminds us that, for every $x \in \mathcal{X}$,

$$\sum_{j \in [K]} l_j(x) \leq 1 \leq \sum_{j \in [K]} u_j(x). \tag{35}$$

In the following, we check the feasibility of $\eta \in S$ for a fixed $x \in \mathcal{X}$ in three aspects.

1. For $\eta_{p_0}(x) = \tilde{u}_{p_0}(x)$, check $\eta_{p_0}(x) \in [l_{p_0}(x), u_{p_0}(x)]$.
   For the upper-bound, by the definition of $\tilde{u}_{p_0}(x)$ in Equation (32), $\eta_{p_0}(x) \leq u_p(x)$ is guaranteed.
   For the lower-bound, on one hand, if $\tilde{u}_{p_0}(x) = u_{p_0}(x)$, then $u_{p_0}(x) \geq l_{p_0}(x)$ naturally holds. On the other hand, suppose $\tilde{u}_{p_0}(x) = [1 - \sum_{j \neq p_0} l_j(x)]$. By Equation (35), we have $1 \geq \sum_{k \in [K]} l_k(x)$, which suggests that $\tilde{u}_{p_0}(x) \geq l_{p_0}(x)$. Hence, $\tilde{u}_{p_0}(x) \geq l_{p_0}(x)$ always holds.

2. For $\eta_{k_0}(x) = \tilde{l}_{k_0}(x)$, check $\eta_{k_0}(x) \in [l_{k_0}(x), u_{k_0}(x)]$.
   For the lower-bound, by the definition of $\tilde{l}_{k_0}(x)$ in Equation (33), $\eta_{k_0}(x) \geq l_{k_0}(x)$ always holds.
   For the upper-bound, by Equation (33), we have

$$\tilde{l}_{k_0}(x) = \left[1 - \sum_{j \neq k_0} u_j(x)\right] \vee l_{k_0}(x).$$

   Equation (35) ensures that $1 - \sum_{j \neq k_0} u_j(x) \leq u_{k_0}(x)$, and on the other hand $l_{k_0}(x) \leq u_{k_0}(x)$ simply holds. Therefore, we establish the upper-bound $\eta_{k_0}(x) \leq u_{k_0}(x)$.

3. Check $\|\eta(x)\|_1 = 1$. We start with computing the summation

$$\eta_{p_0}(x) + \eta_{k_0}(x) = \tilde{u}_{p_0}(x) + \tilde{l}_{k_0}(x).$$

   If the summation stays with in the capacity region $\left[1 - \sum_{j \neq p_0, k_0} u_j(x), 1 - \sum_{j \neq p_0, k_0} l_j(x)\right]$, we can always find a series of feasible solutions $\eta_j(x) \in [l_j(x), u_j(x)]$ for $j \neq p_0, k_0$ such that $\sum_{j \in [K]} \eta_j(x) = 1$ is satisfied. The discussions are summarized below.

   (a) Suppose $u_{p_0}(x) \leq 1 - \sum_{j \neq p_0} l_j(x)$, by the definition of $\tilde{u}_{p_0}(x)$ in Equation (32), we have $\tilde{u}_{p_0}(x) = u_{p_0}(x)$ and $\tilde{l}_{k_0}(x) = [1 - \sum_{j \neq k_0} u_j(x)] \vee l_{k_0}(x)$. We then have

$$
\tilde{u}_{p_0}(x) + \tilde{l}_{k_0}(x) = \begin{cases} 1 - \displaystyle\sum_{j \neq k_0, p_0} u_j(x), & \text{if } l_{k_0}(x) \leq 1 - \displaystyle\sum_{j \neq k_0} u_j(x) \quad \text{(Case 1)} \\ u_{p_0}(x) + l_{k_0}(x), & \text{if } l_{k_0}(x) \geq 1 - \displaystyle\sum_{j \neq k_0} u_j(x) \quad \text{(Case 2)}. \end{cases}
$$

For the Case 1, we can simply let $\eta_j(x) = u_j(x)$ for $j \neq p_0, k_0$, and this arrangement fulfills the requirement $\sum_{j \in [K]} \eta_j(x) = 1$.

For the Case 2, we have

$$1 - \sum_{j \neq k_0, p_0} u_j(x) \leq u_{p_0}(x) + l_{k_0}(x) \leq 1 - \sum_{j \neq k_0, p_0} l_j(x).$$

As a result, there exists a sequence of $\{\eta_j(x)\}_{j \neq p_0, k_0}$ such that $\eta_j(x) \in [l_j(x), u_j(x)]$ for all $j \neq p_0, k_0$ that satisifes $u_{p_0}(x) + l_{k_0}(x) + \sum_{j \neq p_0, k_0} \eta_j(x) = 1$.

(b) Suppose $u_{p_0}(x) \geq 1 - \sum_{j \neq p_0} l_j(x)$, we then have $\tilde{u}_{p_0}(x) = 1 - \sum_{j \neq p_0} l_j(x)$ and

$$\tilde{l}_{k_0}(x) = \left[1 - \tilde{u}_{p_0}(x) - \sum_{j \neq k_0, p_0} u_j(x)\right] \vee l_{k_0}(x)$$

$$= \left[l_{k_0}(x) + \sum_{j \neq k_0, p_0} [l_j(x) - u_j(x)]\right] \vee l_{k_0}(x)$$

$$= l_{k_0}(x),$$

where the last equality holds because $l_j(x) \leq u_j(x)$ for all $j \in [K]$. Therefore, we have

$$\tilde{u}_{p_0}(x) + \tilde{l}_{k_0}(x) = 1 - \sum_{j \neq k_0, p_0} l_j(x).$$

As a result, we can simply let $\eta_j(x) = l_j(x)$ for $j \neq p_0, k_0$, and this arrangement agains fullfills the requirement $\sum_{j \in [K]} \eta_j(x) = 1$.

The above analysis show that validity of the second equality in Equation (34).

$\square$

*Proof of Theorem 5.3 (Binary Classification Setting).* Recall from the first part of Theorem 5.3 that

$$\overline{\mathcal{R}}(t) = \mathbb{E}\left[\sum_{k=1}^{K} w_k(X) \cdot \mathbb{I}\{t(X) = k\}\right], \quad \text{where}$$

$$w_k(X) = \max_{p \neq k, p \in [K]} \left\{\left(\tilde{u}_p(x) - \tilde{l}_k(x)\right)^+\right\}.$$

Consider the binary case when $Y^\star \in \{-1, 1\}$ and $t : \mathcal{X} \mapsto \{-1, 1\}$. Recall that in Theorem 5.2, we derive the partial bound for $\eta_1^\star(x) = \mathbb{P}(Y^\star = 1 \mid X = x)$ as $[l_1(x), u_1(x)]$. By the constraint $\eta_1^\star(x) + \eta_0^\star(x) = 1$ for each $x \in \mathcal{X}$, we define the partial bound $[l_{-1}(x), u_{-1}(x)]$ for $\eta_{-1}^\star(x)$ as $l_{-1}(x) = 1 - u_1(x)$ and $u_{-1}(x) = 1 - l_1(x)$.

We can then compute the "realizable" upper- and lower-bounds for both $\eta_1^\star(x)$ and $\eta_{-1}^\star(x)$ as below:

$$\tilde{u}_1(x) = (1 - l_{-1}(x)) \wedge u_1(x) = u_1(x) \quad \text{and} \quad \tilde{u}_{-1}(x) = (1 - l_1(x)) \wedge u_{-1}(x) = 1 - l_1(x),$$

$$\tilde{l}_1(x) = (1 - u_{-1}(x)) \vee l_1(x) = l_1(x) \quad \text{and} \quad \tilde{l}_{-1}(x) = (1 - u_1(x)) \vee l_1(x) = 1 - u_1(x).$$

The weight functions $w_k(x)$ for $k = \pm 1$ are therefore

$$w_1(x) = \left(\tilde{u}_0(x) - \tilde{l}_1(x)\right)^+ = \left(1 - 2l_1(x)\right)^+ = \left(1 - 2l_1(x)\right) \cdot \mathbb{I}\{l_1(x) \leq 1/2\}$$

$$w_0(x) = \left(\tilde{u}_1(x) - \tilde{l}_0(x)\right)^+ = \left(2u_1(x) - 1\right)^+ = \left(2u_1(x) - 1\right) \cdot \mathbb{I}\{u_1(x) \geq 1/2\}.$$

Hence, in binary outcome case, the risk function $\overline{\mathcal{R}}(t)$ takes the form

$$\overline{\mathcal{R}}(t) = \mathbb{E}\left[\left|1 - 2l_1(X)\right| \cdot \mathbb{I}_{l_1(X) \leq 1/2} \cdot \mathbb{I}\{t(X) = 1\} + \left|2u_1(X) - 1\right| \cdot \mathbb{I}_{u_1(X) \geq 1/2} \cdot \mathbb{I}\{t(X) = 0\}\right]$$

$$= \mathbb{E}\left[\left|1 - 2l_1(X)\right| \cdot \mathbb{I}_{l_1(X) \leq 1/2} \cdot \left(1 - \mathbb{I}\{t(X) = 0\}\right) + \left|2u_1(X) - 1\right| \cdot \mathbb{I}_{u_1(X) \geq 1/2} \cdot \mathbb{I}\{t(X) = 0\}\right]$$

$$= \mathbb{E}\left[\left|1 - 2l_1(X)\right| \cdot \mathbb{I}_{l_1(X) \leq 1/2}\right] + \mathbb{E}\left[\mathbb{I}\{t(X) \neq 1\} \cdot \left(\left|2u_1(X) - 1\right| \cdot \mathbb{I}_{u_1(X) \geq 1/2} - \left|1 - 2l_1(X)\right| \cdot \mathbb{I}_{l_1(X) \leq 1/2}\right)\right]$$

$$= \mathbb{E}\left[\left|1 - 2l_1(X)\right| \cdot \mathbb{I}_{l_1(X) \leq 1/2}\right] + \mathbb{E}\left[\frac{1 - t(X)}{2} \cdot \left(\left|2u_1(X) - 1\right| \cdot \mathbb{I}_{u_1(X) \geq 1/2} - \left|1 - 2l_1(X)\right| \cdot \mathbb{I}_{l_1(X) \leq 1/2}\right)\right].$$

For every fixed $x \in \mathcal{X}$, define a weight function

$$w(x) := |u_1(x) - 1/2| \cdot \mathbb{I}\{u_1(x) \geq 1/2\} - |l_1(x) - 1/2| \cdot \mathbb{I}\{l_1(x) < 1/2\}$$
$$= \max\left(u_1(X) - 1/2, 0\right) + \min\left(l_1(X) - 1/2, 0\right),$$

we then have

$$\overline{\mathcal{R}}(t) = \underbrace{\mathbb{E}\left[|1 - 2l_1(X)| \cdot \mathbb{I}\{l_1(X) < 1/2\}\right]}_{(**)} + \mathbb{E}\left[w(X) \cdot \left(1 - t(X)\right)\right]$$

$$= (**) + \mathbb{E}\left[w(X)\right] + \mathbb{E}\left[-w(X) \cdot t(X)\right]$$

$$= (**) + \mathbb{E}\left[w(X)\right] + \mathbb{E}\left[|w(X)| \cdot \mathrm{sgn}[-w(X)] \cdot t(X)\right] \tag{36}$$

$$= (**) + \mathbb{E}\left[w(X)\right] + \mathbb{E}\left[|w(X)| \cdot \left(2\mathbb{I}\{\mathrm{sgn}[w(X)] \neq t(X)\} - 1\right)\right]$$

$$= (**) + \mathbb{E}\left[w(X) - |w(X)|\right] + 2\mathbb{E}\left[|w(X)| \cdot \mathbb{I}(\mathrm{sgn}[w(X)] \neq \mathrm{sgn}[h(X)])\right].$$

For any fixed $x$, notice that $w(x) - |w(x)| = 0$ when $w(x) \geq 0$ and $w(x) - |w(x)| = 2w(x)$ when $w(x) < 0$, we then have

$$\mathbb{E}\left[w(X) - |w(X)|\right] = \mathbb{E}\left[2w(X) \cdot \mathbb{I}\{w(X) < 0\}\right]$$

$$= \mathbb{E}\left[\left(|2u_1(x) - 1| \cdot \mathbb{I}\{u_1(X) \geq 1\} - |1 - 2l_1(X)| \cdot \mathbb{I}\{l_1(X) < 1/2\}\right) \cdot \mathbb{I}\{w(X) < 0\}\right].$$

Consequently, by observing the establishment of the event

$$w(X) < 0 \quad \Leftrightarrow \quad |2u_1(X) - 1| \cdot \mathbb{I}\{u_1(X) \geq 1/2\} \leq |l_1(X) - 1/2| \cdot \mathbb{I}\{l_1(X) < 1/2\},$$

the first two term in Equation (36) can be further organized as

$$\mathbb{E}\left[|1 - 2l_1(X)| \cdot \mathbb{I}\{l_1(X) < 1/2\}\right] + \mathbb{E}\left[w(X) - |w(X)|\right]$$

$$= \begin{cases} \mathbb{E}\left[|2u_1(X) - 1| \cdot \mathbb{I}\{u_1(X) \geq 1/2\}\right], & \text{if } w(X) < 0 \\ \mathbb{E}\left[|1 - 2l_1(X)| \cdot \mathbb{I}\{l_1(X) < 1/2\}\right], & \text{if } w(X) \geq 0 \end{cases} \tag{37}$$

$$= \min\left(\mathbb{E}\left[|1 - 2l_1(X)| \cdot \mathbb{I}\{l_1(X) < 1/2\}\right], \mathbb{E}\left[|2u_1(X) - 1| \cdot \mathbb{I}\{u(X) \geq 1/2\}\right]\right).$$

Finally, we define

$$w^{\mathrm{partial}}(x) = 2w(x) = \max\left(2u_1(X) - 1, 0\right) + \min\left(2l_1(X) - 1, 0\right)$$

for every $x \in \mathcal{X}$. Combining Equations (36) and (37) together, we have

$$\overline{\mathcal{R}}(h) = \mathbb{E}\left[|w^{\mathrm{partial}}(X)| \cdot \mathbb{I}\{\mathrm{sgn}[w^{\mathrm{partial}}(X)] \neq t(X)\}\right]$$

$$+ \min\left(\mathbb{E}\left[|1 - 2l_1(X)| \cdot \mathbb{I}\{l_1(X) < 1/2\}\right], \mathbb{E}\left[|2u_1(X) - 1| \cdot \mathbb{I}\{u_1(X) \geq 1/2\}\right]\right)$$

$$\square$$

## B.4. Comparisons of Point and Partial Identification

Point identification hinges on decision-makers' **homogeneity** in their use of unobservables, as defined by the generalized NUCEM assumption in Theorem 4.3. Concretely, the conditional irrelevant of $\mathrm{Cov}(D, Z \mid X, U)$ and $\mathbb{E}[Y_k^\star \mid X, U]$ given $X$ reflects the uniformity of decision-makers' reliance on unobserved information. Conversely, partial identification benefits from the **heterogeneity** of decision-makers' use of observables $X$, not requiring the homogeneity of selection on unobservables $U$. This approach uses IV partial bounds that consider variations in the decision rules $\mathbb{P}(D = 1 \mid X, Z)$ and the labeled probabilities $\mathbb{P}(Y = k, D = 1 \mid X, Z)$ across decision-makers $Z$. Such heterogeneity helps construct tighter bounds for $\eta_k^\star(x)$ Moreover, we having the following claims.

**Theorem B.2.** *The partial bounds $l_k(x)$ and $u_k(x)$ in Theorem 5.2 are achieved by the most lenient and most stringent decision-makers, respectively.*

*Proof of Theorem B.2.* We claim that for any fixed $X$ and $U$, the lower-bound is achieved by the most lenient decision-maker, and the upper-bound is achieved by the most stringent decision-maker. To see this, notice that the lower-bound for $\eta_k^\star$ with respect to decision-maker $Z = j$ in Theorem 5.2 satisfies

$$
\begin{aligned}
l_k(x, z) &:= \mathbb{E}\big[DY_k + a_k(X) \cdot (1 - D) \mid X = x, Z = z\big] \\
&= \mathbb{E}\big[DY_k^\star + a_k(X) \cdot (1 - D) \mid X = x, Z = z\big] \\
&= \mathbb{E}\Big[\mathbb{E}\big[DY_k^\star + a_k(X) \cdot (1 - D) \mid U, X, Z = z\big] \mid X = x, Z = z\Big] \\
&= \mathbb{E}\Big[\mathbb{P}(D = 1 \mid U, X, Z) \cdot \mathbb{E}\big[Y_k^\star \mid U, X, Z\big] + a_k(X) \cdot \big(1 - \mathbb{P}(D = 1 \mid U, X, Z)\big) \mid X = x, Z = z\Big] \\
&= a_k(x) + \mathbb{E}\Big[\mathbb{P}(D = 1 \mid U, X, Z) \cdot \big(\mathbb{E}[Y_k^\star \mid X, U] - a_k(X)\big) \mid X = x, Z = z\Big].
\end{aligned}
$$

The second equality holds consistency, and the third equality follows by iterated law of expectation. The fourth equality holds by Assumption 3.1 and the final equality is established by the IV independence in Assumption 3.2 ($Z \perp (U, Y^\star) \mid X$). Note that the Assumption 5.1 states $a_k(X) \leq E[Y_k^\star \mid X, U] \leq b_k(X)$ almost surely, the lower bound $l_k(x, z)$ is an increasing function of decision rule $\mathbb{P}(D = 1 \mid U, X, Z = z)$. Therefore, for any fixed $X$ and $U$, the maximization over lower bounds $\max_{z \in [m]} l_k(x, z)$ is achieved by the most lenient decision-maker with the highest decision rule $\mathbb{P}(D = 1 \mid X, U, Z = z)$, that is,

$$
\begin{aligned}
l_k(x) &= \max_{z \in \mathcal{Z}} l_k(x, z) \\
&= \max_{z \in \mathcal{Z}} a_k(x) + \mathbb{E}\Big[\mathbb{P}(D = 1 \mid U, X, Z = z) \cdot \big(\mathbb{E}[Y_k^\star \mid X, U] - a_k(X)\big) \mid X = x\Big] \\
&= a_k(x) + \mathbb{E}\Big[\big\{\max_{z \in \mathcal{Z}} \mathbb{P}(D = 1 \mid U, X, Z = z)\big\} \cdot \big(\mathbb{E}[Y_k^\star \mid X, U] - a_k(X)\big) \mid X = x\Big]
\end{aligned}
$$

Similarly, for the upper-bound, the minimization $\min_{z \in [m]} u_k(x, z)$ is achieved by the most stringent decision-maker when $X$ and $U$ are fixed. $\square$

## C. Supplements for Unified Cost-sensitive Learning

*Proof of Theorem 6.1.* To facilitate the analysis, for fixed weight function $w : \mathcal{X} \to \mathbb{R}$ and score function $h : \mathcal{X} \to \mathbb{R}^K$, we define the class labels

$$k_0 := \underset{k \in [K]}{\arg\min}\, w_k(x) \quad \text{and} \quad k_h := \underset{k \in [K]}{\arg\max}\, h_k(x).$$

Namely, when $x \in \mathcal{X}$ is fixed, $k_0$ denotes the class label with the minimal weight $w_{k_0}(x)$, while $k_h$ corresponds to the class label with the highest score $h_{k_h}(x)$. According to the definition of cost-sensitive classification risk $\mathcal{R}(h, w)$ in Equation (11), we define the cost-sensitive classification loss as

$$l(h, w; x) = \sum_{k=1}^{K} w_k(x) \mathbb{I}\{\underset{p \in [K]}{\arg\max}\, h_p(x) = k\}.$$

Obviously, the excess classification risk can be written as

$$\mathcal{R}(h, w) - \inf_h \mathcal{R}(h, w) = \mathbb{E}\big[l(h, w; X) - \inf_h l(h, w; X)\big] = \mathbb{E}\big[w_{k_h}(X) - w_{k_0}(X)\big].$$

Correspondingly, we define the cost-sensitive surrogate loss for $\mathcal{R}_{\exp}(\boldsymbol{h}, \boldsymbol{w})$ in Equation (12) as

$$l_{\exp}(h, w; x) = \sum_{k=1}^{K} w_k(x) \frac{\exp(h_k(x))}{\sum_{p=1}^{K} \exp(h_p(x))}.$$

As a result, the excess surrogate risk can be written as

$$\mathcal{R}_{\exp}(h, w) - \inf_h \mathcal{R}_{\exp}(h, w) = \mathbb{E}\big[l_{\exp}(h, w; X) - \inf_h l_{\exp}(h, w; X)\big].$$

Therefore, we are left to analyze the excess surrogate loss $l_{\exp}(h, w; x) - \inf_h l_{\exp}(h, w; x)$ for every fixed $x \in \mathcal{X}$.

Following the idea of (Mao et al., 2023), for every fixed scored function $h : \mathcal{X} \to \mathbb{R}^k$, we define a new function $h^\mu$ induced by some real-valued parameter $\mu$ as follow: for fixed $x \in \mathcal{X}$,

$$\begin{cases} h_k^\mu(x) = h_k(x), & k \neq k_0, k_h \\ h_{k_0}^\mu(x) = \log\big(\exp(h_{k_h}(x)) - \mu\big), & k = k_0 \\ h_{k_h}^\mu(x) = \log\big(\exp(h_{k_0}(x)) + \mu\big), & k = k_h. \end{cases}$$

Since the domain of $\log$ function is $(0, +\infty)$, we should restrict the value of $\mu$ to ensure that $\exp(h_{k_h}(x)) + \mu > 0$ and $\exp(h_{k_0}(x)) - \mu > 0$. Therefore, for fixed $x \in \mathcal{X}$, we define the feasible region of $\mu$ as

$$\mathcal{M}(\mu) := \big\{\mu \in \mathbb{R} : -\exp(h_{k_h}(x)) < \mu < \exp(h_{k_0}(x))\big\}.$$

A direct consequence of above definition is that $\sum_{k=1}^{K} \exp(h_k(x)) = \sum_{k=1}^{K} \exp(h_k^\mu(x))$ for every fixed $x$. Consequently, the surrogate loss $l(h^\mu, w; x)$ takes the form

$$l_{\exp}(h^\mu, w; x) = \sum_{k \neq k_0, k_h} w_k(x) \frac{\exp(h_k(x))}{\sum_{p=1}^{K} \exp(h_p(x))} + w_{k_0}(x) \frac{\exp(h_{k_h}(x)) - \mu}{\sum_{p=1}^{K} \exp(h_p(x))} + w_{k_h}(x) \frac{\exp(h_{k_0}(x)) + \mu}{\sum_{p=1}^{K} \exp(h_p(x))}.$$

Now, the excess surrogate loss $l_{\exp}(h, w; x) - \inf_h l_{\exp}(h, w; x)$ can be lower-bounded by

$$l_{\exp}(h, w; x) - \inf_h l_{\exp}(h, w; x)$$

$$\geq l_{\exp}(h, w; x) - \inf_{\mu \in \mathcal{M}(\mu)} l_{\exp}(h^\mu, w; x)$$

$$= \sup_{\mu \in \mathcal{M}(\mu)} \left\{ w_{k_h}(x) \frac{\exp(h_{k_h}(x)) - \exp(h_{k_0}(x)) - \mu}{\sum_{p=1}^{K} \exp(h_p(x))} + w_{k_0}(x) \frac{\exp(h_{k_0}(x)) - \exp(h_{k_h}(x)) + \mu}{\sum_{p=1}^{K} \exp(h_p(x))} \right\}$$

$$= \sup_{\mu \in \mathcal{M}(\mu)} \left\{ \big(w_{k_h}(x) - w_{k_0}(x)\big) \frac{\exp(h_{k_h}(x)) - \exp(h_{k_0}(x)) - \mu}{\sum_{p=1}^{K} \exp(h_p(x))} \right\}$$

The first inequality holds by the fact that the infimum of $\mu$ over set $\mathcal{M}(\mu)$ is always greater then the infimum of $h$ over all measurable functions. Notice that $k_0$ is the class label with the minimal weight, we have $w_{k_h}(x) - w_{k_0}(x) \geq 0$ for any $x$. Consequently, the supremum of the above expression is achieved when $\mu = -\exp(h_{k_0}(x))$, and the excess surrogate loss is therefore lower-bounded by

$$l_{\exp}(h, w; x) - \inf_h l_{\exp}(h, w; x) \geq \left(w_{k_h}(x) - w_{k_0}(x)\right) \frac{\exp(h_{k_h}(x))}{\sum_{p=1}^K \exp(h_p(x))}$$

$$\geq \frac{1}{K}\left[w_{k_h}(x) - w_{k_0}(x)\right]$$

$$= \frac{1}{K}\left[l(h, w; x) - \inf_h l(h, w; x)\right].$$

The last inequality follows by the definition that $k_h$ is the class label with the highest score $h_{k_h}$, and therefore we must have $\frac{\exp(h_{k_h}(x))}{\sum_{p=1}^K \exp(h_p(x))} \geq \frac{1}{K}$ for any fixed $x \in \mathcal{X}$. By taking the expectation over $X$ on the both sides, we completes the proof.

$\square$

*Proof of Proposition 6.2.* For the weighted classification risk $\mathcal{R}(h, w)$, if we choose $\tilde{h}(x) = w(x)$ for $x \in \mathcal{X}$, we simply have $\mathcal{R}(\tilde{h}, w) = 0$, and this suggests the Bayes optimal risk equals to zero, that is, $\inf_h \mathcal{R}(h, w) = 0$. Therefore, the excess weighted classification risk can be written as

$$\mathcal{R}(h, w) - \inf_h \mathcal{R}(h, w) = \mathcal{R}(h, w) - 0 = \mathbb{E}\left[|w(X)| \cdot \mathbb{I}\{\operatorname{sgn}[h(X)] \neq \operatorname{sgn}[w(X)]\}\right].$$

Here we use the fact that $h^*(x)$ has the same sign as weight function $w(x)$ for any fixed $x \in \mathcal{X}$.

For the $\phi$-risk introduce in Equation (13), we consider the following surrogate loss functions:

$$\begin{aligned}
\text{Hinge loss}: && \phi(\alpha) &= \max\{1 - \alpha, 0\} \\
\text{logistic loss}: && \phi(\alpha) &= \log(1 + e^{-\alpha}) \quad. \\
\text{exponential loss}: && \phi(\alpha) &= e^{-\alpha}
\end{aligned}$$

Notice that for any $\phi \in \{\text{Hinge}, \text{logistic}, \text{exponential}\}$, the surrogate loss $\phi(\alpha)$ is lower-bounded (in fact, $\inf_\alpha \phi(\alpha) = 0$), then we have $\inf_\alpha \mathbb{E}[\phi(\alpha(X))] = \mathbb{E}[\inf_\alpha \phi(\alpha(X))] = 0$. Therefore, the excess $\phi$-risk takes the form

$$\mathcal{R}_\phi(h, w) - \inf_h \mathcal{R}(h, w) = \mathcal{R}_\phi(h, w) - 0 = \mathbb{E}\left[|w(X)| \cdot \phi\big(h(X) \cdot \operatorname{sgn}[w(X)]\big)\right].$$

In the sequel, we only need to check if the surrogate loss $\phi(h(x) \cdot w(x))$ provides an upper-bound for the $0 - 1$ loss $\mathbb{I}\{\operatorname{sgn}[h(x)] \neq \operatorname{sgn}[w(x)]\}$ for each $\phi$ in the set $\{\text{Hinge}, \text{logistic}, \text{exponential}\}$. We fix the weight function $w$ and the observed features $x \in \mathcal{X}$ in the following discussion.

- Consider the case when $\operatorname{sgn}[h(x)] \neq \operatorname{sgn}[w(x)]$, we then have $\mathbb{I}\{\operatorname{sgn} h(x) \neq \operatorname{sgn}[w(x)]\} = 1$. For the $\phi$-risk, we have

$$\phi\big(h(x) \cdot \operatorname{sgn}[w(x)]\big) = \phi(-|h(x)|)$$

  for any $\phi \in \{\text{Hinge}, \text{logistic}, \text{exponential}\}$. Concretely, we have

$$\phi_{\text{Hinge}}(-|h(x)|) = (1 + |h(x)|)^+ \geq 1,$$
$$\phi_{\text{logistic}}(-|h(x)|) = \log_2(1 + e^{|h(x)|}) \geq 1,$$
$$\phi_{\text{exponential}}(-|h(x)|) = e^{|h(x)|} \geq 1.$$

  Therefore, we have

$$\phi(h(x) \cdot \operatorname{sgn}[w(x)]) \geq \mathbb{I}\{\operatorname{sgn}[h(x)] \neq \operatorname{sgn}[w(x)]\}$$

  for any $\phi \in \{\text{Hinge}, \text{logistic}, \text{exponential}\}$.

- Consider the case when $\mathrm{sgn}[h(x)] = \mathrm{sgn}[w(x)]$, we then have $\mathbb{I}\{\mathrm{sgn}\, h(x) \neq \mathrm{sgn}[w(x)]\} = 0$. For the $\phi$-risk, we have

$$\phi\big(h(x) \cdot \mathrm{sgn}[w(x)]\big) = \phi(|h(x)|)$$

for any $\phi \in \{\mathrm{Hinge, logistic, exponential}\}$. Concretely, we have

$$\phi_{\mathrm{Hinge}}(|h(x)|) = (1 - |h(x)|)^{+} \geq 0,$$
$$\phi_{\mathrm{logistic}}(|h(x)|) = \log_2(1 + e^{-|h(x)|}) \geq 0,$$
$$\phi_{\mathrm{exponential}}(|h(x)|) = e^{-|h(x)|} \geq 0.$$

Hence, we have

$$\phi(h(x) \cdot \mathrm{sgn}[w(x)]) \geq \mathbb{I}\{\mathrm{sgn}[h(x)] \neq \mathrm{sgn}[w(x)]\}$$

for any $\phi \in \{\mathrm{Hinge, logistic, exponential}\}$.

Finally, by multiplying the common weight $|w(X)|$ and taking the expectation over $X$, we conclude that when the weight function $w$ is fixed, the excess $\phi$-risk is always an upper-bound for the excess weighted risk for any $\phi \in \{\mathrm{Hinge, logistic, exponential}\}$:

$$\mathcal{R}_\phi(h, w) - \inf_h \mathcal{R}_\phi(h, w) \geq \mathcal{R}(h, w) - \inf_h \mathcal{R}(h, w).$$

$\square$

# D. Generalization Error Bound

In this section, we theoretically analyze the performance of the UCL algorithm in Algorithm 1 by deriving a generalization error bound for the output score function $\widehat{h}$. Specifically, Theorem 6.1 establishes a calibration bound for the cost-sensitive excess risk $\mathcal{R}(\widehat{h}) - \inf_{h \in \mathcal{H}} \mathcal{R}(h)$ induced by the output score function. To leverage this result, we first derive a generalization error bound for the excess surrogate risk $\mathcal{R}_{\exp}(\widehat{h}) - \inf_{h \in \mathcal{H}} \mathcal{R}_{\exp}(h)$.

To facilitate the analysis, we define the surrogate risk induced by the estimated weight functions $\boldsymbol{w}^{[l]}$ from batch $l \in [K]$:

$$\mathcal{R}_{\exp}(\boldsymbol{h}, \widehat{\boldsymbol{w}}^{[l]}) := \mathbb{E}\Big[ \sum_{k=1}^{K} \widehat{w}_k^{[l]}(X) \cdot \frac{\exp(h_k(X))}{\sum_{p=1}^{K} \exp(h_p(X))} \Big].$$

The below lemma provides a decomposition of the upper-bound of the excess surrogate risk into two components: the estimation error of the nuisance functions $\{w_k\}_{k=1}^{K}$ and the empirical process error.

**Lemma D.1.** *The excess surrogate risk $\mathcal{E}(\mathcal{H}) := \mathcal{R}_{\exp}(\widehat{h}) - \inf_{h \in \mathcal{H}} \mathcal{R}_{\exp}(h)$ satisfies*

$$\mathcal{E}(\mathcal{H}) \leq \underbrace{\frac{2}{L} \sum_{l=1}^{L} \sup_{\boldsymbol{h} \in \mathcal{H}} \Big| \mathcal{R}_{\exp}(\boldsymbol{h}, \widehat{\boldsymbol{w}}^{[l]}) - \mathcal{R}_{\exp}(\boldsymbol{h}, \boldsymbol{w}) \Big|}_{\text{Nuisance Estimation Error}} + \underbrace{\frac{2}{L} \sum_{l=1}^{L} \sup_{\boldsymbol{h} \in \mathcal{H}} \Big| \widehat{\mathcal{R}}_{\exp}(\boldsymbol{h}, \widehat{\boldsymbol{w}}^{[l]}) - \mathcal{R}_{\exp}(\boldsymbol{h}, \widehat{\boldsymbol{w}}^{[l]}) \Big|}_{\text{Empirical Process Error}}.$$

The first term in Lemma D.1 captures the error arising from the estimation of the unknown weight function $\boldsymbol{w}$, which is evaluated on the true excess surrogate risk with the supremum over all $\boldsymbol{h} \in \mathcal{H}$. The second term reflects the error incurred by the estimated score function $\widehat{\boldsymbol{h}}$, evaluated as the supremum of an empirical process. To bound these two terms, we need to further specify the estimation errors of the nuisance estimators, as we will do in Assumptions D.2 and D.3.

**Assumption D.2** (Nuisances Estimation). We assume that the true weight functions $\{w_k\}_{k=1}^{[K]}$ and their estimates $\{\widehat{w}_k^{[1]}, \ldots, \widehat{w}_k^{[L]}\}_{k=1}^{K}$ satisfy the following conditions:

1. they are uniformly bounded by some constant $C_w$ over $\mathcal{X}$, namely, $\|w_k\|_\infty < C_w, \|\widehat{w}_k^{[l]}\|_\infty < C_w$ for all $k \in [K]$ and $l \in [L]$;

2. there exists $\gamma > 0$ such that
$$\mathbb{E}\big[\|\widehat{\boldsymbol{w}}^{[l]}(X) - \boldsymbol{w}(X)\|_1\big] = \mathcal{O}(N^{-\gamma}), \ \forall l \in [L].$$

Assumption D.2 requires that the nuisance functions $\{w_k\}_{k=1}^{K}$ and their estimates $\{\widehat{w}_k^{[l]}\}_{k=1}^{K}$ are uniformly bounded over the feature space $\mathcal{X}$ and that the estimation error of the nuisance functions decays at a rate of $\mathcal{O}(N^{-\gamma})$ for some $\gamma > 0$.

**Assumption D.3** (Function Class Complexity). Let $C_h > 0$ and $\tau > 0$ be some constants. For any score function candidate $h \in \mathcal{H}$, we assume $\|h_k\|_\infty < C_h$ for any $k \in [K]$. Moreover, assume that the $\epsilon$-bracketing number of $\mathcal{H}$ with respect to the metric $\rho$, denoted as $\mathcal{N}(\epsilon, \mathcal{H}, \rho)$, satisfies $\log \mathcal{N}(\epsilon, \mathcal{H}, \rho) \leq \epsilon^{-\tau}$ for any $\epsilon \in (0, 1)$ and some $\tau < 2$.

In Assumption D.3, we further restrict the complexity of the hypothesis class $\mathcal{H}$ by imposing the boundedness on each component of score function $h$, and limiting the growth rate of its covering number. The boundedness condition is reasonable, as if any component of score function, say $h_k$, goes to infinity, then we cannot get reasonable classifier by the $\arg\min$ operations. At the same time, limiting the growth rate of covering number is a common way to quantify the complexity of a function class in statistical learning theory (Massart & Nédélec, 2006; Shalev-Shwartz & Ben-David, 2014; Wainwright, 2019).

**Nuisance Estimation Error** To provide a finite-sample error bound for the estimation error, we start with bounding the nuisance estimation error introduced in Lemma D.1.

**Proposition D.4** (Nuisance Estimation Error). *Suppose Assumption D.2 and Assumption D.3 hold. We have, for any $l \in [L]$,*

$$\sup_{h \in \mathcal{H}} \Big| \mathcal{R}_{\exp}(\boldsymbol{h}, \widehat{\boldsymbol{w}}^{[l]}) - \mathcal{R}_{\exp}(\boldsymbol{h}, \boldsymbol{w}) \Big| = \mathcal{O}(N^{-\gamma}).$$

**Empirical Process Error**   We now consider bounding the supremum of empirical process introduced in Lemma D.1. We focus on analyzing the empirical process define on the $l$-th batch, that is,

$$\widehat{\mathcal{R}}_{\exp}(\boldsymbol{h}, \widehat{\boldsymbol{w}}^{[l]}) - \mathcal{R}_{\exp}(\boldsymbol{h}, \widehat{\boldsymbol{w}}^{[l]}) = \frac{1}{|I_l|} \sum_{i \in I_l} \sum_{k=1}^{K} \widehat{w}_k^{[l]}(X_i) \frac{\exp(h_k(X_i))}{\sum_{p=1}^{K} \exp(h_p(X_i))} - \mathbb{E}\left[ \sum_{k=1}^{K} \widehat{w}_k^{[l]}(X) \frac{\exp(h_k(X))}{\sum_{p=1}^{K} \exp(h_p(X))} \right].$$

As the cross-fitting batch size $L$ involved in Algorithm 1 is usually smaller than the sample size $N$, here we simply assume $|I_m| \approx N$. Note that, the estimated weight function in the $l$-th batch $\widehat{\boldsymbol{w}}^{[l]}$ is a random variable which relies on observed variables $(Y, D, X, Z)$. To facilitate our analysis, we denote a new variable $V = (Y, D, X, Z)$ over the domain $\mathcal{V} = \mathcal{Y} \times \mathcal{D} \times \mathcal{X} \times \mathcal{Z}$ with some joint distribution $P_V$. We will omit the superscript $[l]$ for simplicity in the sequel. For every given estimates $\{\widehat{w}\}_{k=1}^{K}$, we define a *excess surrogate risk class* $\mathcal{F}_\mathcal{H}$ as

$$\mathcal{F}_\mathcal{H} := \left\{ f_h : v \mapsto \sum_{k=1}^{K} \hat{w}_k(x) \frac{\exp(h_k(x))}{\sum_{p=1}^{K} \exp(h_p(x))} \,\Big|\, h \in \mathcal{H} \right\} \tag{38}$$

To simplify our analysis, we use shorthand notation $Pf = \mathbb{E}[f(V)]$ and $P_N f = \frac{1}{N} \sum_{i=1}^{N} f(V_i)$, where $P_N$ is the empirical measured associated with the i.i.d. sample of size $N$. Then the supermom of empirical process can be written as

$$\|P_N f - P f\|_{\mathcal{F}_\mathcal{H}} := \sup_{f \in \mathcal{F}_\mathcal{H}} P_N f - P f = \sup_{h \in \mathcal{H}} \widehat{\mathcal{R}}_\Phi(h, \widehat{w}) - \mathcal{R}_\Phi(h, \widehat{w}). \tag{39}$$

Therefore, our goal is now provide a uniform bound for the empirical process $P_N f - P f$ over function class $\mathcal{F}_\mathcal{H}$. To realize this target, we shall notice that $\mathcal{F}_\mathcal{H}$ is a uniformly bounded class, that is, for any $f \in \mathcal{F}_\mathcal{H}$, we have $\|f\|_\infty := \sup_{v \in \mathcal{V}} f(v)$ being bounded. To see this, notice that in Assumption D.2, we assume $\|\widehat{w}_k\|_\infty = \sup_{x \in \mathcal{X}} \widehat{w}_k(x) < C_w$ for any $k \in [K]$. Consequently, we have

$$
\begin{aligned}
|f(v)| = \left| \sum_{k=1}^{K} \widehat{w}_k(x) \frac{\exp(h_k(x))}{\sum_{p=1}^{K} \exp(h_p(x))} \right| &\leq \sum_{k=1}^{K} \left| \widehat{w}_k(x) \frac{\exp(h_k(x))}{\sum_{p=1}^{K} \exp(h_p(x))} \right| \\
&\leq \sum_{k=1}^{K} \left| \widehat{w}_k(x) \right| \cdot \left| \frac{\exp(h_k(x))}{\sum_{p=1}^{K} \exp(h_p(x))} \right| \\
&\leq K C_w := C_f.
\end{aligned}
$$

Now, given a sequence of i.i.d. sample $\{W_1, \ldots, W_N\}$, we define the *empirical Rademacher complexity* with respect to class $\mathcal{F}_\mathcal{H}$ as

$$\widehat{R}_N(\mathcal{F}_\mathcal{H}) := \mathbb{E}_\sigma \left[ \sup_{f \in \mathcal{F}_\mathcal{H}} \left| \frac{1}{N} \sum_{i=1}^{N} \sigma_i f(W_i) \right| \right], \tag{40}$$

where $(\sigma_1, \ldots, \sigma_N)$ is a vector of i.i.d. *Rademacher variables* takin values in $\{-1, +1\}$ with equal probability. The empirical Rademacher complexity provides an upper bound for the supremum of empirical process in Equation (39). See the lemma below.

**Lemma D.5** (Symmetrization Bound, Wainwright (2019) Theorem 4.10). *For the $C_f$-uniformly bounded class of function $\mathcal{F}_\mathcal{H}$, with probability at least $1 - \delta$, we have*

$$\|P_N f - P f\|_{\mathcal{F}_\mathcal{H}} \leq 2\widehat{R}_N(\mathcal{F}_\mathcal{H}) + C_f \sqrt{\frac{2\log(1/\delta)}{N}}.$$

Define the norm $L_2(P_N)$ for the function $f \in \mathcal{F}_\mathcal{H}$ such that $\|f - g\|_{L_2(P_N)} := \sqrt{\frac{1}{N} \sum_{i=1}^{N} \left( f(V_i) - g(V_i) \right)^2}$ for any function $f, g \in \mathcal{F}_\mathcal{H}$. As the function class $\mathcal{F}_\mathcal{H}$ is $C_f$-uniformly bounded, we then have

$$\sup_{f, g \in \mathcal{F}_\mathcal{H}} \|f - g\|_{L_2(P_N)} \leq 2C_f.$$

Moreover, let $\mathcal{N}\left(\epsilon, \mathcal{F}_\mathcal{H}, L_2(P_N)\right)$ be the covering number of class $\mathcal{F}_\mathcal{H}$ with respect to the metric induced by the $L_2(P_N)$ norm. According to the definition of $\mathcal{F}_\mathcal{H}$ in Equation (38), we have $\mathcal{N}\left(\epsilon, \mathcal{F}_\mathcal{H}, L_2(P_N)\right) \leq \mathcal{N}\left(\epsilon, \mathcal{H}, L_2(P_N)\right)$. Along with the limitation of the growing rate of $\mathcal{N}\left(\epsilon, \mathcal{H}, L_2(P_N)\right)$ in Assumption D.3, we establish the following *Dudley's entropy integral* for the empirical Rademacher complexity defined in Equation (40)

**Lemma D.6** (Dudley's Entropy Integral). *Assume Assumption D.3 holds. Given that $\sup_{f,g \in \mathcal{F}_{\mathcal{H}}} \|f - g\|_{L_2(P_N)} \leq 2C_f$, there exists a constant $C_0 > 0$ such that the empirical Rademacher complexity is almost surely upper-bounded by the Dudely's integral: $0 < \tau < 2$,*

$$\widehat{R}_N(\mathcal{F}_{\mathcal{H}}) \leq \frac{C_0}{\sqrt{N}} \int_0^{2C_f} d\epsilon \sqrt{\log \mathcal{N}(\epsilon, \mathcal{F}_{\mathcal{H}}, L_2(P_N))} \leq \frac{C_0}{\sqrt{N}} \int_0^{2C_f} \epsilon^{-\tau/2} d\epsilon = \frac{2C_0(2C_f)^{1-\tau/2}}{2-\tau} N^{-1/2}.$$

Given the establishment of Assumption D.3, we can now provide a generalization error bound for the empirical process error in Lemma D.1 by combining Lemma D.5 and Lemma D.6: with probability at least $1 - \delta$, we have

$$\sup_{\boldsymbol{h} \in \mathcal{H}} \left\{ \widehat{\mathcal{R}}_{\exp}(\boldsymbol{h}, \widehat{\boldsymbol{w}}) - \mathcal{R}_{\exp}(\boldsymbol{h}, \widehat{\boldsymbol{w}}) \right\} = \|P_n f - Pf\|_{\mathcal{F}_{\mathcal{H}}} = \mathcal{O}\left(N^{-1/2} \sqrt{2 \log(1/\delta)}\right). \tag{41}$$

**Generalization Error Bound for Excess Cost-sensitive Risk** Combining with the calibration bound in Theorem 6.1, the decomposition error of the excess surrogate risk in Lemma D.1, the nuisance estimation error in Proposition D.4, and the empirical process error in Equation (41), we can now provide a generalization error bound for original the excess cost-sensitive risk $\mathcal{R}(\widehat{\boldsymbol{h}}) - \inf_{\boldsymbol{h} \in \mathcal{H}} \mathcal{R}(\boldsymbol{h})$ induced by the output score function $\widehat{\boldsymbol{h}}$ in Algorithm 1. Formally speaking, we probability at least $1 - \delta$, we have

$$\mathcal{R}(\widehat{\boldsymbol{h}}, w) - \inf_{\boldsymbol{h} \in \mathcal{H}} \mathcal{R}(\boldsymbol{h}, \boldsymbol{w}) = \mathcal{O}\left(K \cdot \max\left\{N^{-\gamma}, N^{-1/2} \sqrt{\log(1/\delta)}\right\}\right). \tag{42}$$

This result shows that the excess cost-sensitive risk of the output score function $\widehat{\boldsymbol{h}}$ converges to the optimal risk at a rate of $K \cdot N^{-1/2}$, which is the standard rate for the multiclass classification risk in statistical learning theory (Massart & Nédélec, 2006; Shalev-Shwartz & Ben-David, 2014; Wainwright, 2019).

### D.1. Remaining Proofs

*Proof of Lemma D.1.* Let $h_{\mathcal{H}}^{\star} \in \mathcal{H}$ denote the best-in-class score function that minimizes the excess surrogate risk $\mathcal{R}_{\exp}(h, w)$. We use $w$ denote the true weight function and $\widehat{w}^{[l]}$ representes its estiamte from batch $l \in [L]$. By definition, we have

$$\mathcal{E}(\mathcal{H}) = \mathcal{R}_{\exp}(\hat{h}, w) - \mathcal{R}_{\exp}(h_{\mathcal{H}}^{\star}, w)$$

$$= \mathcal{R}_{\exp}(\hat{h}, w) - \frac{1}{L}\sum_{l=1}^{L} \mathcal{R}_{\exp}(\hat{h}, \widehat{w}^{[l]}) + \frac{1}{L}\sum_{l=1}^{L} \mathcal{R}_{\exp}(\hat{h}, \widehat{w}^{[l]}) - \frac{1}{L}\sum_{l=1}^{L} \widehat{\mathcal{R}}_{\exp}(\hat{h}, \widehat{w}^{[l]})$$

$$+ \frac{1}{L}\sum_{l=1}^{L} \widehat{\mathcal{R}}_{\exp}(\hat{h}, \hat{w}^{[l]}) - \frac{1}{L}\sum_{l=1}^{L} \mathcal{R}_{\exp}(h_{\mathcal{H}}^{\star}, \widehat{w}^{[l]}) + \frac{1}{L}\sum_{l=1}^{L} \mathcal{R}_{\exp}(h_{\mathcal{H}}^{\star}, \widehat{w}^{[l]}) - \mathcal{R}_{\exp}(h_{\mathcal{H}}^{\star}, w)$$

$$\leq \underbrace{\frac{2}{L}\sum_{l=1}^{L} \sup_{h \in \mathcal{H}} \left|\mathcal{R}_{\exp}(h, \widehat{w}^{[l]}) - \mathcal{R}_{\exp}(h, w)\right|}_{\text{Nuisance Estimation Error}} + \underbrace{\frac{2}{L}\sum_{l=1}^{L} \sup_{h \in \mathcal{H}} \left|\widehat{\mathcal{R}}_{\exp}(h, \widehat{w}^{[l]}) - \mathcal{R}_{\exp}(h, \widehat{w}^{[l]})\right|}_{\text{Empirical Process Error}}.$$

$\square$

*Proof of Proposition D.4.* Notice that for any function $h \in \mathcal{H}$ and batch $l \in [L]$,

$$
\begin{aligned}
\left| \mathcal{R}_{\exp}(h, \widehat{w}^{[l]}) - \mathcal{R}_{\exp}(h, w) \right| &= \left| \mathbb{E}\left[ \sum_{k=1}^{K} \left( \widehat{w}_k^{[l]}(X) - w_k(X) \right) \cdot \frac{\exp(h_k(X))}{\sum_{p=1}^{K} \exp(h_p(X))} \right] \right| \\
&\leq \mathbb{E}\left[ \sum_{k=1}^{K} \left| \left( \widehat{w}_k^{[l]}(X) - w_k(X) \right) \cdot \frac{\exp(h_k(X))}{\sum_{p=1}^{K} \exp(h_p(X))} \right| \right] \\
&\leq \mathbb{E}\left[ \sum_{k=1}^{K} \left| \widehat{w}_k(X) - w_k(X) \right| \cdot \left| \frac{\exp(h_k(X))}{\sum_{p=1}^{K} \exp(h_p(X))} \right| \right] \\
&\leq \mathbb{E}\left[ \sum_{k=1}^{K} \left| \widehat{w}_k(X) - w_k(X) \right| \right] = \mathbb{E}\left\| \widehat{w}(X) - w(X) \right\|_1.
\end{aligned}
$$

The first inequality comes from the fact that the absolute value function $|\cdot|$ is convex and we use Jensen's inequality. The second inequality follows from Cauchy-Schwarz inequality, and the third inequality follows from the fact that the $\mathrm{softmax}$ function is uniformly bounded by 1 for all $x \in \mathcal{X}$ and $h \in \mathcal{H}$. Finally, combining with Assumption D.2 and the results above, we have for every batch $l \in [L]$

$$
\sup_{h \in \mathcal{H}} \left| \mathcal{R}_{\exp}(h, \widehat{w}^{[l]}) - \mathcal{R}_{\exp}(h, w) \right| = \mathcal{O}(N^{-\gamma}).
$$

$\square$

# E. Supplement to Numeric Experiments

## E.1. Synthetic Dataset

In this section, we construct a synthetic dataset with multi-valued selective labels. The experiment results are illustrated in Figures 4 and 5, which show that our unified cost-sensitive learning (UCL) can achieve superior performance under various strength of selection bias.

**Data Generating Process**  Consider a dataset consisting of observable variables $\mathbf{X} = (X_1, X_2, \ldots, X_p)$, and an unobservable variable $\mathbf{U} = (U_1, U_2, \ldots, U_q)$, following the joint Gaussian distribution. We also introduce the variable $Z$ that represents the random assignment of decision-makers, which is uniformly drawn from the set $\mathcal{Z} = \{1, 2, \ldots, J\}$. The missingness decision $D \in \{0, 1\}$ is modeled as Bernoulli distributed variables with parameters $p_D := \mathbb{P}(D = 1 \mid X, U, Z)$, and the true label $Y^\star \in \{1, \ldots, K\}$ is modeled as a categorical variable with parameters $p_k := \mathbb{P}(Y = k \mid X, U, Z)$ for $k \in [K]$. The complete data generating process is summarized in Equation (43).

$$
\begin{aligned}
\text{Observed variables:} \quad & \mathbf{X} = (X_1, \ldots, X_p) \sim 2 \cdot \mathcal{N}(0, I_p), \;\; Z \sim \text{Uniform}(\{1, \ldots, J\}) \\
\text{Unobserved variables:} \quad & \mathbf{U} = (U_1, \ldots, U_q) \sim 2 \cdot \mathcal{N}(0, I_q) \\
\text{Coefficients:} \quad & \mathbf{W}_{X \to Y} = \big[i + k\big]_{p \times K} \quad i \in \{1, \ldots, p\}, \quad k \in \{1, \ldots, K\} \\
& \mathbf{W}_{U \to Y} = \big[i + k\big]_{q \times K} \quad i \in \{1, \ldots, q\}, \quad k \in \{1, \ldots, K\} \\
& \mathbf{W}_{X \to D} = \big[2 - i\big]_{p \times 1} \quad i \in \{1, \ldots, p\}, \\
& \mathbf{W}_{U \to D} = \big[1 + i\big]_{q \times 1} \quad i \in \{1, \ldots, q\}. \\
\text{Decision (NUCEM):} \quad & p_D = (1 - \alpha_D) \cdot \text{expit}\big\{Z \cdot 2\mathbf{X}\mathbf{W}_{X \to D}\big\} + \alpha_D \cdot \text{expit}\big\{3\mathbf{U}\mathbf{W}_{U \to D}\big\}, \\
\text{Decision (UC):} \quad & p_D = \text{expit}\big\{(1 - \alpha_D) \cdot Z \cdot 2\mathbf{X}\mathbf{W}_{X \to D} + \alpha_D \cdot 3\mathbf{U}\mathbf{W}_{U \to D}\big\}. \\
\text{Outcome:} \quad & p_k = \text{softmax}\big\{(1 - \alpha_Y) \cdot \mathbf{X}\mathbf{W}_{X \to Y} + \alpha_Y \cdot 4\mathbf{U}\mathbf{W}_{U \to Y}\big\}, \quad k \in \{1, \ldots, K\} \\
& Y = \text{Categorical}(p_1, \ldots, p_K).
\end{aligned}
\tag{43}
$$

Here we define the functions $\text{expit} : v \mapsto 1/(1 + \exp(-v))$ and $\text{softmax} : \mathbf{v} \mapsto \exp(\mathbf{v})/\sum_{k=1}^{K} \exp(v_k)$ for any vector $\mathbf{v} \in \mathbb{R}^K$. In both UC and NUCEM models, the parameter $\alpha_D \in (0, 1)$ controls the impact of unobservable variables $\mathbf{U}$ on the probability of missingness $p_D$, while the parameter $\alpha_Y \in (0, 1)$ adjust the magnitude of $\mathbf{U}$ affecting the distribution of outcome $Y^*$. Overall, the pair of $(\alpha_D, \alpha_Y)$ jointly determine the degree of selection bias in our data generating process.

In this experiment, we simulate a full dataset with sample size $N = 10000$, feature dimensions $p = q = 5$, the number of instruments level $J = 5$, and the label classes $K = 3$. Each data point in the dataset is a tuple $\{(X_i, U_i, Z_i, D_i, Y_i^*)\}_{i=1}^{N}$. The observed dataset consists of a tuple of variables $\{(X_i, Z_i, D_i, Y_i)\}_{i=1}^{N}$ with $Y_i := Y_i^\star$ if $D_i = 1$ and $Y_i := \text{NaN}$ if $D_i = 0$ for each $i \in [N]$.

**Model UC and NUCEM**  The missingness model NUCEM captures one of the setting of *No Unmeasured Common Effect Modifier*, and this ensures the satisfaction of sufficient condition in Proposition A.1, and therefore the conditional probabilities $\{p_k\}_{k=1}^{K}$ can be exactly identified according to Theorem 4.3. Meanwhile, the model UC captures a general setting of *Unmeasured Confounding* in which the point-identification condition cannot be realized in this situation.

**Baseline Methods and Our Proposed Methods**  We divide our dataset into two parts: a training set and a testing set, using a 7:3 split ratio, denoted as $\mathcal{S}_{\text{train}}$ and $\mathcal{S}_{\text{test}}$. In the training set, five different meta-learning methods are trained on selectively labeled data, using a simple neural network classifer with softmax output layer. We then evaluate the performance of these methods on the testing set with fully labeled data. The five methods under consideration are as follows:

- The first two methods act as baselines. The *SelectedSample* method involves running multiclass classification algorithm solely on the labeled portion of the training set $\{i \in S_{\text{train}} : D_i = 1\}$. For the second method, *SelectedSample(IPW)*, we firstly estimate the inverse propensity weighting (IPW) for the labeled subset, and then implement the weighted multiclass classification for the labeled training set. These baseline methods establish a "lower bound" for any other meta learning algorithm, as any enhanced method with the use of selectively labeled data should least beat the performance of these two methods.

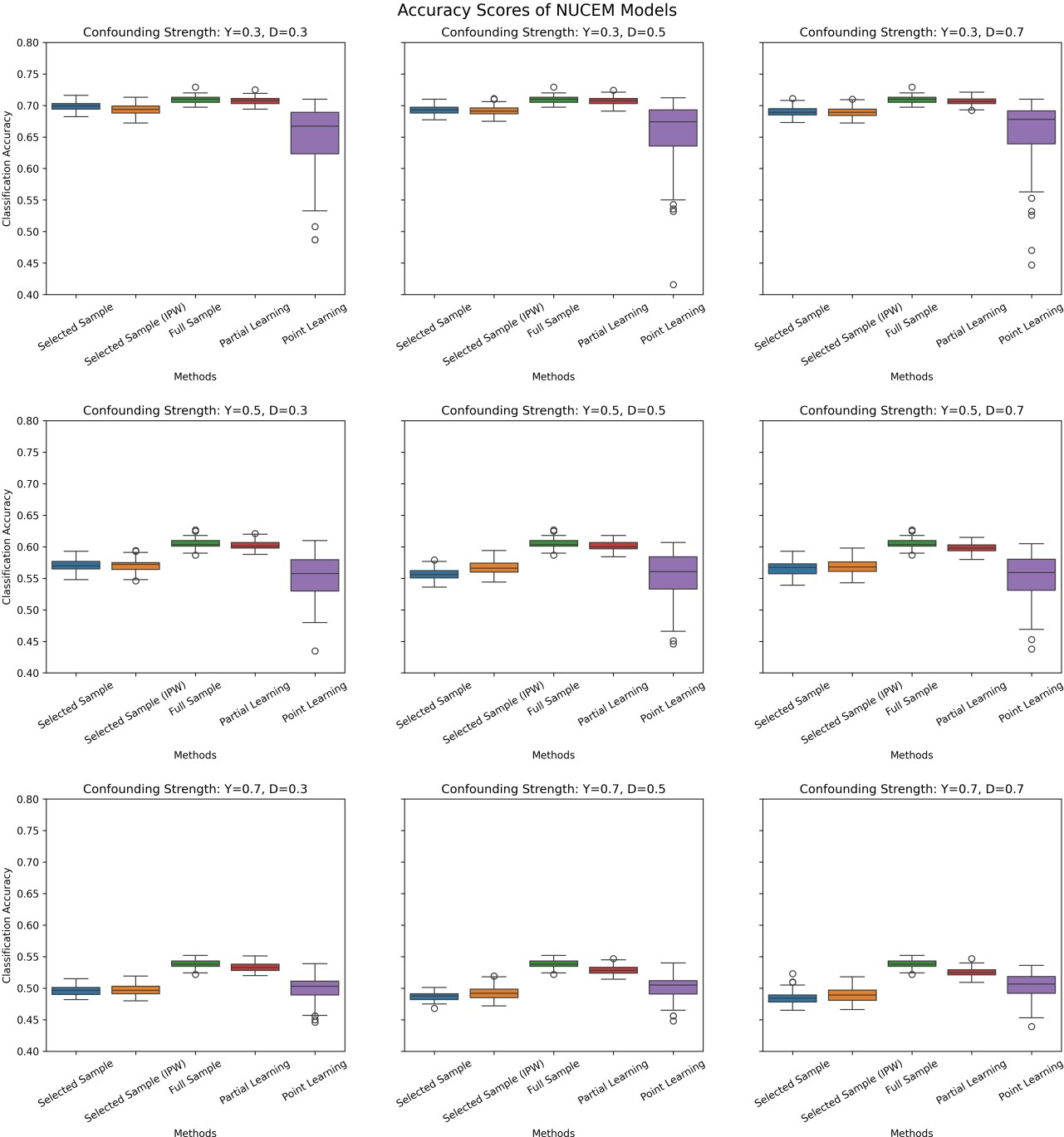

*Figure 4.* The testing accuracy of different methods with $\alpha_Y \in \{0.3, 0.5, 0.7\}$ and $\alpha_D \in \{0.3, 0.5, 0.7\}$ of Model NUCEM in synthetic dataset.

- The third method, *FullSample*, runs a multiclass classification algorithm on the entire training set. This represents an ideal but impractical scenario since we cannot actually observe the true label $Y_i^*$ for the missing data (where $D_i = 0$). This method serves as an 'upper bound' for learning performance, indicating the highest possible effectiveness if full information were available.

- Finally, our two proposed methods correspond to the point- and partial- identification settings, named *PointLearning* and *PartialLearning* respectively. We anticipate that the performance of these methods will fall between the established "lower" and "upper" bounds, representing a realistic estimation of effectiveness under selective label conditions. Similar to the method *SelectedSample(IPW)*, our method involve estimating a series of weight functions $w_k, k = 1, \ldots, K$ to correct for selection bias (refers to the details in Section 6). In this experiment, we estimate these weght functions using *Histogram Gradient Boosting* with a 5-fold cross-fitted approach. We selected all hyperparameter through 5-fold cross-validation. The cost-sensitive classification problem is solved by a simple neural network with a softmax output layer.

**The Role of Observed Variable Z**   It's important to note the random assignment of decision-maker, denoted as $Z$, play different roles in these methods. In our proposed method, $Z$ is treated as an instrumental variable, aiding in correcting for selection bias, while in the baseline methods (*SelectedSample*, *SelectedSample(IPW)*, and *FullSample*), $Z$ is only one of the observed features.

**Experiment Results**   Figures 4 and 5 report the testing accuracy of each method on the data generated by the NUCEM and UC models in Equation (43) respectively. All the experiments are repeated 100 times with confounding strengths $\alpha_D, \alpha_Y \in \{0.3, 0.5, 0.7\}$. Each boxplot then shows the distribution of the testing accuracy for distinct methods under different combinations of confounding strength $\alpha_D$ and $\alpha_Y$.

Transition from the top to the bottom panel, $\alpha_Y$ increases from 0.3 to 0.7, indicating a stronger influence of the unobservable variable $U$ on the true label $Y^\star$. As we can see from Figures 4 and 5, the accuracy of each of five methods decreases as $\alpha_Y$ increases, reflecting the growing challenge in accurately predicting outcomes as the influence of unobservable factors intensifies. Moreover, transition from the left to the right panel, $\alpha_D$ rises from 0.3 to 0.7, indicating an enhanced role of the unobservable variable $U$ in determining the missingness decision $D$. A larger value of $\alpha_D$ within the range $(0, 1)$ suggests that the missingness decision $D$ is predominantly influenced by the unobservable $U$, hinting at a potential larger difference of the distribution among selected group and missing group. This is visually corroborated in Figures 4 and 5, where the performance disparity between the *SelectedSample* and *FullSample* methods widens as $\alpha_D$ increases, illustrating the growing challenge in correcting the selection bias.

The results in Figures 4 and 5 demonstrate the effectiveness of our proposed methods in handling selection bias in the data generating process. In each combinations of $(\alpha_Y, \alpha_D)$, our *PartialLearning* method consistently outperforms the baseline methods, including the *SelectedSample* and *SelectedSample(IPW)* methods. Moreover, the perforamnce of *PartialLearning* is close to the *FullSample* method, which is the upper bound of the performance. The performance of the *PointLearning* method is also competitive, although it is slightly less robust than the *PartialLearning* method. The results in Figure 4 are consistent with those in Figure 5, except that under this setting, the point identification requirement in Theorem 4.3 is not satisfied, and the *PointLearning* method is therefore less effective than the *PartialLearning* method.

Below we provide a detailed analysis of the performance of each method.

- *SelectedSample(IPW)*: As we see from Figures 4 and 5, the performance of the *SelectiveSample(IPW)* approach is close to the baseline of naively running multiclass classification algorithms on the selected subsample (named *SelectedSample*), and it is outperformed by our proposed algorithms. This is not surprising since the propensity score $P(D = 1 \mid X, Z)$ cannot correct selection bias due to the existence of unobserved variables $U$.

- *PointLearning*: Our proposed *PointLearning* method shows promising results in the bottom right panel of Figure 4 with parameter $(\alpha_Y, \alpha_D) = (0.7, 0.7)$, in which case the unobserved confounding strength is large. Although the NUCEM model in Equation (43) satisfies Proposition A.1, which enables the point identification of the conditional probabilities and therefore the classification risk $\mathcal{R}(h, w)$, *PointLearning* does not outperform the baseline methods when the unmeasured confounding strength is not so large. This limitation is attributed to the complexities involved in accurately estimating the weight function $\{w_k\}_{k=1}^K$ as outlined in Equation (6).

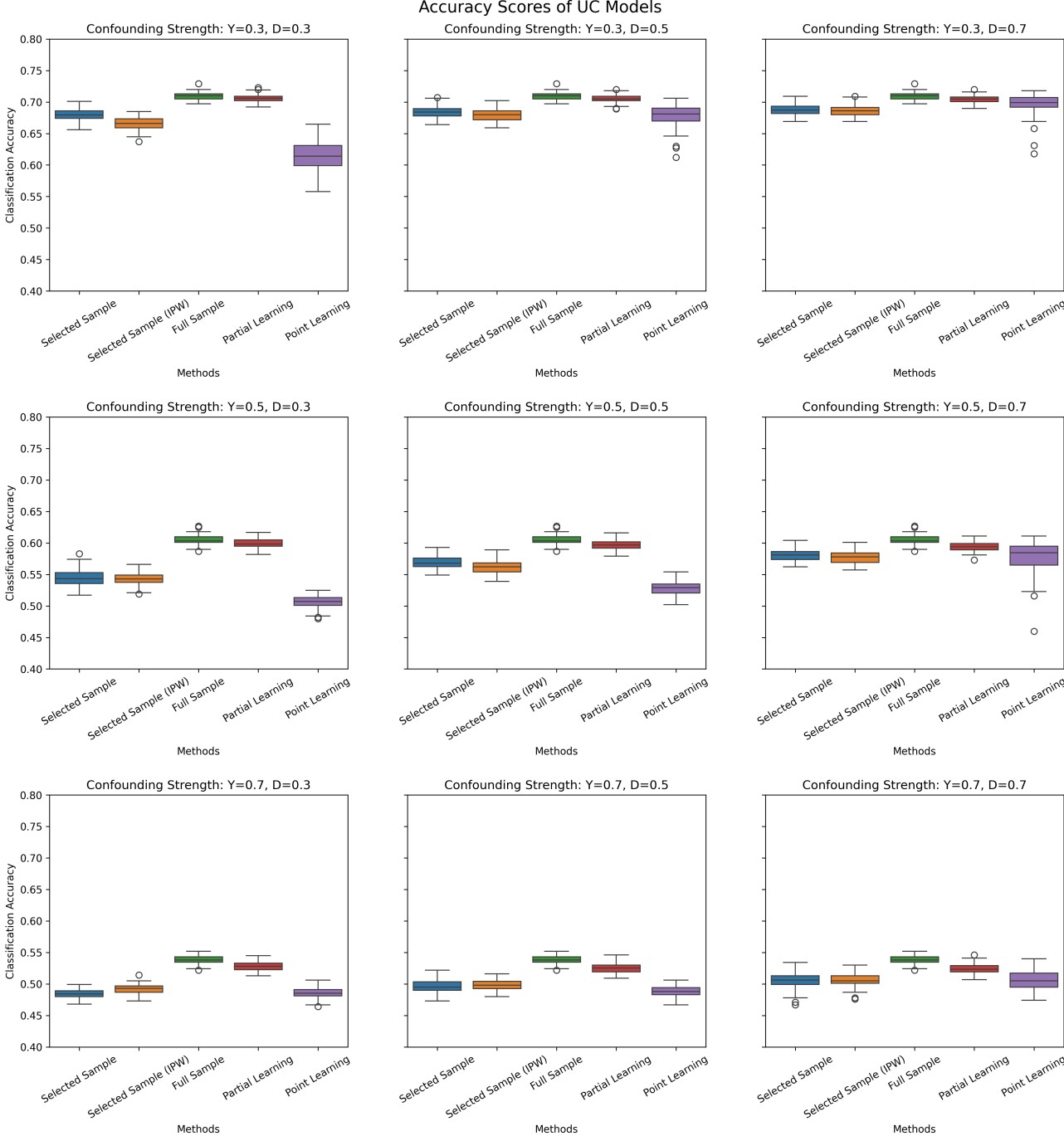

*Figure 5.* The testing accuracy of different methods with $\alpha_Y \in \{0.3, 0.5, 0.7\}$ and $\alpha_D \in \{0.3, 0.5, 0.7\}$ under Model UC of synthetic dataset.

The crux of the challenge lies in estimating the ratio of two conditional covariances, $\mathrm{Cov}(DY_k, Z \mid X)$ and $\mathrm{Cov}(D, Z \mid X)$ for each $k \in [K]$. Estimating conditional covariances is inherently more difficult than estimating conditional expectations, primarily because the process of calculating the ratio of these estimates is fraught with potential inaccuracies. If the denominator is not precisely estimated, the resultant ratio can become unstable, leading to exaggerated or diminished values. Such inaccuracies can severely impact the effectiveness of the downstream cost-sensitive classification, undermining the robustness of *Point Learning* under these conditions.

- *PartialLearning*: The cornerstone of our learning framework is *PartialLearning*. As evidenced by the graphs in Figures 4 and 5, *PartialLearning* significantly enhances the accuracy of performance predictions while maintaining low variance, showcasing its potential for real-world applications. Remarkably, this improvement in prediction accuracy is consistent across almost all combinations of confounding strengths $(\alpha_Y, \alpha_D)$, as well as the decision models NUCEM and UC.

  Reflecting on the conditions outlined in Theorem 5.2, *PartialLearning* only requires the conditional mean function $\mathbb{E}[Y_k^\star \mid U, X = x], k = 1, \ldots, K$ to be almost surely bounded, aside from the basic instrumental variable (IV) conditions. This requirement is generally more feasible in real-world scenarios than the point-identification assumptions detailed in Theorem (referenced as Theorem 4.3). The success of *PartialLearning* can be attributed to two key strategies: Firstly, we adopt a **robust optimization** approach to develop a prediction rule that remains effective under the least favorable conditions of $\eta^\star(X)$, as detailed in our minimax risk function formulation (see Equation (10)). Secondly, the weighting functions $\{w_k\}_{k=1}^K$ are based on the summation of several conditional expectations, which is inherently more stable than the ratio of conditional covariances required for point identification.

Overall, the compelling performance of *PartialLearning* demonstrated in Figures 4 and 5 convinces us of its practical viability. It stands as a reliable method that consistently outperforms baseline approaches, even in situations where the presence of selection bias (missing not at random) in the data generating process is uncertain.

### E.2. Sensitivity Analysis of Weight Function Estimation

The accuracy of weight estimation can directly impact the performance of our method. In fact, our learning guarantee in Appendix D already captures this. The excess risk bound in Equation (42) involves a term capturing the weight estimation error. Notably, the weights to be estimated are different for the point and partial identification settings. Estimating the weight for point identification involves estimating two nuisances and their ratio. In contrast, the weight in the partial identification only involves the sum of a series of nuisance functions and does not involve any ratio. The latter is generally more insensitive to the nuisance estimation error.

To illustrate the impact of weight estimation accuracy, we implement additional experiments using synthetic data with confounding strength $\alpha_D = 0.5$ and $\alpha_Y = 0.7$ on both the NUCEM and UC settings (see details in Equation (43)). To introduce controlled errors into the nuisance function estimates, we inject Gaussian noise as follows:

$$\tilde{\eta}(X_i) = \hat{\eta}(X_i)[1 + \sigma^2 \mathcal{N}(0, 1)].$$

We vary the noise level $\sigma$ across the values $[0.0, 3.0, 5.0, 7.0]$ and repeat each experiment 50 times. Based on the perturbed nuisance estimates—specifically, the conditional probability $\eta_k^\star(x)$ in the point identification setting, and the bounds $l_k(x)$ and $u_k(x)$ in the partial identification setting—we compute corresponding weights and analyze their influence on downstream classification accuracy.

Our results in Table 1 show that inaccuracies in weight estimation degrade classification performance for both *PointLearning* and *PartialLearning*. Nevertheless, the performance of both methods remains stable under small perturbations ($\sigma < 0.3$) in both the NUCEM and UC settings. Notably, *PartialLearning* achieves higher accuracy across most of noise levels, demonstrating greater robustness to errors in nuisance estimation.

### E.3. Semi-synthetic Dataset: FICO

In this section, we evaluate the performance of our proposed algorithm in a semi-synthetic experiment based on the home loans dataset from (FICO, 2018). This dataset consists of 10459 observations of approved home loan applications. The dataset records whether the applicant repays the loan within 90 days overdue, which we view as the true label $Y^\star \in \{0, 1\}$, and various transaction information of the bank account. The dataset also includes a variable called ExternalRisk, which is a

| Settings | Methods | $\sigma = 0.0$ | $\sigma = 0.3$ | $\sigma = 0.5$ | $\sigma = 0.7$ |
|---|---|---|---|---|---|
| NUCEM | Point Learning | 0.601 (0.0124) | 0.592 (0.0161) | 0.586 (0.0212) | 0.534 (0.0381) |
| | Partial Learning | 0.607 (0.0121) | 0.592 (0.0196) | 0.564 (0.0277) | 0.549 (0.0343) |
| UC | Point Learning | 0.553 (0.0241) | 0.561 (0.0232) | 0.559 (0.0235) | 0.439 (0.0604) |
| | Partial Learning | 0.602 (0.0136) | 0.586 (0.0188) | 0.559 (0.0300) | 0.533 (0.0370) |

*Table 1.* The testing accuracy of point learning and partial learning under NUCEM and UC settings with different noise level $\sigma = [0.0, 0.3, 0.5, 0.7]$.

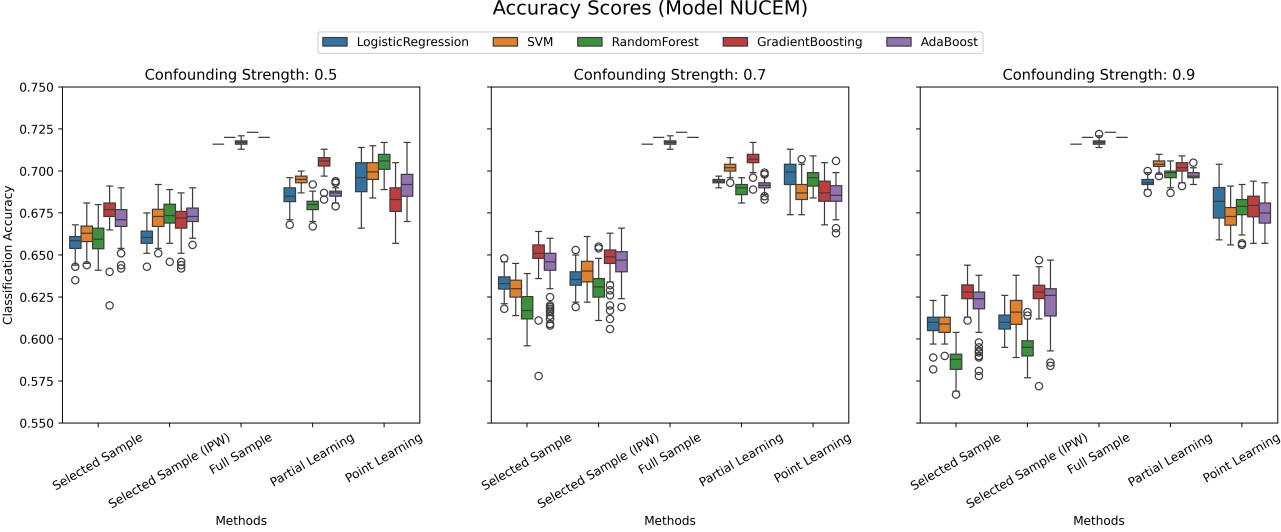

*Figure 6.* The testing accuracy of methods with $\alpha \in \{0.5, 0.7, 0.9\}$ for model NUCEM in FICO dataset.

risk score assigned to each application by a proprietary algorithm. We consider ExternalRisk and all transaction features as the observed features $X$.

**Semi-synthetic Dataset with Selective Labels**  In this dataset the label of interest is fully observed, so we choose to synthetically create selective labels on top of the dataset. Specifically, we simulate 10 decision-makers (e.g., bank officers who handle the loan applications) and randomly assign one to each case. We simulate the decision $D$ from a Bernoulli distribution with a success rate $p_D$ that depends on an "unobservable" variable $U$, the decision-maker identity $Z$, and the ExternalRisk variable (which serves as an algorithmic assistance to human decision-making). We blind the true label $Y^\star$ for observations with $D = 0$. Specifically, we construct $U$ as the residual from a random forest regression of $Y^\star$ with respect to $X$ over the whole dataset, which is naturally dependent with $Y^\star$. We then specify $p_D := \mathbb{P}(D = 1 \mid X, U)$ according to

$$\begin{aligned}
\text{Decision (NUCEM):} \quad p_D &= \alpha \cdot \text{expit}\{U\} + (1 - \alpha) \cdot \text{expit}\{(1 + Z) \cdot \text{ExternalRisk}\}, \\
\text{(UC):} \quad p_D &= \text{expit}\{\alpha \cdot U + (1 - \alpha) \cdot (1 + Z) \cdot \text{ExternalRisk}\}.
\end{aligned} \tag{44}$$

Here the $\text{expit}$ function is given by $\text{expit}(t) = 1/(1 + \exp(-t))$. The parameter $\alpha_D \in (0, 1)$ controls the impact of $U$ on the labeling process and thus the degree of selection bias. We can easily verify that the sufficient condition in Proposition A.1 is guaranteed and therefore the point-identification of classification risk $\mathcal{R}(h, w)$ is realizable under the NUCEM model in Equation (44).

**Baseline Methods and Our Proposed Methods**  We randomly split our data into training and and testing sets at a $7 : 3$ ratio. Similar to what we discuss in Appendix E.1, on the training set, we apply five types of different methods: *Selected Sample*, *SelectedSample(IPW)*, *FullSample*, *PointLearning* and *PartialLearning*. The first three methods serve as the benchmark of the last two methods. For each type of method, we try multiple classification algorithms including AdaBoost, Gradient

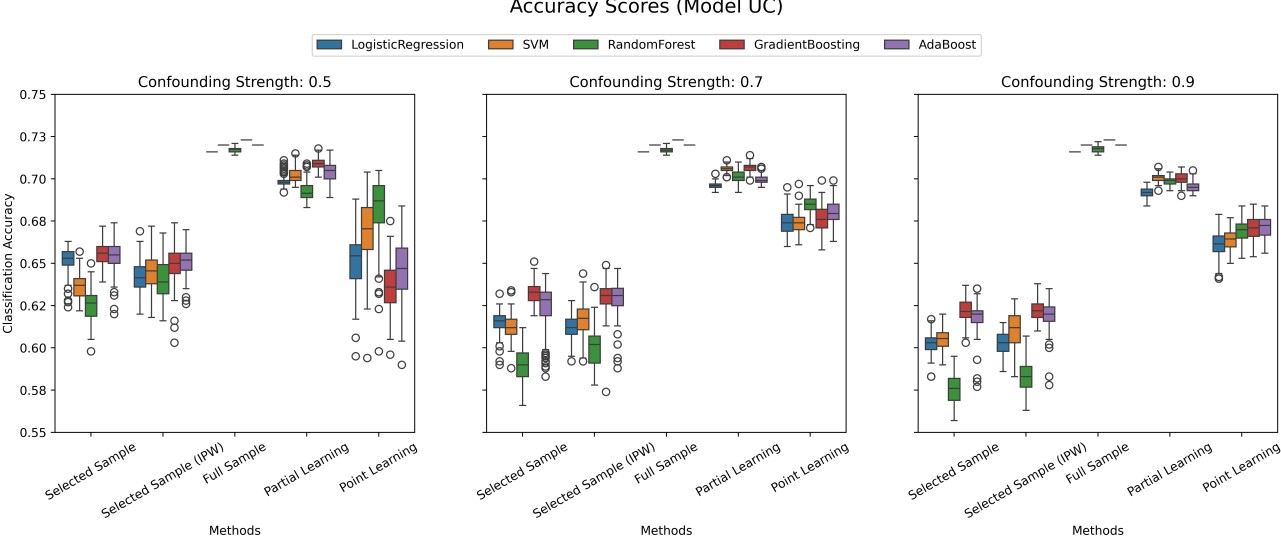

*Figure 7.* The testing accuracy of methods with $\alpha \in \{0.5, 0.7, 0.9\}$ for model UC in FICO dataset.

Boosting, Logistic Regression, Random Forest, and SVM. Our proposed method also need to estimate some unknown weight functions, which we implement by $K = 5$ fold cross-fitted Gradient Boosting. Again, all hyperparameters are chosen via 5-fold cross-validation, and we evaluate the classification accuracy of the resulting classifiers on the testing data.

Similarly, we remark that the decision-maker assignment $Z$ plays different roles in different methods. Our proposed methods (*PointLearning* and *PartialLearning*) treatment the decision-maker assignment $Z$ as an instrumental variable to correct for selection bias. In contrast, the baseline methods (*SelectedSample*, *SelectiveSample(IPW)* and *FullSample*) do not necessarily need $Z$. However, for a fair comparison between our proposals and the baselines, we still incorporate $Z$ as a classification feature in the baseline methods, so they also use the information of $Z$.

**Results and Discussions**    Figures 6 and 7 presents the testing accuracy of each method over 50 experiment replications for $\alpha \in \{0.5, 0.7, 0.9\}$ under both the NUCEM and UC models defined in Equation (44). First, we observe that the performance of *SelectedSample(IPW)* is comparable to the baseline *SelectedSample*, which applies binary classification algorithms directly to selectively labeled data. Notably, as the strength of unmeasured confounding ($\alpha_D$) increases, the gains from using our proposed methods—especially *PartialLearning*—over the baseline methods (*SelectedSample* and *SelectedSample(IPW)*) become more pronounced. Interestingly, the *PartialLearning* method outperforms even under the NUCEM model, where the point-identification condition is satisfied. As discussed in Appendix E.1, this may be because the *PointLearning* method relies on estimating a conditional variance ratio, which is challenging to estimate accurately in practice. In contrast, the *PartialLearning* method requires only the estimation of conditional expectations, making it more stable and robust. This highlights the advantage of the *PartialLearning* method: it is robust to violations of the point-identification assumption and achieves more stable performance even when point-identification holds.

To summarize, among all the methods we evaluated, *PartialLearning* emerges as a reliable approach that consistently outperforms baseline methods, even in scenarios where the unobsreved confounding is strong.

