# OpenReview forum: "Learning with Selectively Labeled Data from Multiple Decision-makers"
_ICML.cc/2025/Conference — ICML 2025 poster_

### Official Review · Reviewer_aAyP · 2025-03-08

**Overall Recommendation:** 3

**Summary:**

The paper tackles the problem of correctly quantifying classification risk in the selectively labeled data setting. In this setting, true outcomes are only observed for samples that receive a certain classification/decision (e.g., default outcomes are only observed for those who are given the loan). To address this selection bias in observed outcome data, the paper proposes a framework that utilizes decision-maker pool heterogeneity to identify the true classification risk, under the assumption that the decision-maker is selected randomly for each sample.

The authors first characterize the requirements of the data-generating process that are necessary to exactly identify the classification risk in this setting. They note that these requirements are overly strong for real-world applications and discuss relaxations under which partial identification is possible. With a cost-sensitive learning approach, they train classifiers using the identified (exact or partial) risk and demonstrate the efficacy of a loan application dataset with synthetic decision-makers.

**Claims And Evidence:**

The main contribution of the paper is the framework to identify classification risk using decision-maker assignment as an IV. The authors claim that, in many real-world applications, decision-making usually involves several decision-makers and the choice of decision-maker for any unlabeled sample is often random. While this methodology is creative, I don’t think the paper does a satisfying job of justifying this claim either using prior work or empirical data. I note two concerns below.

1. In terms of prior work, there is very limited discussion in the introduction to support this claim. The authors cite prior works that consider decision-maker heterogeneity in the judicial setting and discuss them in Appendix A. However, there’s no discussion of whether there's prior evidence for random decision-maker assignment and decision-maker heterogeneity in other domains (including the loan application domain that features prominently across the paper).

2. For empirical analysis, the paper chooses to create synthetic decision-makers, which again makes me question the validity of random (or quasi-random) decision-maker assignments in real-world settings or the availability of data on who made which decision.

**Essential References Not Discussed:**

The paper notes the main related works in this field. However, it's strange to have the entire related works section in the Appendix. Several citations discussed in the "Selective Labels Problem" setting are crucial for motivating the issues associated with selective label problems in real-world applications and would be helpful in contextualizing the problem setting for readers who are unfamiliar with this domain.

I strongly suggest having at least a short related work section in the main body to include a discussion on the primary references.

**Experimental Designs Or Analyses:**

The experimental analysis presented seems mostly sound, although it would be good to provide comparisons against other related works (e.g., Rambachan et al) if possible.

One point that I think needs more discussion in the empirical analysis is how the $a_k, b_k$ (and correspondingly $l_k, u_k$) are specified/learned in the partial learning experiments. Currently, I don't see any discussion on these parameters in Section 6 (let me know if its there and I am missing something) and considering its relevance to the partial identification setting it will be good to discuss how they are being set for the experiments.

**Methods And Evaluation Criteria:**

1. Regarding the methods employed to address the selectively labeled data problem, I like the broad idea the paper utilizes of employing an IV strategy to address selection bias using external factors that control the decision. However, I am not completely convinced that decision-maker assignment offers the best IV pathway, mainly because the paper shows that exact identification requires overly strict assumptions. In light of this restriction, I am curious about the realistic settings (and not just the simple ones noted on Page 4) where the authors think this method can still provide accurate estimates of classification risk.

2. For empirical analysis, the paper focuses on a loan-approval dataset, computing classification risk using the proposed framework in both exact and partial identification settings. Overall, it does seem to lead to more accurate classifiers than simple baselines. I am curious as to why the paper doesn’t compare their method to Rambachan et al. (even if limited to comparisons of risk evaluation using different methods) since they note that both papers consider the IV strategy to tackle the selectively labeled data problem.

3. Additionally, considering that the empirical analysis is based on semi-synthetic data, it would be good to acknowledge the limitations of this analysis somewhere in the paper.

**Other Comments Or Suggestions:**

None

**Other Strengths And Weaknesses:**

The use of decision-maker assignment as an IV is creative and I wonder if it can also be considered as a potential intervention to ensure classification risk identification for future data collection procedures.

**Questions For Authors:**

1. Are there prior works in multiple domains that provide evidence for the fact that decision-maker assignment is mostly random and that the decision-maker pool is heterogeneous?

2. Is there real-world data available in any domain where decision-maker assignments are noted and where the proposed framework can be employed?

3. Is it feasible to empirically compare the proposed framework to prior works (e.g., Rambachan et al) and, if so, does it achieve similar/better performance than prior works?

4. How are parameters $(a_k, b_k)$ and $(l_k, u_k)$ specified/learned in the partial learning experiments?

**Relation To Broader Scientific Literature:**

Overall, the paper definitely tackles an important problem of identification of prediction risk when the available outcome data is selectively labeled. The IV methodology of harnessing decision-maker heterogeneity is interesting and fits well (theoretically) with the decision-making setup of important real-world applications.

However, as I noted above, there are doubts about the feasibility and practicality of the proposed framework which I believe severely limits the impact of the framework.

**Theoretical Claims:**

The theoretical claims made in the paper seem correct.

---

> ### Author Rebuttal · Authors · 2025-04-01
>
> We appreciate your valuable feedback. We now address each of your comments below.
>
> - Evidence of (Quasi-)randomly Assigned Heterogeneous Decision-Makers: The selective label literature has mainly considered two examples for randomly assigned heterogeneous decision-makers:
> 1. Judicial Decision-Making – In US, Judges are often randomly assigned to cases, yet they exhibit heterogeneous decision-making styles and biases (Kleinberg et al., 2018; Marcondes et al., 2019). This example is widely cited in the selective label literature.
> 2. Loan Approval – In fact, loan applications are also randomly assigned to different loan officers in some places. For example, there even exist platforms, such as OppFi (https://ortooapps.com/case-study/oppfi), that implements the random assignment to ensure fairness and equity. Even if the loan applications are not randomly assigned, they may be viewed as quasi-random as long as the loan officers roughly face the same distribution of applications given the observed features. Moreover, Bushman et al. (2021) demonstrate that individual loan officers significantly influence loan contract terms and performance, highlighting the heterogeneity across decision-makers.
>
> - Real-world Data: Kleinberg et al. (2018) indeed uses real-world data on all arrests made in New York City between 2008 and 2013, but the data are not publicly available. Existing selective label literature also considered a few other datasets. But unfortunately these are also not public. This is why we use synthetic or semi-synthetic data in our paper. However, we hope to clarify that synthetic data actually have advantages over real-world selective label data. This is because in the real data, the outcomes of the unlabeled units are unobservable so evaluating the results from a given algorithm is not straightforward. By using synthetic or semi-synthetic data, we do observe the true outcomes for all units so we can more easily assess different algorithms.
>
> - Decision-maker Assignment as IV: we understand your concern about the point identification assumption and your questioning the validity of view decision-maker assignment as IV.
> We hope to clarify that the key defining characteristics of IV are exogeneity and exclusion restriction. These two are satisfied by the decision-maker assignment under our assumptions, and many existing selective label works also leverage these two properties in their heuristic solutions. In contrast, the homogeneity assumption (NUCEM assumption) is not the defining characteristic of IV. There exist many works on IV that does not require this assumption, such as the local average treatment effect (LATE) analysis in Angrist et al. (1996) and the Balk-Pearl partial identification bounds.
> Moreover, one purpose of our point identification is exactly to reveal that it needs strong assumption, which motivates our IV partial identification. In experiments, we do observe that the partial identification approach tends to perform better.
>
> - Comparison with Rambachan et al. (2023): The focus of Rambachan et al. (2023) differs fundamentally from ours. Their work primarily studies evaluating  various error measures  of a given binary classifier based on IV-based partial bounds, whereas we focus on learning a robust classification rule under both point and partial identification settings. We note that it is not clear how to optimize their error measure estimators to effectively train robust classifiers. In contrast, our work develops a unified cost-sensitive learning (UCL) algorithm for both point and partial identification settings. This requires in-depth analyses of the minimax formulation and cost-sensitive learning problems.  So their approach is not directly comparable to ours. We will further clarify this in our revision.
>
> - Specification of Range $a_k$ and $b_k$: In our experiments, we set $a_k(x) = 0$ and $b_k(x) = 1$ for all cases. The partial bounds $l_k(x)$ and $u_k(x)$ are then computed using the estimated nuisance functions along with the specified values of $a_k(x)$ and $b_k(x)$. These nuisance functions are learned from the observed data using the Gradient Boosting algorithm. We will clarify this point in future revisions.

---

> > ### Comment · Reviewer_aAyP · 2025-04-04
> >
> > Thanks to the authors for your responses. The clarification on why DM assignment can serve as a good IV does help better understand the appeal of this approach.
> >
> > I am still not completely convinced of the real-world feasibility since, as the authors note, real-world evaluation is difficult as DM assignments are usually not public. However considering the theoretical and empirical advantages, there might be potential for future work to address the question of feasibility in real-world settings. As such, I am increasing my score to reflect that and would recommend including a robust discussion in the paper on the limitations of the proposed approach.

---

> > > ### Author Response · Authors · 2025-04-07
> > >
> > > Thank you for your feedback and for increasing the score. We sincerely value your engagement in the review process. In our revision, we will follow your suggestion, discussing the challenges with real-world data evaluations and acknowledging the potential limitations of our experiments with synthetic and semi-synthetic data. Thank you again for your time and consideration.

---

### Official Review · Reviewer_Ye3n · 2025-03-12

**Overall Recommendation:** 2

**Summary:**

This paper studies multiclass classification with selectively labeled data, where label distribution is biased due to historical decision-making. By leveraging variations in decision rules across multiple decision-makers, the authors apply an instrumental variable (IV) framework to establish necessary and sufficient conditions for exact classification risk identification. When exact identification is infeasible, they derive sharp partial risk bounds. To mitigate label selection bias, the paper proposes a unified cost-sensitive learning (UCL) approach.

**Claims And Evidence:**

Yes, the claims made in the submission supported by evidence.

**Essential References Not Discussed:**

No.

**Experimental Designs Or Analyses:**

Yes, it provides a semi-synthetic example.

**Methods And Evaluation Criteria:**

Yes, methods and valuation criteria make sense for the problem or application at hand.

**Other Comments Or Suggestions:**

No.

**Other Strengths And Weaknesses:**

The paper primarily builds on existing theoretical frameworks for instrumental variable (IV) methods and cost-sensitive classification, with its results being a special case of prior work when the outcome is discrete. While it provides a structured application of these ideas to multiclass classification with selectively labeled data, the theoretical contributions do not introduce fundamentally new insights, as they closely align with well-established results in the literature.

To enhance its novelty, the paper could explore deeper theoretical results, such as deriving the efficiency bound for the risk function under exact identification.

**Questions For Authors:**

I wonder why the paper focuses only on discrete outcomes. Can the approach be easily extended to continuous variables? If so, why not establish a more general framework for decision-making that encompasses both cases?

**Relation To Broader Scientific Literature:**

The contributions of this paper build upon existing work in instrumental variable (IV) methods and cost-sensitive classification. The theoretical foundations for exact identification under IV have been established by Cui and Tchetgen Tchetgen (2021), while partial identification under IV has been explored by Pu and Zhang (2021). The results in this paper can be seen as a special case of these prior works when the outcome is discrete. Additionally, cost-sensitive learning approaches, particularly in the context of contextual bandits, have been well studied, as highlighted by Bietti et al. (2021). While the paper applies these concepts to multiclass classification with selectively labeled data, its theoretical contributions largely align with existing literature rather than introducing fundamentally new insights.

**Theoretical Claims:**

I did not thoroughly check each proof. However, most of the theoretical claims are not novel and have already been well-established in the literature. Additionally, they do not contradict my prior understanding.

**Exact identification under IV**: Cui, Y. and Tchetgen Tchetgen, E., 2021. A semiparametric instrumental variable approach to optimal treatment regimes under endogeneity. Journal of the American Statistical Association, 116(533), pp.162-173.

**Partial identification under IV**: Pu, H. and Zhang, B., 2021. Estimating optimal treatment rules with an instrumental variable: A partial identification learning approach. Journal of the Royal Statistical Society Series B: Statistical Methodology, 83(2), pp.318-345.

**Cost-sensitive classification**: Bietti, A., Agarwal, A., and Langford, J., 2021. A contextual bandit bake-off. Journal of Machine Learning Research, 22.133, pp.1-49.

---

> ### Author Rebuttal · Authors · 2025-04-01
>
> We sincerely appreciate your valuable feedback and your references. Our work indeed builds on these previous literature and is therefore closely related to them. However, our work is not “a special case of these prior works”. Instead, our work significantly generalizes these existing literature and makes contributions beyond them. Below we provide detailed explanations, which we will also clarify in our revised version.
>
> - The Exact Identification in Cui and Tchetgen (2021). They study policy learning under unmeasured confounding using IV, focusing on a binary IV and binary treatment. They aim to learn a confounding robust treatment rule that maps features to a binary treatment. This is more like a binary classification problem with causal structure. In contrast, our work considers a multi-valued IV and a multi-class label, aiming to learn a classification rule that maps features to a multi-class label. This setting strictly generalizes their framework.
> Notably, their identifying NUCEM assumption involves the difference of certain conditional expectations given the two different IV values. This heavily relies on the binary nature of IV. Instead, we consider a more general assumption that can tackle general IV $Z$. Moreover, their learning procedure only uses hinge loss but we explore more surrogate losses for both binary and multi-class classification.
>
> - The Partial Identification in Pu and Zhang (2021). They explore policy learning under unmeasured confounding with a binary-valued IV, binary treatment, and binary outcome, again learning a robust treatment rule that maps features to a binary treatment. In contrast, we consider multi-valued IVs and multi-class labels.
> Pu and Zhang (2021) directly uses Balke and Pearl’s partial bounds for the binary IV and binary outcome setting. We extend these bound to accommodate multi-valued IVs and any bounded multi-class outcomes (see Assumption 4.1). Moreover, we rigorously prove the tightness of our IV-based partial identification bounds (Appendix C.1), which to our knowledge is new to the literature. Pu and Zhang (2021) also considers minimax learning. Their inner maximization problem can be easily solved in closed-form for the binary outcome and their final minimization is based on hinge surrogate loss. However, we consider multi-class classification where the inner maximization involves much more complex simplex constraints. We managed to give a closed-form solution by carefully analyzing the problem structure.
>
> - Contextual bandits in Bietti et al. (2021). This paper conducts an empirical analysis of several algorithms for online contextual bandits, which studies a fundamentally different problem from ours. We acknowledge that cost-sensitive classification and surrogate losses are not new problem ideas. But we hope to clarify that our contribution lies in in-depth analyses that transform the problems in both point and partial identification settings into a unified cost-sensitive classification form. Under our general framework, we can explore a range of different surrogate losses while Cui and Tchetgen (2021) and Pu and Zhang (2021) focus on only the hinge loss.
>
> We now further respond to your other comments.
>
> - Efficiency bound under exact identification: We appreciate your suggestion but we think this is beyond the scope of our current paper. The main focus of this paper is to provide a unified learning framework for the selective label problem in both point and partial identification settings. As our work is dense already, we leave the efficiency bound for future study.
>
> - Extension from Discrete to Continuous Outcomes: Thanks for the great question. Extension from discrete to continuous outcomes is indeed an important problem but we think this should be left for a separate future study.
> Notably, all existing selective label literature focuses on binary outcomes (see references in Appendix A). Our study of multi-class outcomes already constitutes an extension. Moreover, as we discuss above, our work also strictly generalizes Cui and Tchetgen (2021) and Pu and Zhang (2021), instead of being their special cases with restricted outcomes. In our extensions of these prior literature, our paper overcomes many new technical challenges, generalizing many assumptions, analyses and results in these literature.
> Finally, we hope to briefly touch on the challenges with continuous outcome. Our partial identification analysis involves specifying some bounds $a_k(X)$ and $b_k(X)$ in Assumption 4.1. For classification problems, these can be naturally set as 0 and 1. But for continuous outcome, we may need to specify the range for $E[Y^* | X, U]$, which may has no natural ranges. Moreover, in solving the inner maximization problem, we heavily rely on the simplex structure of the constraints. But such structure no longer applies to continuous outcomes.

---

### Official Review · Reviewer_as5X · 2025-03-12

**Overall Recommendation:** 4

**Summary:**

This paper focuses on the problem setting of classification with selective labeled data, that is, the labeled data at hand can be biased because of decision-making in the past. This paper defines the problem mathematically and solves this problem from the perspective of the instrumental variable (IV) framework. There are two assumption settings: (1) No unmeasured common effect modifiers (NUCEM), which is a strong assumption that leads to a clean solution, and (2) Partial identification, where a reasonable solution can be obtained. Theoretical analyses of two assumption settings are provided. Furthermore, a practical algorithm for both cases is also provided based on weighted empirical risk minimization with calibration guarantee. Synthetic experiments show that the proposed method outperforms baselines.

## update after rebuttal
After the rebuttal, I still think the idea of this paper is novel. It studies the problem setting extensively theoretically and also provides experimental results., Thus, I keep my score (4: accept). The authors clarified in the rebuttal that their work has novelty and also admits some current drawbacks of their methods (e.g., computation time).

**Claims And Evidence:**

1. Strong theoretical results for a problem setting that is quite complicated and has practical relevance.
2. Experimental results show effectiveness of the proposed practical algorithm.

**Essential References Not Discussed:**

No additional requests from me.

**Experimental Designs Or Analyses:**

Experimental designs and analyses are valid.

**Methods And Evaluation Criteria:**

Since the problem setting is quite complicated, I believe there does not exist a benchmark dataset that directly corresponds to this problem. This justifies the use of synthetic datasets that this paper decided to do.

**Other Comments Or Suggestions:**

The paper is quite dense already, but it would be better to explain more about related work in the main body if possible to highlight the novelty of the proposed work as well as reviewing prior work to the reader.

Line 110 (left): if8 -> if
Line 1523: conseuqnece  -> consequence
Line 1995: Vairable -> Variable

**Other Strengths And Weaknesses:**

Strengths
1. Strong theoretical results that improve an understanding of a complicated yet relevant problem setting. It is praiseworthy that this paper not only focuses on a restrictive NUCEM assumption but also considers the partial information assumption.
2. Practical algorithms with theoretical guarantee are provided, which can be relatively easy to implement.
3. Experimental results (although synthetic) show that the proposed method is effective compared with reasonable baselines.

Weaknesses
1. Proposed method's weakness is not much discussed in my understanding. One might be that it could be computationally expensive (I'm not sure). Moreover, the estimation of weight could be incorrect, and we don't see much effect in the experiments, whether this can make the proposed method not work well. I find the comment in the paper, why NUCEM lost to partial under NUCEM assumption quite interesting that NUCEM requires a ratio estimation. I think such discussions could be useful. Or ablation study of the effect of imprecise weight estimation could also be useful. (but I'm also aware that the paper is already dense unfortunately).

**Questions For Authors:**

1. Could you please comment on the comparison of the computational cost of the proposed unified cost-sensitive learning (point), (partial), and vanilla training?
2. Since many weights have to be estimated, how important is the accuracy of weight estimation? Is the solution highly sensitive to this?
3. Is this the first work to use instrumental variable (IV) framework for selective labeled classification?

**Relation To Broader Scientific Literature:**

The paper is related to selective labels problems, where it has been discussed in appendix. It might be also similar to weakly supervised learning or domain adaptation in the sense that the observed labeled dataset has something different from the test distribution and we have to use information at hand somehow to derive a risk minimizer for the test distribution using observed training information.

**Theoretical Claims:**

The proposed method looks reasonable to me. The theoretical claims are sound in my understanding.

---

> ### Author Rebuttal · Authors · 2025-04-01
>
> We sincerely thank you for your insightful comments and positive feedback for our work.
>
> - Computational Cost: The computational cost of our method is higher than vanilla method which directly learn a classifier from the observed (selectively labeled) data. That is because our method consists of two steps: The first step is estimation of nuisance functions through a cross-fitting technique, which are used to construct the classification weights; The second step is a weighted / cost-sensitive classification procedure, which can be efficiently solved. The nuisance estimation indeed entails some additional computational costs. But this typically only involves a series of standard regression/classification fitting. This is usually manageable and also widely adopted in the causal inference literature (Chernozhukov et al., 2018). Moreover, this can be accelerated by fitting the nuisances on different folds of data in parallel. We will clarify the computational aspect in our revision.
>
> - Weight Estimation Accuracy: The accuracy of weight estimation can directly impact the performance of our method. In fact, our learning guarantee in Appendix E already captures this. Our excess risk bound in that part involves a term capture the weight estimation error.  Notably, the weights to be estimated are different for the point and partial identification settings. Estimating the weight for point identification involves estimating two nuisances and their ratio. In contrast, the weight in the partial identification only involves the sum of a series of nuisance functions and does not involve any ratio. The latter is generally more insensitive to the nuisance estimation error.
> During the rebuttal period, we conducted some additional experiments for the simulation setting with confounding strength $\alpha_Y=0.5$ and $\alpha_D = 0.7$ (see setting details in Appendix F) to assess the impact of nuisance estimation error.
> We introduced Gaussian noise into the estimates of nuisance functions to inflate their errors, defined as $\tilde{\eta}(X_i) = \hat{\eta}(X_i) \cdot [1 + \sigma^2 \mathcal{N}(0,1)]$ with noise levels $ \sigma = \{0.0, 1.0, 2.0, 3.0, 4.0 \}$. The experiment is repeated for 10 times. We then computed weights based on these noisy estimates and analyzed their effect on resulting classification accuracy. Our findings reveal that both partial and point learning remain stable under small perturbations.
> However, beyond a certain noise threshold, performance degrades significantly. Notably, partial learning exhibits higher tolerance ($\sigma=4.0$) compared to point learning ($\sigma=2.0$). This demonstrates that the partial learning approach is indeed more resilient to nuisance estimation errors.
> In our revision, we will clarify the impact of nuisance estimation errors and add the extra numerical results.
>
> - First Work to Use IV for the Selective Labels Problem: Thank you for your suggestion on emphasizing the novelty of our work. Previous literature on the selective labels problem (SLP) also leverages the random assignments of heterogeneous decision-makers but their approaches are largely heuristic, as discussed in Appendix A. Our work uses the IV framework to provide principled point and partial identification analyses and derive rigorous learning algorithms.
> We remark that a closely related work by Rambachan et al. (2023) also considers IV-based partial identification in the context of selective label problem. However, our work is substantially different from theirs and makes many unique contributions. Please see our detailed responses to reviewer aAyP. While we already mentioned these in our paper, we will further highlight them in our revision.

---

> > ### Comment · Reviewer_as5X · 2025-04-04
> >
> > I appreciate the authors for providing detailed responses to my concerns and I believe there is no misunderstanding of my review. It is great that the theoretical analysis also covers the weight estimation error.
> >
> > I still have a positive impression of this paper, and I will maintain the same score during the rebuttal period. However, I also saw that another reviewer pointed out that the results in this paper are not novel. I still haven't confirmed it myself since it is still unclear which main results have already been proven in which parts of the several papers the reviewer suggested. If it can be verified that the main results found in this paper have already been discovered, I might decrease the score in the final review.

---

> > > ### Author Response · Authors · 2025-04-07
> > >
> > > Thank you for your positive impression of our work and for clearly articulating your current concern. We truly value your active engagement in the review process. We would like to further emphasize that the main results of our work are novel, making multiple contributions beyond the existing literature.
> > >
> > > 1. Our study is the first to systematically explore both the point and partial identification of selective label classification within a principled instrumental variable (IV) framework. We offer unified cost-sensitive formulations for both scenarios. This not only allows for the use of a variety of surrogate losses but also provides a unified framework for theoretical analysis. In contrast, the existing selective label literature either lacks formal identification analyses or only covers partial identification for classifier evaluation, without considering classifier optimization or learning (Rambachan et al., 2023; see response to Reviewer aAyP).
> > >
> > > 2. Although our point identification result is related to Cui and Tchetgen (2021), our work differs from theirs in several key aspects.
> > > Cui and Tchetgen (2021) focuses on learning the optimal treatment allocation rule with a binary treatment $A$ and a binary IV $Z$. Their point identification result relies on the homogeneity assumption that $P(A = 1 \mid Z = 1, X, U) - P(A = 1 \mid Z = 0, X, U)$ is conditionally uncorrelated with $E[Y(1) – Y(0) | X, U]$ given $X$, where $Y(1), Y(0)$ are the two potential outcomes that are observed when $A=1, 0$ respectively. Notably, this assumption heavily relies on the binary nature of IV $Z$. They then use an inverse propensity weighted formulation to transform the problem into a weighted classification problem, with the treatment or IV serving as the binary label. Then the misclassification zero-one loss is replaced by the hinge loss for optimization.
> > > In contrast, our paper examines selective label classification with the decision-maker assignment as a multi-class IV. Our point identification Assumption 3.2 must account for multi-class IV and thus strictly generalizes the assumption in Cui and Tchetgen (2021). In Theorem 3.2, we further show that our assumption is the sufficient and necessary condition for the Wald-style identification in Theorem 3.3.  Moreover, our selective classification problem with a multi-class label $Y^*$ is quite different from the classification reformulation of policy learning in Cui and Tchetgen (2021), where $A$ is viewed as the label. While our decision indicator $D$ plays a similar role as the treatment $A$ in Cui and Tchetgen (2021), it only controls the missingness of data. Our real  classification targets is the multi-class outcome $Y^{\star}$, and we cannot view the decision $D$ as a label. Therefore, we address a general multi-class classification problem, unlike the binary classification problem considered in Cui and Tchetgen (2021).
> > >
> > > 3. Our partial identification result also makes several contributions beyond Pu and Zhang (2021). Pu and Zhang (2021) studies policy learning under partial identification with a binary treatment, binary IV and binary outcome. They consider the Balke-Pearl Bound or the Siddique Bounds (with an additional Non-Compliant Decision assumption)  for partial identification, both restricted to the binary setting. Our partial identification bound generalizes the Balke-Pearl bound to a setting with a general multi-class IV and multi-class outcome, in the context of selective label classification. We also prove in our paper that this bound is sharp, meaning it provides the tightest bound under our assumptions. To our knowledge, these results are novel and have not been explored in the existing literature.
> > > Moreover, although both our work and Pu and Zhang (2021) consider minimax learning, our learning problem is more challenging due to the more complex partial identification bounds. Pu and Zhang (2021) deal with a binary outcome, so their inner maximization problem only involves a simple interval constraint on the one-dimensional conditional average treatment effect function and can be easily solved in closed form. In contrast, we consider a multi-class label, so our inner maximization involves a more complex simplex constraint on a vector of conditional probability functions. We carefully analyze this problem structure and provide a closed-form solution using the concept of “realizable” partial bound (see Theorem 4.3). This enables us to transform the minimax learning problem into a more tractable cost-sensitive learning problem. These results are novel and significantly generalize the findings in Pu and Zhang (2021) through refined analyses.
> > >
> > > We hope these explanations can address your concern. In our revision, we will clarify these differences more explicitly in both the literature review and the sections on point and partial identification. Thanks you!

---

### Decision · Program_Chairs · 2025-05-01

**Decision:**

Accept (poster)

**Comment:**

This paper uses the instrumental variable framework from causal inference to analyze the problem of selectively having labels from multiple heterogeneous decision-makers. Though it leverages ideas similar to those of prior work, it differs by considering the general multiclass classification setting (multi IV and multi class) and general classification-calibrated losses in the binary setting. The paper analyzes the identifiability of classification risk using strong assumptions (No Unmeasured Common Effect Modifier) and partial identification under weaker assumptions. A cost-sensitive learning approach is then developed for learning in either setting that is then demonstrated on semi-synthetic data.

The theoretical analyses, practical algorithm, and (synthetic) experimental results in generalizations of previously-investigated settings are strengths of the paper.  Real-world feasibility of the approach, the limitations of semi-synthetic experiments, and the lack of comparative baselines are all weaknesses. Additionally, the reviewers suggest improvements for better organizing the paper (including moving related work discussion from the appendix to the body of the paper).

I concur with the averaged assessment of the reviewers and (weakly) recommend for acceptance.